# Kernel conditional tests from learning-theoretic bounds

**Pierre-François Massiani**[1]   **Christian Fiedler**[1,2,*]   **Lukas Haverbeck**[1]

**Friedrich Solowjow**[1]   **Sebastian Trimpe**[1]

[1] Institute for Data Science in Mechanical Engineering, RWTH Aachen University
[2] Department of Mathematics, School of Computation, Information and Technology,
Technical University of Munich, and Munich Center for Machine Learning (MCML)
`{massiani,christian.fiedler,solowjow,trimpe}@dsme.rwth-aachen.de`
`lukas.haverbeck@rwth-aachen.de`

## Abstract

We propose a framework for hypothesis testing on conditional probability distributions, which we then use to construct *statistical tests of functionals of conditional distributions*. These tests identify the inputs where the functionals differ with high probability, and include tests of conditional moments or two-sample tests. Our key idea is to transform confidence bounds of a learning method into a test of conditional expectations. We instantiate this principle for kernel ridge regression (KRR) with subgaussian noise. An intermediate data embedding then enables more general tests — including *conditional two-sample tests* — via kernel mean embeddings of distributions. To have guarantees in this setting, we generalize existing pointwise-in-time or time-uniform confidence bounds for KRR to previously-inaccessible yet essential cases such as infinite-dimensional outputs with non-trace-class kernels. These bounds also circumvent the need for independent data, allowing for instance online sampling. To make our tests readily applicable in practice, we introduce bootstrapping schemes leveraging the parametric form of testing thresholds identified in theory to avoid tuning inaccessible parameters. We illustrate the tests on examples, including one in process monitoring and comparison of dynamical systems. Overall, our results establish a comprehensive foundation for conditional testing on functionals, from theoretical guarantees to an algorithmic implementation, and advance the state of the art on confidence bounds for vector-valued least squares estimation.

## 1 Introduction and related work

Deciding whether the inputs and outputs of two phenomena obey the same conditional relationship is essential in many areas of science. In control and robotics, one may want to detect changes in the dynamics over time or detect unusual perturbations; in industrial monitoring, to identify the performance of an equipment in varying operating conditions; and in medical studies, to compare treatment responses based on patient characteristics. These examples exhibit the common characteristic that they concern the *conditional distribution* of an output (next state, performance, treatment response,

---

*Work conducted while with the Institute for Data Science in Mechanical Engineering, RWTH Aachen University

39th Conference on Neural Information Processing Systems (NeurIPS 2025).

...) given an input parameter, which we call the *covariate* (current state, patient characteristics, ...). The *functional* of interest is the aspect of the output distribution we care about (expectation, other moment, full distribution, ...). While there are methods to detect global changes or mismatches between (functionals of) probability distributions, they generally have at least one of the following limitations: (i) they already detect a mismatch if the marginal distributions of the covariates differ while the conditional ones coincide [1]; (ii) they already detect a mismatch if they detect a *global* difference between the conditional distributions, disallowing questions such as at what covariate they differ or whether they differ at specific points of interest [2–8]; or (iii) they only provide guarantees in the infinitely-many samples limit [2–9]. In contrast, we propose a framework enabling finite-sample guarantees for conditional hypothesis testing at arbitrary covariates, generalizing such guarantees for classical hypothesis testing in the sense of Neyman-Pearson [10, Sections 8.1–8.3.1]. To our knowledge, we are the first to formalize guarantees on the accuracy of the *covariate rejection region*, which is the set of points where the null hypothesis is rejected. This region and guarantees thereon enable identifying pointwise discrepancies between functionals of conditional distributions, rather than global discrepancies between joint or conditional distributions.

Regardless of the covariate rejection region, conditional testing is notoriously *hard* — there exists no nontrivial test with prescribed level against all alternatives [3, 4]. This should be highly unsurprising to readers familiar with no-free-lunch theorems in statistical learning theory [11], as conditional testing typically requires making (high-probability) statements about conditional distributions at covariates where no samples were observed, and thus requires extrapolation power. We provide a systematic way to escape this negative result by making explicit the ties between conditional testing and learning theory: *learning precedes testing*, as confidence bounds of learning algorithms yield guarantees for conditional tests. This connection enables turning source conditions in statistical learning into "prior sets", that is, classes of alternatives for which a conditional test has guarantees. To our knowledge, we are the first to formalize this connection.

The natural assumption on the data in conditional testing is that outputs are sampled according to conditional distributions given inputs. In particular, outputs are not necessarily *identically distributed*. They may even not be *independent*; e.g., in the online setting [12], where data is generated sequentially as future sampling locations depend on previous measurements. This is a technical challenge for finite-sample guarantees and contrasts starkly with classical hypothesis testing [10, Chapter 8], where data is independent and identically distributed (i.i.d.) [1], approximately i.i.d. (mixing, ...) [13], or at least identically distributed given previous samples (online setting) [14, 15]. A common approach in learning theory is to rely on a measure along which the covariates concentrate [16–18]. We must alleviate such assumptions, however, as we seek answers that are pointwise rather than measure-dependent. We achieve this by relying on *confidence bounds* of learning algorithms, that is, pointwise probabilistic bounds on the error of the learned functions. This allows us not only to provide finite-sample guarantees, but also to operate on non-i.i.d. data, whereas previous studies are confined to asymptotic tests with i.i.d. data [2–9]. From a practical point of view, sequential data generation calls for *repeated testing* as more data becomes available, often requiring time-uniform or anytime confidence bounds [12, 19–21]. The specific test we propose is based on such a bound, which we generalize to previously inaccessible cases thanks to a new assumption we introduce and a strengthening of subgaussianity [22]. In particular, our bound is directly time-uniform.

The learning problems involved in conditional testing require mapping covariates to the functional of interest. In some cases, this may be the full conditional distribution; e.g., in two-sample or independence testing [4]. *Distributional learning* usually addresses this by embedding said distributions as expectations in Hilbert spaces [23, 24]. We leverage this idea by casting general conditional functional test as a comparison of conditional expectations with correspondingly-embedded data. It requires, however, confidence bounds applicable to such outputs. For instance, kernel ridge regression (KRR) admits a well-known time-uniform confidence bound for scalar outputs [12, 21], but its only extension to infinite-dimensional outputs [20] requires strong assumptions on the kernel which exclude the data embeddings that allow testing for complex functionals such as those we identify for two-sample tests. We address this by establishing a time-uniform bound that holds under significantly weaker assumptions — this is our main technical contribution. It is the first bound to hold under such general assumptions to our knowledge, and recovers exactly the ones of Abbasi-Yadkori [12] and Chowdhury and Gopalan [20] in their settings. As a result, it also enjoys the interpretation of being a frequentist correction of the Bayesian uncertainty bound obtained when interpreting KRR as a Gaussian process (GP) posterior mean [25, 26].

Our guarantees depend on quantities common in learning theory but inaccessible in practice. Furthermore, learning-theoretic bounds are commonly conservative as they are worst-case in the prior set, resulting in overly cautious tests. Estimation schemes for sharper bounds are thus essential. To that end, we provide heuristic bootstrapping schemes using the parametric structure of the testing threshold identified in theory. One is adapted from Singh and Vijaykumar [27] by modifying their standard error term in accordance with our results. In principle, they can be replaced by any alternative.

Finally, our approach is connected to several other methods, like multiple testing [28], though the hypotheses we test simultaneously are connected (through the prior set). Due to the uniform bounds, our test also enables sequential testing [29] and drift detection [30, 31], though the latter is generally performed on *marginal distributions*; e.g., by testing for independence between a time process and the process of interest. In addition, our tools can be used to perform anomaly detection [32], which we illustrate on an example, cf. Section 5. Finally, our general framework can be instantiated also to conditional independence testing, which in turn is known to be equivalent to conditional two-sample testing in some instances [4]. Further such investigations are left for future work.

**Contributions and outline**   We provide a general framework for conditional testing and its finite-sample guarantees, and instantiate it for KRR together with bootstrapping schemes. Specifically:

1. We formalize the covariate rejection region and guarantees thereon, laying a basis for finite-sample conditional tests (Section 3).
2. We turn confidence bounds of learning algorithms into statistical tests comparing conditional expectations (Theorem 4.2). Our instantiation for KRR (Theorem 4.4) is the first to enjoy finite-sample guarantees, and also to allow non-i.i.d. data.
3. We specialize the KRR instantiation to a test for functionals of conditional distributions which allows conditional two-sample tests (Section 4.3).
4. We provide and implement heuristic bootstrapping schemes of the resulting test.
5. We prove confidence bounds for KRR with infinite-dimensional outputs under unprecedentedly weak assumptions (Theorem 4.3); specifically, Definition 2.4. The interest of this result exceeds conditional testing.

We expand on Contribution 3 above in Appendix A with its theoretical grounding and supporting experiments. The bootstrapping schemes and their numerical investigation are presented in Appendix B. Explicit algorithms for the test, and computing the effective test statistic and bootstrapped thresholds are in Appendix C. Appendix D further elaborates on how Theorem 4.3 relates to existing bounds in the literature. It relies on the more general Theorem E.23 on vector-valued least squares in separable Hilbert spaces, which we show and present as a standalone result in Appendix E. Appendix F is the proof of Theorem 4.2, and Appendix G has further information on our numerical experiments.

## 2   Preliminaries and notation

The set of positive integers is denoted by $\mathbb{N}_{>0}$, and for all $n \in \mathbb{N}_{>0} \cup \{\infty\}$, $[n]$ is the interval of integers between 1 and $n$, with $[\infty] := \mathbb{N}_{>0}$. If $u$ is a sequence and $m, n \in \mathbb{N}_{>0}$, with $m \leq n$, we write $u_{m:n}$ for the tuple with $n - m + 1$ elements $(u_m, \ldots, u_n)$. If $m = 1$, we simply write $u_{:n}$. If $\mathcal{H}, \mathcal{G}$ are Hilbert spaces, $\mathcal{L}(\mathcal{H}, \mathcal{G})$ (resp., $\mathcal{L}_{\mathrm{b}}(\mathcal{H}, \mathcal{G})$) is the space of linear operators (resp., bounded linear operators) from $\mathcal{H}$ to $\mathcal{G}$. If $\mathcal{G} = \mathcal{H}$, we simply write $\mathcal{L}(\mathcal{H})$ and $\mathcal{L}_{\mathrm{b}}(\mathcal{H})$. Finally, for a self-adjoint $A \in \mathcal{L}_{\mathrm{b}}(\mathcal{H})$, we define $\langle g, h \rangle_A = \langle Ag, h \rangle$, and denote the induced seminorm by $\| \cdot \|_A$.

**Markov kernels, conditional expectations**   We consider a complete probability space $(\Omega, \mathcal{A}, \mathbb{P})$ and input and output measurable spaces $(\mathcal{X}, \mathcal{A}_{\mathcal{X}})$ and $(\mathcal{Y}, \mathcal{A}_{\mathcal{Y}})$, respectively. We assume that $\mathcal{Y} \subset \mathcal{G}$ as metric and measurable spaces, where $\mathcal{G}$ is a separable Hilbert space equipped with its Borel $\sigma$-algebra, and that $\mathcal{Y}$ is closed. We define $\mathcal{M}_1^+(\mathcal{Z})$ as the set of probability measures on a measurable space $\mathcal{Z}$. We identify measures on $\mathcal{Y}$ with their unique extension to $\mathcal{G}$ with support contained in $\mathcal{Y}$. Then, a Markov kernel from $\mathcal{X}$ to $\mathcal{Y}$ is a map $p : \mathcal{A}_{\mathcal{Y}} \times \mathcal{X} \to [0, 1]$ such that $p(\cdot, x)$ is a probability measure on $\mathcal{Y}$ for all $x \in \mathcal{X}$ and $p(A, \cdot)$ is a measurable function for all $A \in \mathcal{A}_{\mathcal{Y}}$ [33]. If $P \in \mathcal{M}_1^+(\mathcal{G})$ is a probability measure such that the identity on $\mathcal{G}$ is (Bochner-)integrable w.r.t. $P$, we introduce its first moment $\mathbb{E}(P) = \int_{\mathcal{Y}} y \, \mathrm{d}P(y) \in \mathcal{G}$. We say that a Markov kernel $p$ from $\mathcal{X}$ to $\mathcal{Y}$ has a first moment if $p(\cdot, x)$ does so for all $x \in \mathcal{X}$. Then, we use the shorthand $\mathbb{E}(p)(x) := \mathbb{E}(p(\cdot, x))$. Finally, we define the prior set $\Theta$ as a set of Markov kernels from $\mathcal{X}$ to $\mathcal{Y}$. Section 3 allows any such choice of $\Theta$, and we specify in Section 4 assumptions on it under which we can provide guarantees.

**Stochastic processes, subgaussianity** A *family of data-generating processes* is a family $D = \{D_p \mid p \in \Theta\}$, where $D_p$ is an $\mathcal{X} \times \mathcal{Y}$-valued process indexed by $\mathbb{N}_{>0}$. Such a $D_p$ is called a *data-generating process*. If, additionally, $D_p := [(X_{p,n}, Y_{p,n})]_{n \in \mathbb{N}_{>0}}$ satisfies

$$\mathbb{P}[Y_{p,n} \in \cdot \mid X_{p,:n}, Y_{p,:n-1}] = p(\cdot, X_{p,n}), \quad \forall n \in \mathbb{N}_{>0} \tag{1}$$

almost surely (a.s.), we say that $D_p$ is a *process of transition pairs*, and that $D$ is *family of processes of transition pairs* if (1) holds for all $p \in \Theta$. Finally, we say that $D_p$ is a process of *independent* transition pairs if (1) holds and $(X_{p,n}, Y_{p,n})_{n \in \mathbb{N}_{>0}}$ is an independent process, and similarly for $D$ itself. For all $n \in \mathbb{N}_{>0}$, we introduce $\mathcal{D}_n = (\mathcal{X} \times \mathcal{Y})^n$ and $\mathcal{D} = \bigcup_{n=1}^{\infty} \mathcal{D}_n$. Elements of $\mathcal{D}$ are called *data sets*, and for $D := [(x_n, y_n)]_{n \in [N]} \in \mathcal{D}$ we define $\mathrm{len}(D) = N$. Finally, a core assumption of our results is subgaussian noise. We consider the strengthening introduced in Mollenhauer and Schillings [22]. In contrast to the reference, however, we do not require that the variance proxy is trace-class, as our results only require a weakening of that condition.

**Definition 2.1.** Let $R \in \mathcal{L}_{\mathrm{b}}(\mathcal{G})$ be self-adjoint, positive semi-definite. We say that a $\mathcal{G}$-valued process $\eta = (\eta_n)_{n \in \mathbb{N}_{>0}}$ adapted to a filtration $\mathcal{F} = (\mathcal{F}_n)_{n \in \mathbb{N}_{>0}}$ is $R$-subgaussian conditionally on $\mathcal{F}$ if

$$\mathbb{E}\left[\exp\left(\langle g, \eta_n \rangle_{\mathcal{G}}\right) \mid \mathcal{F}_n\right] \leq \exp\left(\|g\|_R^2\right), \quad \forall g \in \mathcal{G}, \quad \forall n \in \mathbb{N}_{>0}, \quad \text{a.s.} \tag{2}$$

If (2) holds with $R = \rho \, \mathrm{id}_{\mathcal{G}}$ for some $\rho \in \mathbb{R}_{\geq 0}$, we say that $\eta$ is $\rho$-subgaussian conditionally on $\mathcal{F}$. Similarly, we say that a Markov kernel $p \in \Theta$ from $\mathcal{X}$ to $\mathcal{Y} \subset \mathcal{G}$ is $R$-subgaussian if

$$\int_{\mathcal{Y}} \exp\left(\langle g, y - \mathbb{E}(p)(x) \rangle_{\mathcal{G}}\right) p(\mathrm{d}y, x) \leq \exp\left(\|g\|_R^2\right), \quad \forall g \in \mathcal{G}, \quad \forall x \in \mathcal{X}, \tag{3}$$

and that it is $\rho$-subgaussian if (3) holds with $R = \rho \, \mathrm{id}_{\mathcal{G}}$.

**Kernel ridge regression** We only provide a minimal treatment of reproducing kernel Hilbert spaces (RKHSs) and KRR and refer the reader to dedicated references [34, 35] for details.

**Definition 2.2.** A $\mathcal{G}$-valued RKHS $\mathcal{H}$ on $\mathcal{X}$ is a Hilbert space $(\mathcal{H}, \langle \cdot, \cdot \rangle_{\mathcal{H}})$ of functions from $\mathcal{X}$ to $\mathcal{G}$ such that for all $x \in \mathcal{X}$, the evaluation operator $S_x : f \in \mathcal{H} \mapsto f(x) \in \mathcal{G}$ is bounded. Then, we define[2] $K(\cdot, x) := S_x^\star$ and $K(x, x') = S_x S_{x'}^\star$, for all $x, x' \in \mathcal{X}$. The map $K : \mathcal{X} \times \mathcal{X} \to \mathcal{L}_{\mathrm{b}}(\mathcal{G})$ is called the (operator-valued) (reproducing) kernel of $\mathcal{H}$.

**Theorem 2.3.** *Let $\mathcal{H}$ be a $\mathcal{G}$-valued RKHS with kernel $K$. Then, $K$ is Hermitian, positive semi-definite[3], and the reproducing property holds for all $x \in \mathcal{X}$, $f \in \mathcal{H}$, and $g \in \mathcal{G}$: $\langle f(x), g \rangle_{\mathcal{G}} = \langle f, K(\cdot, x)g \rangle_{\mathcal{H}}$.*

It is well known that, for every Hermitian, positive semi-definite function $K : \mathcal{X} \times \mathcal{X} \to \mathcal{L}_{\mathrm{b}}(\mathcal{G})$, there exists a unique $\mathcal{G}$-valued RKHS of which $K$ is the unique reproducing kernel. The kernel thus fully characterizes the RKHS, and we write $(\mathcal{H}, \langle \cdot, \cdot \rangle_{\mathcal{H}}) =: (\mathcal{H}_K, \langle \cdot, \cdot \rangle_K)$. We say that a kernel is trace-class if $K(x, x')$ is, for all $(x, x') \in \mathcal{X}$. We assume throughout that kernels are strongly measurable. We also introduce the following notion, which is new to the best of our knowledge and generalizes the common choice of a diagonal kernel, $K = k \cdot \mathrm{id}_{\mathcal{G}}$.

**Definition 2.4.** We say that the kernel $K$ of a $\mathcal{G}$-valued RKHS $\mathcal{H}_K$ is *uniform-block-diagonal (UBD)* *(with isometry $\iota_{\mathcal{G}}$)* if there exists separable Hilbert spaces $\tilde{\mathcal{G}}$ and $\mathcal{V}$, an isometry $\iota_{\mathcal{G}} \in \mathcal{L}_{\mathrm{b}}(\tilde{\mathcal{G}} \otimes \mathcal{V}, \mathcal{G})$, and a kernel $K_0 : \mathcal{X} \times \mathcal{X} \to \mathcal{L}_{\mathrm{b}}(\tilde{\mathcal{G}})$ (called an *elementary block*) such that

$$K = \iota_{\mathcal{G}}(K_0 \otimes \mathrm{id}_{\mathcal{V}})\iota_{\mathcal{G}}^{-1}.$$

In the following, we often assume that $K_0$ is trace-class; this is weaker than $K$ itself being trace-class. Informally, uniform-block-diagonality is equivalent to the existence of a base of $\mathcal{G}$ where, for all $(x, x') \in \mathcal{X}^2$, the operator $K(x, x')$ has the block-diagonal, operator-valued matrix representation

$$K(x, x') \approx \begin{pmatrix} K_0(x, x') & 0 & \cdots \\ 0 & K_0(x, x') & \ddots \\ \vdots & \ddots & \ddots \end{pmatrix}$$

---

[2]We emphasize that we deviate from the usual convention as, formally, $K(\cdot, x)$ is *not* defined as the function $x' \in \mathcal{X} \mapsto K(x', x) := S_{x'} S_x^\star \in \mathcal{L}_{\mathrm{b}}(\mathcal{G})$, but as an element of $\mathcal{L}_{\mathrm{b}}(\mathcal{G}, \mathcal{H})$.

[3]Recall that a bivariate function $\phi : \mathcal{X} \times \mathcal{X} \to \mathcal{L}_{\mathrm{b}}(\mathcal{G})$ is Hermitian if $\phi(x, x') = \phi(x', x)^\star$. It is positive semi-definite if for all $n \in \mathbb{N}_{>0}$, $(x_i)_{i=1}^n \in \mathcal{X}^n$, and $(g_i)_{i=1}^n \in \mathcal{G}^n$, $\sum_{i=1}^n \sum_{j=1}^n \langle g_i, \phi(x_i, x_j)g_j \rangle_{\mathcal{G}} \geq 0$.

Finally, given a kernel $K$, data set $D = [(x_n, y_n)]_{n \in [N]} \in \mathcal{D}$, and regularization coefficient $\lambda \in \mathbb{R}_{>0}$, we define $f_{D,\lambda}$ as the unique solution of the KRR problem, which has an explicit form,

$$f_{D,\lambda} = \arg\min_{f \in \mathcal{H}_K} \sum_{n=1}^{N} \|y_n - f(x_n)\|_{\mathcal{G}}^2 + \lambda \|f\|_K^2 = M_D^{\star}(M_D M_D^{\star} + \lambda \mathrm{id}_{\mathcal{G}^N})^{-1} y_{:N}, \qquad (4)$$

where $M_D = \left( h \in \mathcal{H}_K \mapsto (h(x_n))_{n \in [N]} \in \mathcal{G}^N \right) \in \mathcal{L}_{\mathrm{b}}(\mathcal{H}_K, \mathcal{G}^N)$ is the sampling operator.

## 3  Conditional testing

We formalize conditional testing and introduce guarantees on the covariate rejection region.

**Setup and outcome**   The general goal of conditional testing is to decide based on data whether a specific statement about a conditional distribution holds at any user-specified conditioning location $x \in \mathcal{X}$, which we call the *covariate (parameter)*. Formally, a *(conditional) hypothesis* is a map $H : \mathcal{X} \times \Theta \to \{0, 1\}$ such that, for every $p \in \Theta$, the *fulfillment region* of $H$ for $p$, defined as $\varphi_H(p) := \{x \in \mathcal{X} \mid H(x, p) = 1\}$, is measurable. Hypothesis testing then involves two hypotheses $H_0$ and $H_1$, called the *null* and *alternative* hypothesis, respectively. We assume throughout that $H_1(x, p) = \neg H_0(x, p)$, where $\neg$ denotes the logical negation. A *conditional test* is then a measurable map $\mathcal{T} : \mathcal{X} \times \mathcal{D} \to \{0, 1\}$, interpreted as $\mathcal{T}(x, D) = 1$ if, and only if, we reject the null hypothesis $H_0$ at the covariate $x \in \mathcal{X}$ based on the data $D \in \mathcal{D}$.

**Types of guarantees**   The outcome of a conditional test $\mathcal{T}$ is the function $\mathcal{T}(\cdot, D)$, which is equivalently characterized by the covariate rejection region.

**Definition 3.1.** The *covariate rejection region* of a conditional test $\mathcal{T}$ is the subset of $\mathcal{X}$ on which the null hypothesis is rejected based on the data. It is defined for all $D \in \mathcal{D}$ as

$$\chi(D) = \mathcal{T}(\cdot, D)^{-1}(\{1\}) \in \mathcal{A}_{\mathcal{X}}.$$

We emphasize that this definition is finer-grained than most related approaches [2–8], which correspond to taking $\max_{\mathcal{X}} \mathcal{T}(\cdot, D)$ for the outcome of the test. A conditional test has a "perfect" outcome when the covariate rejection region $\chi(D)$ exactly recovers the complement of the fulfillment region of $H_0$, which satisfies $\varphi_0(p)^C = \varphi_1(p)$ and where we introduced the shorthands $\varphi_i := \varphi_{H_i}, i \in \{0, 1\}$. It follows that a test can make two "types" of errors at a covariate $x \in \mathcal{X}$, corresponding respectively to whether the covariate belongs to the sets

$$E_{\mathrm{I}}(p, D) = \chi(D) \cap \varphi_0(p), \quad \text{or} \quad E_{\mathrm{II}}(p, D) = \chi(D)^C \cap \varphi_1(p).$$

We call the first kind a *type* I *error* and the second kind a *type* II *error*; the sets $E_{\mathrm{I}}(p, D)$ and $E_{\mathrm{II}}(p, D)$ are called the *error regions* of the corresponding type. This terminology is consistent with the case of non-conditional hypothesis testing [10]; indeed, $E_{\mathrm{I}}(p, D)$ contains the covariates where we incorrectly reject the null hypothesis, and $E_{\mathrm{II}}(p, D)$ contains those where we fail to reject it. Then, a "guarantee" is a probabilistic bound on the accuracy of $\chi(D)$; that is, on whether one of these regions intersects a region of interest at times of interest.

**Definition 3.2.** Let $D = (D_p)_{p \in \Theta}$ be a family of data-generating processes and $i \in \{\mathrm{I}, \mathrm{II}\}$. The error function of type $i$ for $D$ is the function $\eta_i : \Theta \times \mathcal{A}_{\mathcal{X}} \times 2^{\mathbb{N}_{>0}} \to [0, 1]$ defined as

$$\eta_i(p, \mathcal{S}, \mathcal{N}) = \mathbb{P}[\exists n \in \mathcal{N}, \mathcal{S} \cap E_i(p, D_{p,:n}) \neq \emptyset], \quad \forall (p, \mathcal{S}, \mathcal{N}) \in \Theta \times \mathcal{A}_{\mathcal{X}} \times 2^{\mathbb{N}_{>0}}$$

For $(\mathcal{S}, \mathcal{N}) \in \mathcal{A}_{\mathcal{X}} \times 2^{\mathbb{N}_{>0}}$, we say that the test $\mathcal{T}$ has type $i$ $(\mathcal{S}, \mathcal{N})$-*guarantee* $\alpha \in (0, 1)$ if $\sup_{p \in \Theta} \eta_i(p, \mathcal{S}, \mathcal{N}) \leq \alpha$. A type I $(\mathcal{S}, \mathcal{N})$-guarantee is called an $(\mathcal{S}, \mathcal{N})$-*level*. Finally, if $\mathcal{S} = \{x\}$ or $\mathcal{N} = \{n\}$, we abuse notation in the above definitions and replace the subset by its only element.

We emphasize the difference between having an $(x, \cdot)$-guarantee for all $x \in \mathcal{X}$ and an $(\mathcal{X}, \cdot)$-guarantee. The former allows using the test at a *single* arbitrary covariate, whereas the latter guarantees the performance when using the test *jointly* on arbitrary covariates and enables trusting the rejection region as a whole. Similarly, $(\cdot, n)$-guarantees for all $n \in \mathbb{N}_{>0}$ allow for a one-time use of the test at an arbitrary time $n \in \mathbb{N}_{>0}$, whereas a $(\cdot, \mathbb{N}_{>0})$-guarantee allows for sequential testing. Our tests have $(\mathcal{X}, \mathbb{N}_{>0})$- or $(\mathcal{X}, n)$-guarantees, depending on which assumptions are made.

While the formal definitions of type I and type II error functions are symmetric, their *reasonable targets* are not. Type I errors are those we seek to control; their frequency must be less than the prescribed level everywhere in $\mathcal{S}$, including in regions with little data. A false rejection of $H_0$ must not be caused by lack of information. As a result, control of type II errors is only achievable in regions of the covariate space where samples provide sufficient information (either through local sample mass or extrapolation enabled by the prior set). In other words, the type II error function can only be low where the data actually enable distinguishing the alternative from the null.

## 4  Testing of conditional expectations and functionals

We specialize the preceding developments to *tests of conditional expectations* with finite-sample guarantees. We are interested in testing the null hypothesis[4]

$$H_0(x, p_1, p_2) : \mathbb{E}(p_1)(x) = \mathbb{E}(p_2)(x) \quad \text{v.s.} \quad H_1(x, p_1, p_2) : \mathbb{E}(p_1)(x) \neq \mathbb{E}(p_2)(x), \quad (5)$$

where $(p_1, p_2) \in \Theta_1 \times \Theta_2$ and $\Theta_i$ is a set of Markov kernels from $\mathcal{X}$ to $\mathcal{Y}$ with first moments, $i \in \{1, 2\}$. First, we provide an abstract result to go from *confidence bounds* of a learning algorithm to a test of conditional expectations (Theorem 4.2). We then specialize it to KRR in the well-specified case, obtaining two tests (Theorem 4.4) with similar thresholds but differing guarantees and assumptions; they are based on new confidence bounds for KRR which we present first (Theorem 4.3). We finally show how to specialize the test to functionals other than the expectation, allowing (among others) a *conditional two-sample test*. This last point is further developed in Appendix A.

*Remark* 4.1. The hypothesis (5) can be equivalently reformulated into the setup of Section 3, but we forego doing it explicitly for brevity. We emphasize, however, that we abuse notation and redefine the test $\mathcal{T}$, covariate rejection region $\chi$, error regions $E_i$, and error functions $\eta_i$, $i \in \{\mathrm{I}, \mathrm{II}\}$, to take as inputs (where applicable) two data sets $D_1$ and $D_2$, Markov kernels $p_1$ and $p_2$, and sets of integers of interest $\mathcal{N}_1$ and $\mathcal{N}_2$. Specifically, they are now defined as $\chi(D_1, D_2) = \mathcal{T}(\cdot, D_1, D_2)^{-1}(\{1\})$, $E_{\mathrm{I}}(p_1, p_2, D_1, D_2) = \chi(D_1, D_2) \cap \varphi_0(p_1, p_2)$, $E_{\mathrm{II}}(p_1, p_2, D_1, D_2) = \chi(D_1, D_2)^C \cap \varphi_1(p_1, p_2)$, and

$$\eta_i(p_1, p_2, \mathcal{S}, \mathcal{N}_1, \mathcal{N}_2) = \mathbb{P}\left[\exists (n_1, n_2) \in \mathcal{N}_1 \times \mathcal{N}_2, \, \mathcal{S} \cap E_i(p_1, p_2, D_{p_1, :n_1}^{(1)}, D_{p_2, :n_2}^{(2)}) \neq \emptyset\right],$$

respectively, where $i \in \{\mathrm{I}, \mathrm{II}\}$ and $D^{(j)} = (D_p^{(j)})_{p \in \Theta_j}$, $j \in \{1, 2\}$, are the considered families of data-generating processes.

### 4.1  From confidence bounds to hypothesis testing

We now provide an abstract result that connects finite-sample confidence bounds of a learning algorithm to type I guarantees for a conditional two-sample test. For this, recall that a learning method $\mathfrak{L}$ is a map from data sets $D \in \mathcal{D}$ to functions $f_D : \mathcal{X} \to \mathcal{Y}$.

**Theorem 4.2.** *For $i \in \{1, 2\}$, let $\mathfrak{L}_i : D \in \mathcal{D} \mapsto f_D^{(i)}$ be a measurable[5] learning method, and $D^{(i)}$ be a family of data-generating processes. For $\mathcal{S} \in \mathcal{A}_\mathcal{X}$, $\mathcal{N}_i \subseteq \mathbb{N}_{>0}$, and $\delta_i \in (0, 1)$, let $B_i := B_i^{(\delta_i, \mathcal{S}, \mathcal{N}_i)} : \mathcal{D} \times \mathcal{X} \to \mathbb{R}_{>0}$ be a measurable function such that for all $p \in \Theta_i$*

$$\mathbb{P}\left[\forall (x, n) \in \mathcal{S} \times \mathcal{N}_i, \, \left\|f_{D_{p, :n}^{(i)}}^{(i)}(x) - \mathbb{E}(p)(x)\right\| \leq B_i\left(D_{p, :n}^{(i)}, x\right)\right] \geq 1 - \delta_i, \quad \forall p \in \Theta_i. \quad (6)$$

*Let $\mathcal{T} := \mathcal{T}_{(B_1, B_2)}$ be the test defined as*

$$\mathcal{T} : (x, D_1, D_2) \in \mathcal{X} \times \mathcal{D} \times \mathcal{D} \mapsto \begin{cases} 1, & \text{if } \|f_{D_1}^{(1)}(x) - f_{D_2}^{(2)}(x)\| > \sum_{i=1}^2 B_i\left(D_i, x\right), \\ 0, & \text{otherwise.} \end{cases} \quad (7)$$

*Then, $\mathcal{T}$ is a test of $H_0$ with $(\mathcal{S}, \mathcal{N}_1, \mathcal{N}_2)$-level $\delta_1 + \delta_2$ when used on the families $D^{(1)}$ and $D^{(2)}$.*

The proof is in Appendix F. In more intuitive terms, this theorem guarantees that if one has two learning methods that come with associated confidence bounds (6) for a specific data-generating process (given by the functions $B_i$) uniformly in $\Theta$, then one can leverage these intervals for a conditional two-sample test with (covariate-dependent) statistic $\|f_{D_1}^{(1)}(x) - f_{D_2}^{(2)}(x)\|$. This result is very natural, and its value resides in clarifying the simple relationship between estimation and testing.

---

[4]Formally, $H_0(x, p_1, p_2) = 1$ if, and only if, $\mathbb{E}(p_1)(x) = \mathbb{E}(p_2)(x)$.

[5]See Steinwart and Christmann [11, Definition 6.2].

## 4.2 A kernel test of conditional expectations

We now turn our attention to a specific learning method for which we show that the assumptions of Theorem 4.2 hold under sufficient regularity of the elements of $\Theta$. Specifically, we consider KRR with kernel $K : \mathcal{X} \times \mathcal{X} \to \mathcal{L}_{\mathrm{b}}(\mathcal{G})$, as introduced in Section 2. The estimate $f_{D,\lambda}$ for a data set $D \in \mathcal{D}$ and regularization coefficient $\lambda > 0$ is given by (4).

### 4.2.1 Confidence bounds for vector-valued KRR

Section 4.1 has established that it suffices to provide confidence bounds for KRR with $\mathcal{G}$-valued outputs. We provide such bounds in the following result.

**Theorem 4.3.** *Let $\mathfrak{L} : D \in \mathcal{D} \mapsto f_D$ be KRR with regularization coefficient $\lambda > 0$ and kernel $K : \mathcal{X} \times \mathcal{X} \to \mathcal{L}_{\mathrm{b}}(\mathcal{G})$. Let $p$ be a Markov kernel from $\mathcal{X}$ to $\mathcal{G}$ such that $\mathbb{E}(p) \in \mathcal{H}_K$, $S \in \mathbb{R}_{>0}$ be such that $\|\mathbb{E}[p]\|_K \le S$, and $R \in \mathcal{L}_{\mathrm{b}}(\mathcal{G})$ be self-adjoint and positive semi-definite. Let $D = [(X_n, Y_n)]_n$ be a process of transition pairs of $p$. Introduce the usual notations for evaluations of the kernel on sequences:*

$$
K(\cdot, X_{:n}) = \left( (g_m)_{m \in [n]} \in \mathcal{G}^n \mapsto \sum_{m=1}^n K(\cdot, X_m) g_m \in \mathcal{H}_K \right) \in \mathcal{L}_{\mathrm{b}}(\mathcal{G}^n, \mathcal{H}_K),
$$
$$
K(x, X_{:n}) = K(\cdot, x)^\star K(\cdot, X_{:n}), \quad K(X_{:n}, x) = K(x, X_{:n})^\star,
$$
$$
K(X_{:n}, X_{:n}) = K(\cdot, X_{:n})^\star K(\cdot, X_{:n}),
$$

(8)

*for all $x \in \mathcal{X}$ and $n \in \mathbb{N}_{>0}$, as well as the shorthands*

$$
\Sigma_{D_{:n},\lambda}(x) = K(x,x) - K(x, X_{:n})(K(X_{:n}, X_{:n}) + \lambda \mathrm{id}_{\mathcal{G}^N})^{-1} K(X_{:n}, x),
$$
$$
\text{and } \sigma_{D_{:n},\lambda}(x) = \sqrt{\|\Sigma_{D_{:n},\lambda}(x)\|_{\mathcal{L}_{\mathrm{b}}(\mathcal{H}_K)}}.
$$

(9)

*Assume that $p$ is $R$-subgaussian, and assume one of the following and define $\beta_\lambda$ and $\mathcal{N}$ accordingly:*

1. *the kernel $K$ is UBD with trace-class elementary block $K_0$ and isometry $\iota_{\mathcal{G}} \in \mathcal{L}_{\mathrm{b}}(\tilde{\mathcal{G}} \otimes \mathcal{V}, \mathcal{G})$, and there exist $\rho \in \mathbb{R}_{>0}$ and $R_\mathcal{V} \in \mathcal{L}_{\mathrm{b}}(\mathcal{V})$ of trace class such that $R = \rho^2 \iota_{\mathcal{G}}(\mathrm{id}_{\tilde{\mathcal{G}}} \otimes R_\mathcal{V}) \iota_{\mathcal{G}}^{-1}$. Then, define $\mathcal{N} = \mathbb{N}_{>0}$ and, for all $(n, \delta) \in \mathbb{N}_{>0} \times (0,1)$,*

$$
\beta_\lambda(D_{:n}, \delta) = S + \frac{\rho}{\sqrt{\lambda}} \sqrt{2 \mathrm{Tr}(R_\mathcal{V}) \ln\left[ \frac{1}{\delta} \left\{ \det\left( \mathrm{id}_{\tilde{\mathcal{G}}^n} + \frac{1}{\lambda} K_0(X_{:n}, X_{:n}) \right) \right\}^{1/2} \right]}; \quad (10)
$$

2. *$R$ is of trace class, and $D$ is a process of* independent *transition pairs of $p$. Then, define $\mathcal{N} = \{N\}$ for some $N \in \mathbb{N}_{>0}$ and, for all $(n, \delta) \in \mathbb{N}_{>0} \times (0,1)$,*

$$
\beta_\lambda(D_{:n}, \lambda) = S + \frac{1}{\sqrt{\lambda}} \sqrt{\mathrm{Tr}(T_{D_{:n},\lambda}) + 2\sqrt{\ln\left( \frac{1}{\delta} \right)} \|T_{D_{:n},\lambda}\|_2 + 2 \ln\left( \frac{1}{\delta} \right) \|T_{D_{:n},\lambda}\|_{\mathcal{L}_{\mathrm{b}}(\mathcal{H})}},
$$

(11)

*where we introduced $T_{D_{:n},\lambda} = (V_{D_{:n}} + \lambda \mathrm{id}_{\mathcal{H}_K})^{-1/2} K(\cdot, X_{:n})(R \otimes \mathrm{id}_{\mathbb{R}^n}) K(\cdot, X_{:n})^\star (V_{D_{:n}} + \lambda \mathrm{id}_{\mathcal{H}_K})^{-1/2}$, with $V_{D_{:n}} = K(\cdot, X_{:n}) K(\cdot, X_{:n})^\star$.*

*In both cases, for all $\delta \in (0,1)$, it holds that*

$$
\mathbb{P}\left[ \forall n \in \mathcal{N}, \forall x \in \mathcal{X}, \|f_{D_{:n}}(x) - \mathbb{E}[p](x)\|_{\mathcal{G}} \le \beta_\lambda(D_{:n}, \delta) \cdot \sigma_{D_{:n},\lambda}(x) \right] \ge 1 - \delta.
$$

For the readers familiar with GP regression [36, 37, 26], this result guarantees that the graph of the ground truth lies in a tube centered at the GP posterior mean with width the posterior variance scaled by $\beta_\lambda$, with high probability. The bound is *frequentist*, however; the standard Bayesian assumption of a GP prior is replaced by the one that the conditional expectation lies in the RKHS. This result follows from Theorem E.23, which provides confidence bounds for vector-valued least-squares (not necessarily in an RKHS) under general assumptions. Theorem 4.3 and its generalization Theorem E.23 are — together with the general setup of Section 3 — the main theoretical contributions of this work. A detailed discussion of the connections between Theorems 4.3 and E.23 and similar bounds is available in Appendix D.

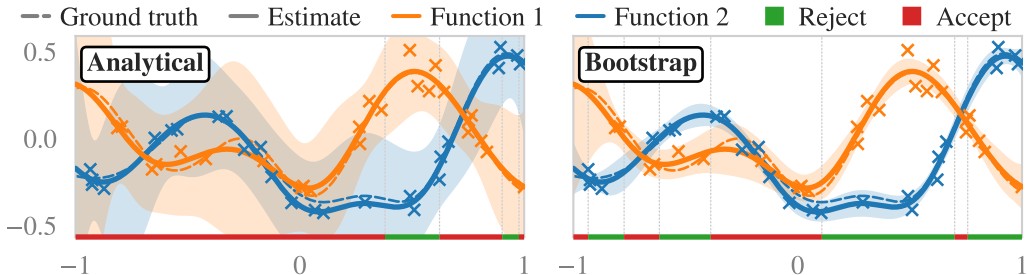

Figure 1: Analytical and bootstrapped tests on a toy example. The tests reject $H_0$ (green thick line) when the confidence intervals (shaded regions) do not overlap, and accept it (red thick line) otherwise.

#### 4.2.2 Application to testing

We can immediately apply the previous bounds to obtain kernel-based tests.

**Theorem 4.4.** *In the setup of Theorem 4.2 and for all $i \in \{1, 2\}$, assume that $\mathfrak{L}_i$ is KRR with regularization coefficient $\lambda_i > 0$ and kernel $K : \mathcal{X} \times \mathcal{X} \to \mathcal{L}_b(\mathcal{G})$. Also assume that there exists $S_i > 0$ and $R_i \in \mathcal{L}_b(\mathcal{G})$, self-adjoint, positive semi-definite, such that any $p \in \Theta_i$ is $R_i$-subgaussian and $\mathbb{E}(p) \in \mathcal{H}_K$ with $\|\mathbb{E}(p)\|_K \leq S_i$. Let $D^{(i)}$ be a family of processes of transition pairs, and introduce the shorthands (9). Let $\alpha_i \in (0, 1)$. Assume one of the following and define $\beta_i$, $B_i$, and $\mathcal{N}_i$ accordingly:*

- (i) *The assumptions of Case 1 in Theorem 4.3 hold for $R_i$ and $K$. Then, define $\mathcal{N}_i = \mathbb{N}_{>0}$, and take $B_i(D, x) = \beta_i(D, \alpha_i) \cdot \sigma_{D,\lambda_i}(x)$, for all $(D, x) \in \mathcal{D} \times \mathcal{X}$, where $\beta_i := \beta_{\lambda_i}$ is defined in (10) with the corresponding choice of parameters.*

- (ii) *The assumptions of Case 2 in Theorem 4.3 hold for $R_i$ and $D_p^{(i)}$ for all $p \in \Theta$. Then, define $\mathcal{N}_i = \{N_i\}$ for some $N_i \in \mathbb{N}_{>0}$, and take $B_i(D, x) = \beta_i(D, \alpha_i) \cdot \sigma_{D,\lambda_i}(x)$, for all $(D, x) \in \mathcal{D} \times \mathcal{X}$, where $\beta_i := \beta_{\lambda_i}$ is defined in (11) with the corresponding choice of parameters.*

*Then, with this choice for $B_i$ in (6) for all $i \in \{1, 2\}$, (7) is a test of $H_0$ with $(\mathcal{X}, \mathcal{N}_1, \mathcal{N}_2)$-level $\alpha_1 + \alpha_2$ when used on the families $D^{(1)}$ and $D^{(2)}$.*

This result formalizes the natural idea of comparing the norm difference between outcomes of KRR ran on both data sets to the sum of the (appropriately scaled) posterior variances of the associated Gaussian processes. The test is not fully distribution-free [1] since it requires subgaussian Markov kernels, but allows data generated online. It also allows *pointwise* testing with guarantees — regardless of the input data distribution — thanks to the RKHS membership assumption. In practice, we leverage the bootstrapping schemes described in Appendix B to avoid tuning $\beta_i$.

### 4.3 From conditional expectations to functionals

We now consider the announced case where we are interested in testing for equality of a functional $\mathcal{F}$ of the conditional distributions; that is, we are interested in a test of the null hypothesis

$$H_0(x, p_1, p_2) : \mathcal{F}(p_1(\cdot, x)) = \mathcal{F}(p_2(\cdot, x)) \quad \text{v.s.} \quad H_1(x, p_1, p_2) : \mathcal{F}(p_1(\cdot, x)) \neq \mathcal{F}(p_2(\cdot, x)).$$
(12)

Examples involve $\mathcal{F}$ being moments of the distributions, or even the identity. Our core idea is to make (12) amenable to the abstract test of Theorem 4.2 by reformulating it as a test of conditional expectations. Specifically, we make the following assumption on $\mathcal{F}$. For consistency, we assume in this section that $p_1$ and $p_2$ are Markov kernels from $(\mathcal{X}, \mathcal{A}_{\mathcal{X}})$ to a measurable set $(\mathcal{Z}, \mathcal{A}_{\mathcal{Z}})$ which we call the *measurement set*, instead of mapping to $\mathcal{Y}$ directly.

**Assumption 4.5.** *There exists a separable Hilbert space $\mathcal{G}$ and a map $\Phi_{\mathcal{F}} : \mathcal{Z} \to \mathcal{G}$ called a representation map of $\mathcal{F}$ such that $\mathcal{Y} := \Phi_{\mathcal{F}}(\mathcal{Z}) \subset \mathcal{G}$ is closed, $\Phi_{\mathcal{F}}$ is Bochner-integrable w.r.t. any probability measure on $\mathcal{Z}$, and*

$$\forall (P, Q) \in \mathcal{M}_1^+(\mathcal{Z})^2, \left( \mathcal{F}(P) = \mathcal{F}(Q) \quad \Longleftrightarrow \quad \int_{\mathcal{Z}} \Phi_{\mathcal{F}}(z) \mathrm{d}P(z) = \int_{\mathcal{Z}} \Phi_{\mathcal{F}}(z) \mathrm{d}Q(z) \right).$$

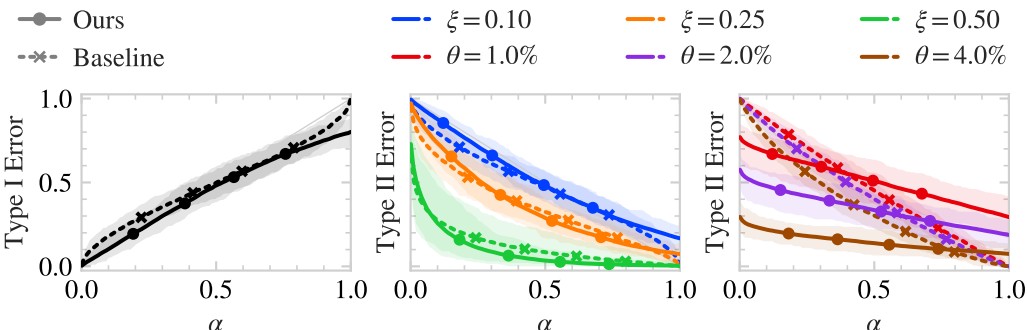

Figure 2: Empirical type I or II errors. **Left:** Both tests uphold and achieve the level under $H_0$. **Middle:** Larger difference $\xi$ between conditional expectations decrease the type II error. **Right:** Our local test can confidently detect a difference in rarely (probability $\theta$) sampled regions.

Under Assumption 4.5, (12) reduces to (5) where the Markov kernels involved are the pushforward kernels of $p_1$ and $p_2$ through $\Phi_{\mathcal{F}}$, which we denote with $q_1$ and $q_2$. Crucially, if a process of transition pairs $D^{(i)} := ((X_n, Z_n))_{n \in \mathbb{N}_{>0}}$ of $p_i$ is available, the modified process $D^{(i)}_{\Phi_{\mathcal{F}}} := ((X_n, \Phi_{\mathcal{F}}(Z_n)))_{n \in \mathbb{N}_{>0}}$ is of transition pairs of $q_i$. In other words, any learning method with guarantees for $\mathcal{G}$-valued outputs can be used to test (12) via the data transformation $Y_n = \Phi_{\mathcal{F}}(Z_n)$.

Assumption 4.5 is satisfied for many functionals of common interest, such as the ones extracting the moments of distributions or the identity — the latter makes (12) a *conditional two-sample test*. The test of Theorem 4.4 remains tractable even with infinite-dimensional $\mathcal{G}$: its complexity is exactly that of KRR. Furthermore, its assumptions such as subgaussianity are now on the pushforward kernels $q_i$ instead of the kernels $p_i$, $i \in \{1, 2\}$. We detail this and provide numerical examples in Appendix A.

## 5 Numerical results

We begin with an example illustrating the different components. We then evaluate performance (type I and II errors) in controlled settings compared to the baseline of Hu and Lei [5]. Next, we illustrate benefits of our test and its pointwise answers compared to global conditional tests, thanks to the covariate rejection region and the fact that we obtain lower type II error when the tested functions differ only in rarely-sampled covariate regions. Finally, we showcase an application on change detection for a linear dynamical system. We remind the reader that Appendices A and B contain further numerical studies respectively comparing more general functionals (such as the two-sample one) and investigating our bootstrapping schemes. Finally, we emphasize that the purpose of this numerical study is *not* to establish dominance over other methods, but to investigate numerically how the framework works in well-understood scenarios. Additional information on experiments, hyperparameters, and the link to the code are in Appendix G.

**Metrics and setup** We refer below to "type I or II errors" and "positive rates" without referring to the corresponding triple $(\mathcal{S}, \mathcal{N}_1, \mathcal{N}_2)$. They are meant as $(\mathcal{S}, n, n)$-errors of type I or II and as the positive rate of the test triggering anywhere in the subset of interest $\mathcal{S}$ with data sets of lengths $n \in \mathbb{N}_{>0}$. We approximate triggering anywhere in $\mathcal{S}$ by evaluating the test at the locations present in the data set. A trigger is a type II error if it occurs in regions where $H_0$ is not enforced. We say we "pick" a function when we randomly choose it from the unit sphere of the RKHS $\mathcal{H}_K$ on $\mathcal{X}$. Except in the last experiment, $K$ is the scalar Gaussian kernel with bandwidth $\gamma^2 = 0.25$ [38]. Unless stated, data sets have 100 uniformly-sampled points, and all results use the "naive" bootstrap (Algorithm 2).

**Illustrative example** With $\mathcal{X} = [-1, 1]$, we pick two functions taking the role of conditional means to compare and generate two data sets of 25 noisy measurements. Figure 1 shows the outcome of KRR and of the test with analytical (left) and bootstrapped (right) confidence intervals ($\alpha = 0.05$). The test rejects the null hypothesis exactly where the confidence intervals do not overlap. The bootstrapped intervals are much tighter, yielding higher power. Though we do not guarantee it, we find empirically that the true functions lie in the bootstrapped shaded region.

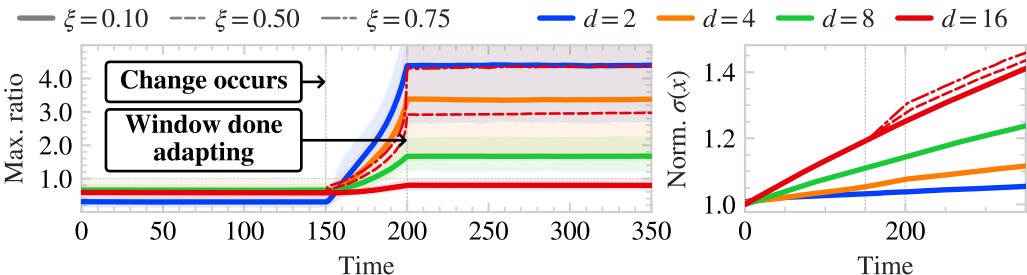

Figure 3: **Left:** Ratio between test statistic and threshold for the process monitoring example for various dimensions ($d$) and perturbation magnitudes ($\xi$). Triggers correspond to exceeding 1. Solid lines are means; shaded regions are 5% and 95% quantiles. **Right:** Ratio of $\sigma_{D,\lambda}(x)$ in (9) on the reference data $D$ between $x$ in the window and the initial state at $t = 0$, averaged over the window.

**Empirical type I and II errors** With $\mathcal{X} = [-1,1]^2$, we pick $f_1$ and $f_2$, with $\|f_2\|_K = \xi$. We perform the boostrapped test between noisy data sets twice: first, with both sets generated from $f_1$ (evaluating the type I error), and then with one set generated from $f_1$ and the other from $f_1 + f_2$ (evaluating the type II error). Figure 2 (left, middle) shows the empirical type I and II errors as functions of $\alpha$ and $\xi$, as well as the baseline [5] parameterized by density estimators using the same KRR estimate as ours, together with the ground truth noise distribution. We uphold the type I level everywhere, have similar type I and II errors as the baseline *without* informing our test of the noise distribution, and achieve decreasing type II errors with increasing function difference.

**Local sensitivity** We now set $\mathcal{X} = [-3,3]^2$, pick $f_1$, and set $f_2(\cdot) = k(\cdot, 0)$. We generate data from $f_1$ and $f_1 + f_2$, which differ on $\mathcal{X}_{\text{diff}} := \{x \in \mathcal{X} \mid f_2(x) \geq 10^{-2}\}$ and (approximately) coincide on $\mathcal{X}_{\text{same}} := \mathcal{X} \setminus \mathcal{X}_{\text{diff}}$. Data is sampled via a mixture of uniform distributions on $\mathcal{X}_{\text{same}}$ and on $\mathcal{X}_{\text{diff}}$. Figure 2 (right) shows the type II error as a function of $\theta$, the weight of $\mathcal{X}_{\text{diff}}$ in the mixture. Densities are not estimated but set to their ground truth for the baseline [5]. Still, our test achieves higher power for low values of $\theta$. This directly results from the locality of our tests (whereas the baseline computes one aggregated statistic), which is enabled by the local guarantees of learning-theoretic bounds.

**Process monitoring** We illustrate how the test allows dependent sampling by testing on the trajectories of a linear dynamical system on $\mathcal{X} = \mathbb{R}^d$; the goal is to detect a change in the dynamics. We consider a fixed reference data set of the transition pairs of 5 trajectories of length 400 with a fixed initial state. We compare it to a sliding window of length 50 of the transition pairs of the current trajectory by performing the test at every point in the window. We perturb the dynamics after 200 steps with a magnitude controlled monotonically by a scalar parameter $\xi$. The results are shown in Fig. 3. We successfully detect the change reliably, even before the window is completely filled with data from the perturbed dynamics. This supports that the test performs well despite correlations in sampling locations. We hypothesize based on Fig. 3 (right) and the interpretation of $\sigma_{D,\lambda}(x)$ as a GP posterior variance [37] that the degraded performance for a fixed $\xi$ as $d$ increases is mainly due to the perturbed dynamics driving the system in regions underexplored before the perturbation.

## 6  Conclusion

We rigorously introduce a framework for testing conditional distributions with local guarantees. The framework is broadly applicable and provides a general recipe for turning learning-theoretic bounds into statistical tests. We instantiate it for conditional two-sample testing with KRR, enabling the comparison of not only expectations but other functionals such as arbitrary moments or full conditional distributions. The guarantees are based on a generalization we show of popular confidence bounds for KRR and regularized least squares to accommodate UBD kernels. To our knowledge, this is the first generalization to support such a broad class of possibly non-trace-class kernels. We present a complete pipeline from conceptual foundations and theoretical analysis to an algorithmic implementation, and validate the approach through numerical experiments. Our framework offers a principled foundation for developing powerful, flexible, and theoretically grounded conditional tests, laying a basis for a wide range of future applications or theoretical investigations.

## Acknowledgment

Friedrich Solowjow is supported by the German federal state of North Rhine-Westphalia through the KI-Starter grant. Christian Fiedler acknowledges support from DFG Project FO 767/10-2 (eBer-24-32734) "Implicit Bias in Adversarial Training".

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

# A    Elaboration on functional tests

The goal of this section is to expand on the observation of Section 4.3 that Theorem 4.2 provides a test for the generalized hypothesis $H_0$ defined in (12) for any functional $\mathcal{F}$ satisfying Assumption 4.5. Specifically, we show and explain the following:

1. There are many functionals $\mathcal{F}$ that satisfy Assumption 4.5.

2. There is a general catalog of representation maps $\Phi_{\mathcal{F}}$ of functionals capturing specific properties of interest of the Markov kernels — such as moments or full distributions. The catalog consists of kernel functions, and the properties are the ones captured by the corresponding kernel mean embeddings (KMEs). This is summarized in Observation A.2.

3. Using KRR with a diagonal kernel $K = k \cdot \mathrm{id}_{\mathcal{G}}$ as the underlying learning algorithm enables computing of the thresholds and statistics even when the space $\mathcal{G}$ is infinite-dimensional. This last point is the reason why it is essential that Theorem 4.3 applies to infinite-dimensional outputs with non-trace-class kernels.

Together, these explanations show that we can specialize the test of conditional expectations into a test of more complex functionals of Markov kernels — including a conditional two-sample test — while preserving computational feasibility. Indeed, the resulting computational cost is equivalent to that of KRR (with the chosen bootstrapping scheme, if thresholds are boostrapped). We conclude this section with a numerical study highlighting the effectiveness of the test for different functionals.

We now study whether there are many functionals of interest that satisfy Assumption 4.5, and how to pick or find the corresponding map $\Phi_{\mathcal{F}}$. We begin with an example exhibiting a suitable map $\Phi_{\mathcal{F}}$ for the functional $\mathcal{F}$ extracting the conditional expectation and variance, and see how to leverage KMEs of distributions to capture a large class of functionals with explicit maps $\Phi_{\mathcal{F}}$, as it is required in the test procedure. Then, we explain how using KRR with a diagonal kernel together with this idea yields the popular framework of conditional mean embeddings (CMEs) [38], enabling tractable computations even when $\mathcal{G}$ is infinite-dimensional. We conclude this section with a numerical study highlighting how different choices of $\mathcal{F}$ and $\Phi_{\mathcal{F}}$ affect the test outcome.

## A.1    Example: conditional expectation and variance

We begin with an example exhibiting a suitable representation map $\Phi_{\mathcal{F}}$ of the functional $\mathcal{F}$ extracting the conditional expectation and variance. Assume that $\mathcal{Z} \subset \mathbb{R}$, and that $p_1(\cdot, x)$ and $p_2(\cdot, x)$ have well-defined first and second moments for all $x \in \mathcal{X}$. This is for instance the case if $\mathcal{Z}$ is bounded. Introduce for $i \in \{1, 2\}$ the first two conditional moments:

$$\mu_1^{(i)}(x) = \mathbb{E}_{Z \sim p_i(\cdot, x)}[Z], \quad \text{and} \quad \mu_2^{(i)}(x) = \mathbb{E}_{Z \sim p_i(\cdot, x)}[Z^2],$$

for all $x \in \mathcal{X}$. Now define the functional $\mathcal{F}$ that maps a measure to the vector of its first two moments. For $i \in \{1, 2\}$, we have

$$\begin{aligned}
\mathcal{F}(p_i(\cdot, x)) &:= \begin{pmatrix} \mu_1^{(i)}(x) \\ \mu_2^{(i)}(x) \end{pmatrix} \\
&= \begin{pmatrix} \mathbb{E}_{Z \sim p_i(\cdot, x)}[Z] \\ \mathbb{E}_{Z \sim p_i(\cdot, x)}[Z^2] \end{pmatrix} \\
&= \begin{pmatrix} \int_{\mathcal{Z}} z \, p_i(\mathrm{d}z, x) \\ \int_{\mathcal{Z}} z^2 \, p_i(\mathrm{d}z, x) \end{pmatrix} \\
&= \int_{\mathcal{Z}} \Phi_{\mathcal{F}}(z) p_i(\mathrm{d}z, x),
\end{aligned}$$

where we introduced $\Phi_{\mathcal{F}} : z \in \mathcal{Z} \mapsto \begin{pmatrix} z & z^2 \end{pmatrix}^{\top}$. This shows that the functional $\mathcal{F}$ satisfies Assumption 4.5 with $\mathcal{G} = \mathbb{R}^2$.

Let us now spell out the method summarized in Section 4.3 to obtain a test of equality of the first two moments. Assuming that one has access to processes of transition pairs $D^{(i)} = ((X_n^{(i)}, Z_n^{(i)}))_{n \in \mathbb{N}_{>0}}$,

replace them with the processes

$$D_{\Phi_{\mathcal{F}}}^{(i)} = ((X_n^{(i)}, Y_n^{(i)}))_{n \in \mathbb{N}_{>0}} := ((X_n^{(i)}, \Phi_{\mathcal{F}}(Z_n^{(i)})))_{n \in \mathbb{N}_{>0}} = \left( \left( X_n^{(i)}, \begin{pmatrix} Z_n^{(i)} \\ Z_n^{(i)^2} \end{pmatrix} \right) \right)_{n \in \mathbb{N}_{>0}},$$

and use a learning algorithm to learn the map from $(X_n^{(i)})_n$ to $(Y_n^{(i)})_n$ for each $i \in \{1, 2\}$ (formally, the learning algorithm should seek to approximate the conditional expectation function of $Y_n^{(i)}$ conditioned on $X_n^{(i)}$, which we assume is independent of $n$). As a consequence of the data augmentation step $y = \Phi_{\mathcal{F}}(z)$, notice that the output of this learning algorithm should be 2-dimensional. Theorem 4.2 then guarantees that comparing the distance between the predictions of the two models to the sum of confidence bounds on the models' accuracy provides a test with prescribed level of the hypothesis that the first and second moments of $p_1$ and $p_2$ coincide.

## A.2   Generalization via kernel mean embeddings of distributions

We now generalize the ideas of the above example to *find* appropriate representation maps $\Phi_{\mathcal{F}}$ of functionals of interest, as the testing procedure requires knowing these maps. The generalization is based on the observation that, for a rich class of functions $\Phi : \mathcal{Z} \to \mathcal{G}$, the integrals involved in Assumption 4.5 are well-studied objects and capture specific properties of the underlying distributions. Furthermore, we argue that *any* functional $\mathcal{F}$ satisfying Assumption 4.5 has a feature map belonging to the aforementioned class (Proposition A.1), justifying restricting our interest to this class.

Let us begin with this second point and assume that we have a map $\Phi : \mathcal{Z} \to \mathcal{G}$ available, where $\mathcal{G}$ is a Hilbert space, and let $\mathcal{F}$ be a functional of which $\Phi$ is a representation map. Another name for such a $\Phi$ is a *feature map* on $\mathcal{Z}$, as it can be thought of as extracting information from a point $z \in \mathcal{Z}$ and representing it in $\mathcal{G}$. Readers familiar with RKHS theory may now recognize that the pair $(\Phi, \mathcal{G})$ is a feature map/feature space pair, and it defines a unique RKHS on $\mathcal{Z}$ via the usual canonical isomorphism between a feature space and the associated RKHS [11, Theorem 4.21]. In more practical terms, the function $\Phi$ defines a kernel

$$\kappa(z, z') = \langle \Phi(z), \Phi(z') \rangle_{\mathcal{G}},$$

and its RKHS $\mathcal{H}_\kappa$ is isometrically isomorphic to the closed span of the family $\{\Phi(z) \mid z \in \mathcal{Z}\}$ in $\mathcal{G}$. It follows immediately that the map

$$\Phi_\kappa : z \in \mathcal{Z} \mapsto \kappa(\cdot, z) \in \mathcal{H}_\kappa$$

also is a representation map of the *same* functional $\mathcal{F}$. Summarizing, it is always possible to pick $\mathcal{G}$ to be an RKHS in Assumption 4.5, and $\Phi$ the associated canonical feature map. We formalize this in the following result.

**Proposition A.1.** *Let $\mathcal{F}$ be a functional satisfying Assumption 4.5, and denote by $\Phi_0 : \mathcal{Z} \to \mathcal{G}_0$ a representation map of $\mathcal{F}$. Let*

$$\kappa : (z, z') \in \mathcal{Z} \mapsto \langle \Phi_0(z), \Phi_0(z') \rangle_{\mathcal{G}_0}.$$

*Then, $\kappa$ is a kernel, and the map*

$$\Phi_\kappa : z \in \mathcal{Z} \mapsto \kappa(\cdot, z) \in \mathcal{H}_\kappa \tag{13}$$

*also satisfies Assumption 4.5 for $\mathcal{F}$.*

**Consequences**   Given a functional $\mathcal{F}$ satisfying Assumption 4.5, let us take a kernel function $\kappa$ such that $\Phi_\kappa$ is a representation map of $\mathcal{F}$. The assumption guarantees that, for any $P, Q \in \mathcal{M}_1^+(\mathcal{Z})$, we have $\mathcal{F}(P) = \mathcal{F}(Q)$ if, and only if,

$$\int_{\mathcal{Z}} \kappa(\cdot, z) \mathrm{d}P(z) = \int_{\mathcal{Z}} \kappa(\cdot, z) \mathrm{d}Q(z), \tag{14}$$

where the Bochner integrals are assumed to exist. We can immediately recognize that (14) is an equality between the KMEs of the measures $P$ and $Q$ for the kernel $\kappa$. Introducing the KME map

$$\Pi_\kappa : M \in \mathcal{M}_1^+(\mathcal{Z}) \mapsto \int_{\mathcal{Z}} \kappa(\cdot, z) \mathrm{d}M(z),$$

the condition (14) simply rewrites as $\Pi_\kappa(P) = \Pi_\kappa(Q)$. This observation is essential in practice, as the question of what information about a distribution is captured by its KME is a well-studied area of research [38]. For instance, the inhomogeneous polynomial kernel of order $m \in \mathbb{N}_{>0}$ captures the first $m$ moments of a distribution; the exponential kernel captures the moment-generating function; and the Gaussian kernel captures the full distribution as it is characteristic [38, Sections 3.1.1 and 3.3.1]. In other words — and this is the concluding observation of this section:

*Observation* A.2 (Picking $\Phi$ in Assumption 4.5). When the map $\Phi$ is chosen as the partial evaluation of a kernel function $\kappa$ as in (13), the corresponding hypothesis (12) asserts the equality of the corresponding KMEs of the conditional distributions. Therefore, the properties being tested are directly the ones these KMEs capture, and they depend on the choice of the kernel function $\kappa$.

### A.3 The special case of KRR with a diagonal kernel: CMEs and their implementation

The challenge when following Observation A.2 and picking the representation map to be a kernel canonical feature map is that the data is now composed of functions; specifically, the data sets have the form $((X_n^{(i)}, \kappa(\cdot, Z_n^{(i)})))_n$, and the model output should be an element of $\mathcal{H}_\kappa$. This may be impossible to handle for some learning methods; e.g., feedforward neural networks, which need an output of fixed, finite dimension. Furthermore, even when it is possible to train and evaluate the algorithm with such data, computing the test statistic or threshold in (7) may still reveal intractable.

Fortunately, both of these concerns are void for KRR with a diagonal kernel $K = k \cdot \mathrm{id}_{\mathcal{H}_\kappa}$ (with $k$ a scalar kernel): it accepts functional data seamlessly, and the computations are tractable. As a matter of fact, KRR with a diagonal kernel and such data recovers exactly the popular framework of CMEs [39].

Before proceeding, we emphasize that we now have *two distinct kernel functions*: the kernel $K$, which is used to perform KRR, and the kernel $\kappa$, which is chosen depending on the functional we wish to test for in (12) accordingly to the developments of the previous section. The two are connected by the relation $K = k \cdot \mathrm{id}_{\mathcal{H}_\kappa}$, where $k$ is a freely-chosen scalar kernel.

**Computation of statistic and thresholds** We are interested in evaluating the statistic and threshold involved in (7). The computations to achieve this are commonly known [38, Section 4.1.2], and we report them here for completeness.

A first observation is that the quantities involved in the threshold — specifically, those defined in (8)–(11) — only involve the kernel $K$. This kernel indirectly depends on $\kappa$ since $K$ takes values in $\mathcal{L}_\mathrm{b}(\mathcal{H}_\kappa)$, but we assume that we are able to compute the quantities involved for $K$. In particular, under the UBD assumption, this assumption directly translates to the feasibility of these computations for $K_0$, which is trivially satisfied if $\tilde{\mathcal{G}}$ is finite-dimensional. This is the case in the standard case where $K = k \cdot \mathrm{id}_{\mathcal{H}_\kappa}$, since $\tilde{\mathcal{G}}$ is simply $\mathbb{R}$.

Next is the computation of the statistic, $\|f_{D_1,\lambda_1}(x) - f_{D_2,\lambda_2}(x)\|_{\mathcal{H}_\kappa}$, $x \in \mathcal{X}$. Importantly, for every $x \in \mathcal{X}$, ignoring the subscripts $_1$ and $_2$ indexing over the two processes at hand for readability,

$$f_{D,\lambda}(x) = K(x, X_{:N})(K(X_{:N}, X_{:N}) + \lambda \mathrm{id}_{\mathcal{H}_\kappa^N})^{-1} \begin{pmatrix} \kappa(\cdot, Z_1) \\ \vdots \\ \kappa(\cdot, Z_N) \end{pmatrix}.$$

We now leverage the fact that $K = k \cdot \mathrm{id}_{\mathcal{H}_\kappa}$. The identity operator on $\mathcal{H}_\kappa$ factors out and we obtain

$$f_{D,\lambda}(x) = \left( z \in \mathcal{Z} \mapsto k(x, X_{:N})(k(X_{:N}, X_{:N}) + \lambda \mathrm{id}_{\mathbb{R}^N})^{-1} \begin{pmatrix} \kappa(z, Z_1) \\ \vdots \\ \kappa(z, Z_N) \end{pmatrix} \right).$$

Introducing now the coefficients

$$\alpha_n^{(i)}(x) = (k(X_{:N}^{(i)}, X_{:N}^{(i)}) + \lambda_i \mathrm{id}_{\mathbb{R}^N})^{-1} k(X_{:N}^{(i)}, x),$$

for all $i \in \{1, 2\}$ and with $N = \text{len}(D_i)$, we obtain

$$\|f_{D_1, \lambda_1}(x) - f_{D_2, \lambda_2}(x)\|_{\mathcal{H}_\kappa}^2$$

$$= \sum_{n,m=1}^{\text{len}(D_1)} \alpha_n^{(1)}(x) \alpha_m^{(1)}(x) \kappa(Z_n^{(1)}, Z_m^{(1)}) + \sum_{n,m=1}^{\text{len}(D_2)} \alpha_n^{(2)}(x) \alpha_m^{(2)}(x) \kappa(Z_n^{(2)}, Z_m^{(2)})$$

$$- 2 \sum_{n=1}^{\text{len}(D_1)} \sum_{m=1}^{\text{len}(D_2)} \alpha_n^{(1)}(x) \alpha_m^{(2)}(x) \kappa(Z_n^{(1)}, Z_m^{(2)}).$$

Here, we used the identity $\kappa(z, z') = \langle \kappa(\cdot, z), \kappa(\cdot, z') \rangle_{\mathcal{H}_\kappa}$, for all $z, z' \in \mathcal{Z}$. Concluding, if the underlying learning algorithm is KRR with kernel $K = k \cdot \text{id}_{\mathcal{H}_\kappa}$, it is possible to evaluate the statistic and threshold of the test of Theorem 4.4. We provide in Appendix C algorithms that summarize these computations for implementation purposes.

### A.3.1 Numerical study

We illustrate the preceding ideas by showcasing how the choice of kernel $\kappa$ influences what is being tested, and the power of the test for a fixed amount of data. The metrics and vocabulary used in the setup are identical to those introduced in Section 5, which we recommend to read first.

We consider the Gaussian kernel $k$ on $\mathcal{X} = [-1, 1]^2$, pick a function in the RKHS of $k$, and generate measurements with two different additive noises: one is $\mathcal{N}(0, s^2)$, and the other is a mixture of $\mathcal{N}(-\mu, s^2)$ and $\mathcal{N}(\mu, s^2)$, for different values of $\mu$ but fixed $s^2$. This choice guarantees that the noise distributions have both zero means, but differ for higher moments. We embed the measurements in $\mathcal{G} = \mathcal{H}_\kappa$, where $\kappa$ is either the inhomogeneous linear or the Gaussian kernel. The corresponding conditional expectations in $\mathcal{G}$ differ for the Gaussian kernel (since it is characteristic [38]), but coincide for the linear kernel. As a result, a test with such data should trigger with $\kappa$ the Gaussian kernel, and not trigger with $\kappa$ the linear one. We run vector-valued KRR with kernel $K = k \cdot \text{id}_{\mathcal{H}_\kappa}$, bootstrap test thresholds, and run the test on the locations present in the data set. Figure 4 shows the resulting empirical positive rate for different $\alpha$-levels, confirming that the Gaussian kernel confidently sees a difference between the distributions, whereas the linear kernel only compares means and therefore does not reject more often than by chance.

However, a rich kernel like the Gaussian one can be less efficient at comparing lower moments of distributions. We show this by comparing empirical error rates of the test for inhomogeneous polynomial kernels of different degrees and Gaussian kernels of different bandwidths for $\kappa$. We use the same setup as before, but this time report type I and II errors instead of just the positive rate. For the type I error, we use the same function from $\mathcal{X}$ to $\mathcal{Z}$ and noise distribution to generate both data sets. For the type II error, we use different functions from $\mathcal{X}$ to $\mathcal{Z}$, but the same additive Gaussian noise distribution, so that the distributions on $\mathcal{Z}$ differ by their means but higher moments coincide. Figure 5 shows that rich kernels (like the Gaussian one with small bandwidth) — which see fine-grained distributional differences — are less efficient at recognizing differences in the mean than less rich kernels (e.g., linear kernel) tailored to that particular moment of the distribution.

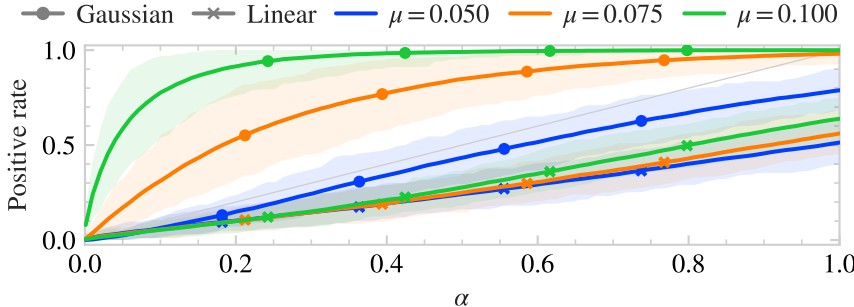

Figure 4: Comparing the Gaussian and inhomogeneous linear kernels on different distributions having the same conditional mean on the measurement set $\mathcal{Z}$. A trigger is a true positive for the Gaussian kernel, and a false positive for the linear one. The results are averaged over 100 runs of the experiment, with the shaded regions reporting the 2.5% and 97.5% quantiles.

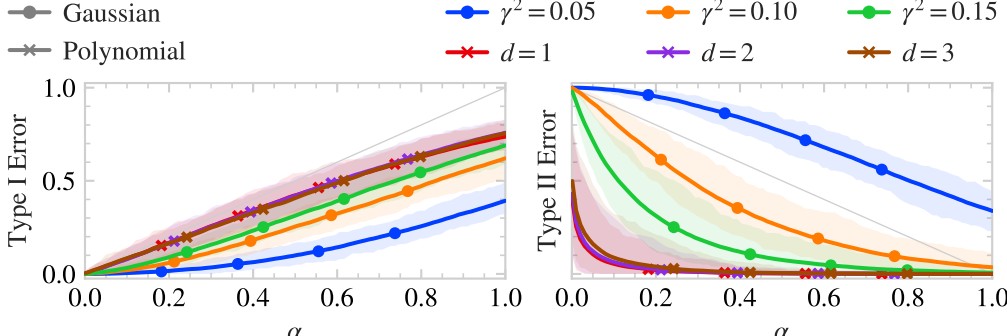

Figure 5: Empirical type I and II errors for different output kernels. For the type II error, conditional expectations in $\mathcal{Z}$ differ but higher moments coincide. We see that Gaussian kernels are more conservative and achieve higher type II error. The results are averaged over 100 runs of the experiment, with the shaded regions reporting the 2.5% and 97.5% quantiles.

# B    Bootstrapping

The general outcome of Theorem 4.4 is that, under the appropriate assumptions on both $\Theta_i$ and the sampling process or kernel, a suitable choice for $B_i(D, x)$ has the form $B_i(D, x) = \beta_i \cdot \sigma_{D,\lambda_i}(x)$, where $\beta_i > 0$. There are three main reasons, however, for which the practical use of the specific expression of $\beta_i$ may be limited. First, it significantly depends on parameters that are hard to infer, such as the RKHS norm upper bound $S_i$ and the sub-gaussianity constant or operator $R_i$ [25, 40]. Second, even when such parameters are known, evaluating the constant numerically may still be challenging due to the presence of (possibly) infinite-dimensional objects in the expression. Finally, these bounds may be very conservative, as they are essentially worst-case in $\Theta_i$. This last point is well-known in practice and is further illustrated by our numerical experiments below.

These considerations motivate bootstrapping values of $\beta_1$ and $\beta_2$ suitable for the problem at hand. Let $D_1$ and $D_2$ denote the observed data sets of transition pairs from $p_1$ and $p_2$, with respective sample sizes $n_1 = \text{len}(D_1)$ and $n_2 = \text{len}(D_2)$. We assume a sufficiently dense finite subset $\mathcal{M} = \{x_j \mid j \in [J]\} \subset \mathcal{S}$, obtained for instance by gridding or random sampling, on which we want the type I error calibration to hold. We neglect bootstrapping over time, meaning that the resulting values are only meaningful for $(\mathcal{S}, n_1, n_2)$-guarantees. We further fix $B_i = \beta_i \cdot \sigma_{D_i, \lambda_i}$ so that the test $\mathcal{T}$ is parameterized by the vector $\beta = (\beta_1, \beta_2) \in \mathbb{R}^2_{>0}$. Our goal is finding numerically a minimally conservative value for $\beta$ as measured by the empirical type II error on test scenarios while preserving a type I error of at most $\alpha$.

In the following, we detail two possible approaches that calibrate $\beta_1$ and $\beta_2$ independently of each other. The first one essentially bootstraps each value by applying the test on resamplings of the same data set. The second one bootstraps the confidence bounds of the underlying KRR. Both methods work by generating $M$ bootstrap replicates, obtaining statistics $(\bar{\beta}_{i,m})_{m \in [M]}$ for each $i \in \{1, 2\}$, and taking

$$\beta_1 = (1 - \alpha t)\text{-quantile of } \{\bar{\beta}_{1,m}\}, \qquad \beta_2 = (1 - \alpha(1 - t))\text{-quantile of } \{\bar{\beta}_{2,m}\},$$

with a user-specified $t \in (0, 1)$. When needed, grid-search over $t$ is inexpensive. We use $t = \frac{1}{2}$ in all simulations.

The following two schemes now differ only in how the families $(\bar{\beta}_{i,m})_{m \in [M]}$ are generated. Table 1 summarizes the difference in computational complexity.

**Naive resampling**    A straightforward approach is to resample the data directly. For each $i \in \{1, 2\}$, we draw with replacement $M$ pairs of data sets $((\Delta_m, \Delta'_m))_{m \in [M]}$ from $D_i$, where each data set in the pair has the same size as $D_i$. For each $m$, we then compute $\bar{\beta}_{i,m}$ as the minimal $\bar{\beta}_m \in \mathbb{R}_{>0}$ such that

$$\left\| f_{\Delta_m, \lambda_i}(x) - f_{\Delta'_m, \lambda_i}(x) \right\| \leq \bar{\beta}_m \left( \sigma_{\Delta_m, \lambda_i}(x) + \sigma_{\Delta'_m, \lambda_i}(x) \right)$$

for all $x \in \mathcal{M}$. See Algorithm 2 for more details.

**Wild bootstrap**    A more scalable alternative is the wild bootstrap, which replaces repeated resampling and re-fitting by random perturbations of the residuals of the nominal KRR estimate. Following Singh and Vijaykumar [27], we adopt anti-symmetric multipliers, which in the scalar-output case were shown to mitigate regularization bias and improve calibration accuracy. Such an analysis is unavailable in our more general setting, but we apply the same construction by straightforward extension to multi-dimensional outputs. However, we use the standard error term $\sigma_{D,\lambda}(x)$ defined in (9) for studentization to remain consistent with the theory-backed form of the confidence bounds identified in Theorem 4.3. For each $i \in \{1, 2\}$, draw independent mean-zero multipliers $(q^{(m)} \in \mathbb{R}^{n_i})_{m \in [M]}$ with covariance $I_{n_i} - 11^\top / n_i$ and collect the perturbed residual data set

$$\Delta_m = \left( \left( X_j, q_j^{(m)} \cdot (Y_j - f_{D_i, \lambda_i}(x_j)) \right) \right)_{(X_j, Y_j) \in D_i},$$

from which $\bar{\beta}_{i,m}$ is obtained as the minimal $\bar{\beta}_m \in \mathbb{R}_{>0}$ satisfying

$$\| f_{\Delta_m, \lambda_i}(x) \| \leq \bar{\beta}_m \, \sigma_{\Delta_m, \lambda_i}(x)$$

for all $x \in \mathcal{M}$. This method requires only a single KRR fit since the Gram matrix for any $\Delta_m$ is the same as for $D_i$. See Algorithm 3 for more details.

Table 1: Asymptotic time complexity for computing the test threshold with different calibration methods, in excess of fitting the models $f_{D_i,\lambda_i}$, for $i \in \{1,2\}$, and evaluating the test statistic $\|f_{D_1,\lambda_1}(x) - f_{D_2,\lambda_2}(x)\|$ over all $x \in \mathcal{S}$. Here $n_i$ denotes the data set size, $m = |\mathcal{M}|$ the grid size, and $M$ the number of bootstrap replicates.

| Method | Analytical bound | Naive bootstrap | Wild bootstrap |
|---|---|---|---|
| Runtime | $\mathcal{O}(1)$ | $\mathcal{O}(Mn_i^3 + Mmn_i^2)$ | $\mathcal{O}(Mmn_i^2)$ |

(a) Influence of data set size $n = n_1 = n_2$.

(b) Influence of regularization parameter $\lambda = \lambda_1 = \lambda_2$.

Figure 6: Type I and II errors for the test with different bootstrapping schemes, at varying data set sizes $n$ and regularization levels $\lambda$. Type I error becomes faithful to the level $\alpha$ when $n$ and $\lambda$ are sufficiently large. Power increases for larger $n$ and smaller $\lambda$. The results are averaged over 100 runs of each experiment. Confidence intervals are omitted for readability. "Singh et al." refers to the method proposed in [27].

**Validation**   We assess calibration and power for three bootstrapping schemes: (i) our naive resampling bootstrap; (ii) our wild bootstrap; and (iii) the wild bootstrap of Singh and Vijaykumar [27], which differs from our wild bootstrap only in the studentization term, using

$$\sigma_{D_i,\lambda_i} = \left\| \left( \left( k(X^{(i)}_{:n_i}, X^{(i)}_{:n_i}) + \lambda_i I_{n_i} \right)^{-1} k(X^{(i)}_{:n_i}, x) \right) \operatorname{diag}\left( \varepsilon_1, ..., \varepsilon_{n_i} \right) \right\|_{\mathcal{G}^{n_i}},$$

where $D_i = ((X^{(i)}_j, Y^{(i)}_j))^{n_i}_{j=1}$ is the data set and $\varepsilon_j = Y^{(i)}_j - f_{D_i,\lambda_i}(X^{(i)}_j)$ are the residuals as in our wild bootstrap.

We compare the performance of these methods in varying conditions on synthetic data sets generated in a way that either satisfies the null (for the type I error), or violates it (for the type II error). For that, we set $\mathcal{X} = [-1,1]^2$, take $k$ as the Gaussian kernel with bandwidth $\gamma^2 = 0.25$, and generate data as measurements of functions in $\mathcal{H}_k$ with additive Gaussian noise. For the type I error, we use equal mean functions with Gaussian noise, and for the type II error use different mean functions with the same noise distribution. For each configuration, we bootstrap the confidence thresholds and run the test.

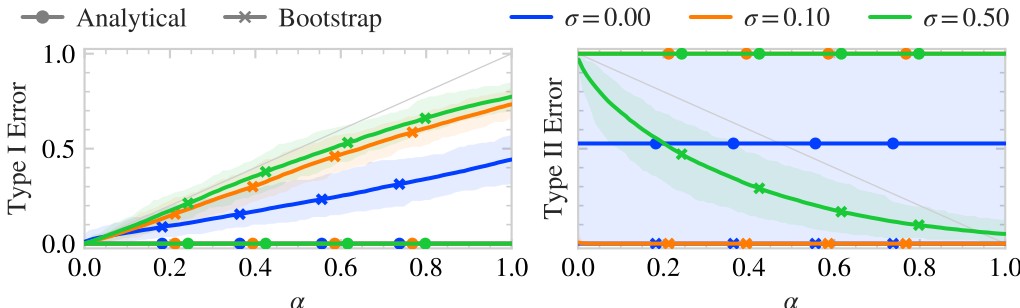

Figure 7: Type I and II errors for the test with the bootstrapped and analytical thresholds. Analytical thresholds yield a very conservative test (level is less than $\alpha$, but the type I and II errors are independent of $\alpha$. In contrast, bootstrapping preserves the guarantee and increases power. The results are averaged over 100 runs of the experiment, with the shaded regions reporting the 2.5% and 97.5% quantiles.

**Varying data set size**   Figure 6a reports results for data set sizes $n = n_1 = n_2 \in \{20, 50, 100\}$. Type I error increases with $\alpha$ and improves with $n$. They all uphold the level, except for the naive method with $n = 20$. For fixed regularization $\lambda = 0.5$, all methods become more conservative as $n$ grows. Accordingly, type II error improves for increasing $\alpha$ and $n$.

**Varying regularization**   In Figure 6b, we now vary the regularization $\lambda = \lambda_1 = \lambda_2 \in \{10^{-3}, 10^{-1}, 1\}$. At a fixed data set size $n = 500$, larger $\lambda$ yields more conservative calibration, while smaller $\lambda$ result in overly aggressive bootstraps and the type I error exceeding the prescribed level $\alpha$.

Put together, these results indicate that sufficient data and a suitable choice of $\lambda$ are critical for good calibration. Across both experiments, naive resampling is more aggressive (higher type I and lower type II error), our wild bootstrap is more conservative (lower type I and higher type II error), and the variant by Singh and Vijaykumar [27] lies in between.

**Power against analytical calibration**   To confirm the increase in power gained by bootstrapping relative to analytical thresholds, we now fix $n$ and $\lambda$, and vary the noise variance $\sigma$. Because the outputs of functions in $\mathcal{H}_k$ are one-dimensional, the sub-Gaussian noise operator reduces to a scalar, which we set to the ground-truth variance. For the bootstrapping scheme, we only compare against our naive resampling method. Figure 7 shows that the analytical thresholds are markedly conservative: type I error remains well below $\alpha$ and both errors are essentially independent of $\alpha$. In contrast, the type I error of the bootstrap tracks the level $\alpha$ more faithfully and achieves significantly lower type II error. Overall, this indicates that bootstrap calibration yields strictly higher power at fixed $\alpha$ without sacrificing control of the type I error.

For more details on the experiments in this section, see Appendix G.

## C   Summary of algorithms

Algorithm 1 summarizes our implementation of the test of Theorem 4.4. The bootstrapping of the multiplicative constants $\beta_i$, $i \in \{1, 2\}$ it relies on is detailed in Appendix B, and summarized in Algorithms Algorithms 2 and 3. We emphasize that, in principle, any other bootstrapping algorithm for KRR can be used. All routines assume the form $K = k \cdot \mathrm{id}_\kappa$ (c.f Appendix A). Algorithm 4 shows how to compute the test statistic, called the conditional maximum mean discrepancy (CMMD).

---

**Algorithm 1** Kernel conditional two-sample test

---

**Require:** data sets $D_1, D_2$; tested covariate region $\mathcal{S} \subseteq \mathcal{X}$; bootstrap covariate region $\mathcal{M} \subseteq \mathcal{X}$
**Ensure:** Covariate rejection subregion of $\mathcal{S}$, with type I error at most $\alpha$
  1: $\tilde{\beta}_1 \leftarrow \text{Bootstrap}(D_1, \mathcal{M})$            ▷ use either Algorithm 2 or 3
  2: $\tilde{\beta}_2 \leftarrow \text{Bootstrap}(D_2, \mathcal{M})$
  3: **return** $\{x \in \mathcal{S} \mid \text{CMMD}(D_1, D_2, k, \kappa, \lambda, x) > \tilde{\beta}_1 \cdot \sigma_{D_1,\lambda}(x) + \tilde{\beta}_2 \cdot \sigma_{D_2,\lambda}(x)\}$

---

**Algorithm 2** Naive bootstrap

---

**Require:** data set $D$; grid $\mathcal{M}$
**Ensure:** Chooses $\tilde{\beta}$ s.t. test does not reject on $\mathcal{M}$ for an $(1 - \alpha)$-fraction of bootstrapped data sets
  1: **for** $j \leftarrow 1$ **to** $M$ **do**
  2:      $D_1, D_2 \sim D$           ▷ resample $D_1, D_2$ from $D$ uniformly with replacement
  3:      $\mathcal{B}_j \leftarrow \max_{x \in \mathcal{M}} \left( \frac{\text{CMMD}(D_1, D_2, k, \kappa, \lambda, x)}{\sigma_{D_1,\lambda}(s) + \sigma_{D_2,\lambda}(s)} \right)$      ▷ smallest $\tilde{\beta}$ s.t. test does not reject on $D_1, D_2$
  4: **return** $(1 - \alpha)$-quantile of $\{\mathcal{B}_j \mid j \in [M]\}$

---

**Algorithm 3** Wild bootstrap

---

**Require:** data set $D$; grid $\mathcal{M}$
**Ensure:** Chooses $\tilde{\beta}$ s.t. test does not reject on $\mathcal{M}$ for an $(1 - \alpha)$-fraction of bootstrapped data sets
  1: $\mathbf{K} \leftarrow [k(x, x')]_{(x,\cdot) \in D_i}^{(x',\cdot) \in D_i}$
  2: $\mathbf{L} \leftarrow [\kappa(z, z')]_{(\cdot,z) \in D_i}^{(\cdot,z') \in D_i}$
  3: $\mathbf{k}_x \leftarrow [k(x, x')]_{(x',\cdot) \in D_i}^{\top}$           $\forall x \in \mathcal{M}$
  4: $\mathbf{A} \leftarrow (\mathbf{K} + \lambda I_{|D|})^{-1} \mathbf{K}$
  5: $\mathbf{v}_x \leftarrow (\mathbf{K} + \lambda I_{|D|})^{-1} \mathbf{k}_x$           $\forall x \in \mathcal{M}$
  6: **for** $j \leftarrow 1$ **to** $M$ **do**
  7:      $\mathbf{q} \sim \mathcal{N}(\mathbf{0}, I_{|D|} - \frac{1}{|D|} \mathbf{1}\mathbf{1}^{\top})$           ▷ wild multipliers
  8:      $\xi_x \leftarrow \mathbf{q} \odot v_x$           $\forall x \in \mathcal{M}$      ▷ $\odot$ denotes Hadamard product
  9:      $\mathcal{B}_j \leftarrow \max_{x \in \mathcal{M}} \left( \frac{1}{\sigma_{D,\lambda}(x)} \left( \xi^{\top} (I_{|D|} - \mathbf{A})^{\top} \mathbf{L} (I_{|D|} - \mathbf{A}) \xi \right)^{1/2} \right)$
  10: **return** $(1 - \alpha)$-quantile of $\{\mathcal{B}_j \mid j \in [M]\}$

---

**Algorithm 4** CMMD: Conditional maximum mean discrepancy

---

**Require:** covariate $x \in \mathcal{X}$; data sets $D_1; D_2$
**Ensure:** Computes $\|f_{D_1,\lambda}(x) - f_{D_2,\lambda}(x)\|_{\mathcal{H}_\kappa}$
  1: $\mathbf{K}_i \leftarrow [k(x, x')]_{(x,\cdot) \in D_i}^{(x',\cdot) \in D_i}$           $\forall \quad i \in \{1, 2\}$
  2: $\mathbf{k}_i \leftarrow [k(x, x')]_{(x',\cdot) \in D_i}^{\top}$           $\forall \quad i \in \{1, 2\}$
  3: $\mathbf{a}_i \leftarrow (\mathbf{K}_i + \lambda I_{|D_i|})^{-1} \mathbf{k}_i$           $\forall \quad i \in \{1, 2\}$
  4: $\mathbf{C}_{ij} \leftarrow [\kappa(z, z')]_{(\cdot,z) \in D_i}^{(\cdot,z') \in D_j}$           $\forall i, j \in \{1, 2\}$
  5: **return** $(\mathbf{a}_1^{\top} \mathbf{C}_{11} \mathbf{a}_1 - 2 \cdot \mathbf{a}_1^{\top} \mathbf{C}_{12} \mathbf{a}_2 + \mathbf{a}_2^{\top} \mathbf{C}_{22} \mathbf{a}_2)^{1/2}$

---

# D  Connections between Theorem 4.3 and existing bounds

Case 1 in Theorem 4.3 generalizes Corollary 3.4 in Abbasi-Yadkori [12] to outputs in separable Hilbert spaces without introducing conservatism, as it recovers exactly the same bound as in this reference under the assumptions there. It also generalizes Theorem 1 in Chowdhury and Gopalan [20] by removing the regularity assumptions connecting the input set $\mathcal{X}$ and the kernel $K$ (in the reference, $\mathcal{X}$ is compact and $K$ is bounded). It further generalizes this same result in yet another way by removing the assumption that $K$ should be trace-class, replacing it by the (significantly milder) assumption that $K_0$ should be so and hereby allowing the special case of a diagonal kernel $K = k \cdot \mathrm{id}_{\mathcal{G}}$. To the best of our knowledge, this is the first time-uniform uncertainty bound for KRR allowing this essential case.

The assumption that the kernel $K$ is UBD in Case 1 above is the central methodological contribution of the proof. It enables us to generalize the proof method of Abbasi-Yadkori [12] by leveraging a *tensor factorization* of the RKHS $\mathcal{H}_K$ (Proposition E.12). Specifically, Proposition E.12 allows avoiding the usual problems encountered with non-trace-class kernels — more precisely, that trace-dependent quantities explode — by replacing them with the corresponding quantities for the elementary block $K_0$. Intuitively, all of the "estimation machinery" is performed in the RKHS of $K_0$, which is a trace-class kernel and for which classical methods work well, and we simply need to ensure that the noise is "compatible" with this structure. Here, this compatibility takes the form of the operator $R_{\mathcal{V}}$ being trace-class. Interestingly, a special case of Proposition E.12 has been used in Li et al. [41] (Theorem 1) to identify $\mathcal{H}_K$ with a space of Hilbert-Schmidt operators in the case $K = k \cdot \mathrm{id}_{\mathcal{G}}$, hereby avoiding problems with the trace of $K$ by working with that of $k$, similarly to us. While Li et al. [41] is focused on learning rates for conditional mean embeddings, the subsequent work [42] investigates (in some settings minimax optimal) learning rates for vector-valued KRR, utilizing essentially the same strategy of identifying KRR using a kernel of the form $K = k \cdot \mathrm{id}_{\mathcal{G}}$ with regularized estimation of Hilbert-Schmidt operators. Conceptually, both [41, 42] and our work use a tensor space construction to avoid problems with quantities related to trace-class operators, however, the technical details significantly differ. First, [41, 42] uses an identification with Hilbert-Schmidt operators, whereas we work with more general tensor product structures, that allow us formulate our results using UBD kernels. Second, [41, 42] aim at learning rates in a typical supervised learning setup, in particular, with random covariates, whereas we are interested in uniform (in the covariates) confidence bounds, and hence the proof techniques are rather different. Finally, the regularized least-squares algorithm analyzed by [41, 42] is not affected by whether the used kernel is trace-class or not, only the analysis needs to take this into account. In contrast, we need the tensor structure as formalized by our UBD assumption to even have a well-defined expression for the confidence bound due to appearance of the Fredholm determinant. We would like to remark that it appears that many arguments of Li et al. [41, 42] also hold under the more general assumption that $K$ is UBD. An interesting avenue for future work is thus to investigate to what extent this generalization carries over, and whether the derived learning rates, e.g., [41, Theorem 2], hold under UBD. Summarizing, we believe the UBD assumption and the resulting structure of the RKHS identified in Proposition E.12 is a relevant new idea to handle a large class of non-trace-class kernels and can be used in other areas of kernel methods such as learning theory with loss functions compatible with the structure.

We emphasize that the subgaussianity assumption of Case 1 is in general a strengthening of the assumption that $p$ is $\bar{\rho}$-subgaussian for $\bar{\rho} \in \mathbb{R}_{>0}$, but also a weakening of the one that $p$ is $R'$-subgaussian for $R' \in \mathcal{L}_{\mathrm{b}}(\mathcal{G})$ self-adjoint, positive semi-definite, and of trace class. This last case was first introduced in Mollenhauer and Schillings [22], to which we refer for an interpretation. The structure we require on $R$ clearly identifies how much control we need on each of the components of the noise: those that are handled by the RKHS of the trace-class kernel $K_0$ may be simply $\rho$-subgaussian, while the those that are not must have a trace-class variance operator. In fact, if the kernel $K$ is itself of trace class, we may take $K_0 = K$ in the theorem and the assumption simplifies to $p$ being $\rho$-subgaussian.

Finally, Case 2 does not require *any* structural assumptions on $K$ (such as trace-class or UBD), but assumes *independent* transition pairs (excluding online sampling) and yields a time-local bound (since $\mathcal{N} = \{N\}$). It is thus of independent interest to Case 1, as they apply in different settings. It is also worth noting that the proof of Case 2 is considerably simpler than that of Case 1, as it follows directly from results on concentration of quadratic forms of subgaussian vectors with independent coefficients [22].

# E  Generalized confidence bounds in vector-valued least squares

We provide in this section the proof of Theorem 4.3. Specifically, we show a more general result which does not assume that the space $\mathcal{H}_K$ in which the estimator lies is an RKHS; only that it is a generic Hilbert space $\mathcal{H}$. The result we show is given later in this section in Theorem E.23 and pertains to confidence bounds in vector-valued least squares estimation under general assumptions on the sampling operators; KRR is then obtained for a specific choice of the measurement operators and Hilbert space where estimation is performed.

This section is organized as follows. We first provide in Section E.1 the general setup of vector-valued least squares estimation, and identify that the main challenge for confidence bounds is bounding a specific term involving the noise. We then provide in Sections E.2 and E.3 methods to bound this term under different assumptions that we formalize. The main intermediate results of those sections are Lemma E.4, Theorem E.14, and Corollary E.20, and are essential in the proof of Theorem E.23, which we only state and prove in Section E.4 as we provide a gentle introduction to its assumptions first. We finally leverage it in Section E.5 to show Theorem 4.3.

## E.1  Setup and goal

Let $\mathcal{H}$ and $\mathcal{G}$ be separable Hilbert spaces, and $D = [(L_n, y_n)]_{n \in \mathbb{N}_{>0}}$ be an $\mathcal{L}_{\mathrm{b}}(\mathcal{H}, \mathcal{G}) \times \mathcal{G}$-valued stochastic process. The operators $L_n$ are called the (random) *evaluation operators* and the vectors $y_n$ the evaluations (or measurements), $n \in \mathbb{N}_{>0}$. We assume that there exists $h^\star \in \mathcal{H}$ and a centered $\mathcal{G}$-valued process $\eta = (\eta_n)_{n \in \mathbb{N}_{>0}}$ such that $y_n = L_n h^\star + \eta_n$. Let now $\lambda > 0$ and define the regularized least-squares problem with data $D$ at time $N \in \mathbb{N}_{>0}$ as

$$\arg\min_{h \in \mathcal{H}} \sum_{n=1}^{N} \|y_n - L_n h\|_{\mathcal{G}}^2 + \lambda \|h\|_{\mathcal{H}}^2. \tag{15}$$

This optimization problem has a well-known solution, which we recall below together with the proof for completeness.

**Lemma E.1.** *For all $N \in \mathbb{N}_{>0}$, the optimization problem (15) has a unique solution $h_{N,\lambda}$, and*

$$h_{N,\lambda} = (M_N^\star M_N + \lambda \mathrm{id}_{\mathcal{H}})^{-1} M_N^\star y_{:N}, \tag{16}$$

*where we introduced the operator $M_N$ and its adjoint as*

$$M_N : h \in \mathcal{H} \mapsto (L_n h)_{n \in [N]} \in \mathcal{G}^N, \quad \text{and} \quad M_N^\star : (g_1, \ldots, g_N) \in \mathcal{G}^N \mapsto \sum_{n=1}^{N} L_n^\star g_n. \tag{17}$$

*Proof.* Rewriting the objective function and completing the square leads to

$$\sum_{n=1}^{N} \|y_n - L_n h\|_{\mathcal{G}}^2 + \lambda \|h\|_{\mathcal{H}}^2 = \sum_{n=1}^{N} \langle y_n - L_n h, y_n - L_n h \rangle_{\mathcal{G}} + \lambda \langle h, h \rangle_{\mathcal{H}}$$

$$= \sum_{n=1}^{N} \|y_n\|_{\mathcal{G}}^2 - 2 \sum_{n=1}^{N} \langle y_n, L_n h \rangle_{\mathcal{G}} + \sum_{n=1}^{N} \langle L_n h, L_n h \rangle_{\mathcal{G}} + \lambda \langle h, h \rangle_{\mathcal{H}}$$

$$= \left\langle \left( \sum_{n=1}^{N} L_n^* L_n + \mathrm{id}_{\mathcal{H}} \right) h, h \right\rangle_{\mathcal{H}} - 2 \left\langle \sum_{n=1}^{N} L_n^* y_n, h \right\rangle_{\mathcal{H}} + \sum_{n=1}^{N} \|y_n\|_{\mathcal{G}}^2$$

$$= \|h\|_{\sum_{n=1}^{N} L_n^* L_n + \lambda \mathrm{id}_{\mathcal{H}}}^2 - 2 \left\langle \left( \sum_{n=1}^{N} L_n^* L_n + \lambda \mathrm{id}_{\mathcal{H}} \right)^{-1} \sum_{n=1}^{N} L_n^* y_n, h \right\rangle_{\sum_{n=1}^{N} L_n^* L_n + \lambda \mathrm{id}_{\mathcal{H}}}$$

$$+ \left\| \left( \sum_{n=1}^{N} L_n^* L_n + \lambda \mathrm{id}_{\mathcal{H}} \right)^{-1} \sum_{n=1}^{N} L_n^* y_n \right\|_{\sum_{n=1}^{N} L_n^* L_n + \mathrm{id}_{\mathcal{H}}}^2$$

$$- \left\| \left( \sum_{n=1}^{N} L_n^* L_n + \lambda \mathrm{id}_{\mathcal{H}} \right)^{-1} \sum_{n=1}^{N} L_n^* y_n \right\|_{\sum_{n=1}^{N} L_n^* L_n + \lambda \mathrm{id}_{\mathcal{H}}}^2$$

$$+ \sum_{n=1}^{N} \|y_n\|_{\mathcal{G}}^2$$

$$= \left\| h - \left( \sum_{n=1}^{N} L_n^* L_n + \lambda \mathrm{id}_{\mathcal{H}} \right)^{-1} \sum_{n=1}^{N} L_n^* y_n \right\|_{\sum_{n=1}^{N} L_n^* L_n + \lambda \mathrm{id}_{\mathcal{H}}}^2 + \text{Terms without } h$$

Since $\sum_{n=1}^{N} L_n^* L_n + \lambda \mathrm{id}_{\mathcal{H}} = M_N^\star M_N + \lambda \mathrm{id}_{\mathcal{H}}$ is positive definite, and $\sum_{n=1}^{N} L_n^* y_n = M_N^\star y_{:N}$, this shows that the unique solution to the optimization problem (15) is indeed given by (16). □

Our general goal is to obtain a finite-sample confidence region for the estimator $h_{N,\lambda}$ of $h^\star$. While we only state the precise result in Section E.4, we want — in short — a high-probability bound on the quantity $\|\bar{L} h_{N,\lambda} - \bar{L} h^\star\|_{\bar{\mathcal{G}}}$. Here $\bar{\mathcal{G}}$ is a Hilbert space and $\bar{L} \in \mathcal{L}_\mathrm{b}(\mathcal{H}, \bar{\mathcal{G}})$ is a bounded operator capturing an appropriate property we are interested in; they are degrees of freedom of the problem. We begin with the following observation. This result, which is well-known in the case of KRR (see for example Abbasi-Yadkori [12, Chapter 3] or Chowdhury and Gopalan [19]), results from elementary algebraic manipulations, the triangle inequality, and properties of the operator norm.

**Lemma E.2.** *For any $N \in \mathbb{N}_{>0}$ and $\lambda \in \mathbb{R}_{>0}$, it holds that*

$$h_{N,\lambda} = h^\star - \lambda (M_N^\star M_N + \lambda \mathrm{id}_{\mathcal{H}})^{-1} h^\star + (M_N^\star M_N + \lambda \mathrm{id}_{\mathcal{H}})^{-1} M_N^\star \eta_{:N}. \tag{18}$$

*Consequently, for any Hilbert space $\bar{\mathcal{G}}$, bounded operator $\bar{L} \in \mathcal{L}_\mathrm{b}(\mathcal{H}, \bar{\mathcal{G}})$, and $N \in \mathbb{N}_{>0}$, the following holds:*

$$\|\bar{L} h_{N,\lambda} - \bar{L} h^\star\|_{\bar{\mathcal{G}}} \le \|\bar{L}(V_N + \lambda \mathrm{id}_{\mathcal{H}})^{-1/2}\|_{\mathcal{L}_\mathrm{b}(\mathcal{H}, \bar{\mathcal{G}})} \left( \|S_N\|_{(V_N + \lambda \mathrm{id}_{\mathcal{H}})^{-1}} + \lambda \|h^\star\|_{(V_N + \lambda \mathrm{id}_{\mathcal{H}})^{-1}} \right), \tag{19}$$

*where we introduced*

$$V_N = M_N^\star M_N, \quad and \quad S_N = M_N^\star \eta_{:N}.$$

The general strategy to bound $\|\bar{L} h_{N,\lambda} - \bar{L} h^\star\|_{\bar{\mathcal{G}}}$ is to bound the two terms involved in the right-hand side (RHS) of (19) independently. The term involving $h^\star$ is bounded as follows:

$$\|h^\star\|_{(M_N^\star M_N + \lambda \mathrm{id}_{\mathcal{H}})^{-1}} \le \lambda^{-1/2} \|h^\star\|_{\mathcal{H}}, \tag{20}$$

and we will assume a known upper-bound on $\|h^\star\|_{\mathcal{H}}$. Indeed, $\lambda \mathrm{id}_{\mathcal{H}} \preceq M_N^\star M_N + \lambda \mathrm{id}_{\mathcal{H}}$ since $M_N^\star M_N$ is positive semi-definite. Similarly, the operator norm rewrites as

$$
\begin{aligned}
&\|\bar{L}(M_N^\star M_N + \lambda \mathrm{id}_{\mathcal{H}})^{-1/2}\|_{\mathcal{L}_{\mathrm{b}}(\mathcal{H},\bar{\mathcal{G}})} \\
&= \sqrt{\left\| \bar{L}(M_N^\star M_N + \lambda \mathrm{id}_{\mathcal{H}})^{-1/2} \left( \bar{L}(M_N^\star M_N + \lambda \mathrm{id}_{\mathcal{H}})^{-1/2} \right)^\star \right\|_{\mathcal{L}_{\mathrm{b}}(\mathcal{H})}} \\
&= \sqrt{\frac{1}{\lambda} \left\| \bar{L} \left( \mathrm{id}_{\mathcal{H}} - (M_N^\star M_N + \lambda \mathrm{id}_{\mathcal{H}})^{-1} M_N^\star M_N \right) \bar{L}^\star \right\|_{\mathcal{L}_{\mathrm{b}}(\mathcal{H})}}.
\end{aligned}
\tag{21}
$$

Introducing the shorthands

$$
\Sigma_{D_{:N},\lambda}(\bar{L}) = \bar{L} \left( \mathrm{id}_{\mathcal{H}} - (M_N^\star M_N + \lambda \mathrm{id}_{\mathcal{H}})^{-1} M_N^\star M_N \right) \bar{L}^\star, \quad \text{and}
$$

$$
\sigma_{D_{:N},\lambda}(\bar{L}) = \sqrt{\|\Sigma_{D_{:n},\lambda}(\bar{L})\|_{\mathcal{L}_{\mathrm{b}}(\mathcal{H})}},
$$

we have

$$
\|\bar{L}(M_N^\star M_N + \lambda \mathrm{id}_{\mathcal{H}})^{-1/2}\|_{\mathcal{L}_{\mathrm{b}}(\mathcal{H},\bar{\mathcal{G}})} = \frac{1}{\sqrt{\lambda}} \sigma_{D_{:N},\lambda}(\bar{L}).
\tag{22}
$$

We are thus left with bounding the noise-dependent term $\|S_N\|_{(M_N^\star M_N + \lambda \mathrm{id}_{\mathcal{H}})^{-1}}$. Importantly, since this term is independent of $\bar{L}$, such a method naturally yields a high-probability bound of $\|\bar{L}h_{N,\lambda} - \bar{L}h^\star\|_{\bar{\mathcal{G}}}$ where the high-probability set where the bound holds is independent of $\bar{L}$.

### E.2 Noise bound for deterministic sampling

The first assumption under which we are able to provide a bound on $\|S_N\|_{(M_N^\star M_N + \lambda \mathrm{id}_{\mathcal{H}})^{-1}}$ is that of *deterministic sampling*, meaning that the sampling operators are not random and the noise is independent.

**Assumption E.3** (Deterministic sampling). *The process $L = (L_n)_{n \in \mathbb{N}_{>0}}$ is not random, and $\eta$ is an independent process.*

The bound is based on Corollary 2.6 in Mollenhauer and Schillings [22], which pertains to the concentration of quadratic forms of $R$-subgaussian random variables. We recall it here with our notations for completeness.

**Lemma E.4.** *Assume that $\eta$ is an independent process and that $\eta_n$ is $R$-subgaussian for all $n \in \mathbb{N}_{>0}$, with $R \in \mathcal{L}_{\mathrm{b}}(\mathcal{G})$, positive semi-definite and trace-class. Let $A \in \mathcal{L}_{\mathrm{b}}(\mathcal{G}^N, \mathcal{H})$. Then, $\eta_{:N}$ is $R \otimes \mathrm{id}_{\mathbb{R}^N}$-subgaussian, and for all $\delta \in (0,1)$, it holds with probability at least $1 - \delta$ that*

$$
\|A\eta_{:N}\|^2 \leq \mathrm{Tr}(T) + 2\sqrt{\ln \frac{1}{\delta}}\|T\|_2 + 2 \ln \frac{1}{\delta} \|T\|_{\mathcal{L}_{\mathrm{b}}(\mathcal{H})},
\tag{23}
$$

*where $T = A \cdot (R \otimes \mathrm{id}_{\mathbb{R}^N}) \cdot A^\star \in \mathcal{L}_{\mathrm{b}}(\mathcal{H})$ is of trace class and $\|B\|_2 = \sqrt{\mathrm{Tr}(TT^\star)}$ is its Hilbert-Schmidt norm.*

### E.3 Noise bound for online sampling with coefficients-transforming sampling operators

The previous case makes the central assumption that $(L_n)$ is a deterministic process. While it is possible to extend it to the case where $(L_n)_n$ is a uniformly bounded random process independent of $\eta$ (leveraging Lemma 4.1 in Mollenhauer and Schillings [22]), this still excludes *online sampling*, formalized as follows:

**Assumption E.5** (Online sampling). *We have access to a filtration $\mathcal{F} = (\mathcal{F}_N)_{N \in \mathbb{N}_{>0}}$ for which $L$ is predictable and $\eta$ is adapted.*

Under this assumption, the choice of $L_{N+1}$ is random and depends on $L_{:N}$ and on $y_{:N}$. The main difficulty then resides in the fact that $S_N$ and $(M_N^\star M_N + \lambda \mathrm{id}_{\mathcal{H}})^{-1}$ are now correlated random variables.

We address this correlation by showing a *self-normalized* concentration inequality for

$$
\|S_N\|_{(M_N^\star M_N + \lambda \mathrm{id}_{\mathcal{H}})^{-1}}.
$$

The inequality is also time-uniform, meaning that the high-probability set on which the bound holds does not depend on the time $N \in \mathbb{N}_{>0}$. It is a generalization of Theorem 3.4 (and Corollary 3.6) in Abbasi-Yadkori [12] from $\mathcal{G} = \mathbb{R}$ to a general separable Hilbert space $\mathcal{G}$. These results are based on the method of mixtures [43]. While we mostly follow the proof strategy from Abbasi-Yadkori [12], we introduce a general structural assumption on the sampling operators (Definition E.11 and Proposition E.12) to handle the case where the sampling operators are not trace-class (in the RKHS case, the case where the kernel is not trace-class). Furthermore, in contrast to Abbasi-Yadkori [12] we do not employ an explicit stopping-time construction to obtain time uniformity, but rather use Ville's inequality as in Whitehouse et al. [21].

### E.3.1 Challenge and goal

For the sake of discussion, let us assume that $\mathcal{H} = \mathcal{H}_K$, where $K$ is an operator-valued kernel. While extending Corollary 3.6 of Abbasi-Yadkori [12] to the case where $K$ is trace-class is relatively straightforward by following the proof technique, the method breaks if $K$ is not trace-class. In fact, it is not even clear *what* the corresponding bound on $\|S_N\|_{(\lambda \mathrm{id}_{\mathcal{H}} + V_N)^{-1}}$ is in that case, as using the expression given in Corollary 3.5 in the reference and replacing the scalar kernel with the operator-valued one yields a bound involving the quantity

$$\det \left[ \mathrm{id}_{\mathcal{G}^N} + \frac{1}{\lambda} M_N M_N^\star \right], \tag{24}$$

which is meaningless since $M_N M_N^\star$ is not of trace class. However, in the special case where $K$ is UBD with trace-class block decomposition $K_0$, we have a meaningful candidate to replace the determinant above: that involving $K_0$. Our goal then becomes upper bounding $\|S_N\|_{(V_N + \lambda \mathrm{id}_{\mathcal{H}})^{-1}}$ by a quantity involving $\det(\mathrm{id}_{\tilde{\mathcal{G}}^N} + \lambda^{-1} \tilde{M}_N \tilde{M}_N^\star)$, where $\tilde{M}_N \in \mathcal{L}_\mathrm{b}(\mathcal{H}_{K_0}, \tilde{\mathcal{G}}^N)$ is defined as $\tilde{M}_N \tilde{h} = (\tilde{h}(x_n))_{n \in [N]} \in \tilde{\mathcal{G}}^N$ for all $\tilde{h} \in \mathcal{H}_{K_0}$. Finally, while the discussion above assumes $\mathcal{H} = \mathcal{H}_K$ and the regularizer $V = \lambda \mathrm{id}_{\mathcal{H}_K}$, we aim at providing a noise bound for general spaces and regularizers.

**Proof summary**  Our proof is based on the method of mixtures of Abbasi-Yadkori [12]. A close look at the proof of Theorem 3.4 in the reference reveals that the problematic determinant (24) arises when integrating the function $G_N$ (in the notations of the reference) against a Gaussian measure on $\mathcal{H}_K$. We circumvent this integration by leveraging a *factorization of $\mathcal{H}_K$ as a tensor space*. While such a factorization $\mathcal{H}_K \cong H_1 \otimes H_2$ trivially exists for choices of separable $H_1$ and $H_2$ based on dimensional arguments (i.e., $\dim(\mathcal{H}_K) = \dim(H_1) \cdot \dim(H_2)$), we choose one in which all of the sampling operators $L_n$, $n \in \mathbb{N}_{>0}$, admit themselves the simple form $L_n \cong \tilde{L}_n \otimes \mathrm{id}_{H_2}$, and for which $\tilde{L}_n$ has nice properties (specifically, $\tilde{L}_n^\star \tilde{L}_n$ is trace-class). In the case of a diagonal kernel — that is, $K = k \cdot \mathrm{id}_{\mathcal{G}}$ and $L_n = K(\cdot, x_n)$, this construction is known and natural: the RKHS $\mathcal{H}_K$ is the (closure of the) span of vectors of the form $f \cdot g$, where $f \in \mathcal{H}_k$ and $g \in \mathcal{G}$; see for instance Paulsen and Raghupathi [35, Exercise 6.3]. Then, for any $h = \sum_{m=1}^{\infty} f_m \cdot g_m \in \mathcal{H}_{k \cdot \mathrm{id}_{\mathcal{G}}}$, introducing $h_\otimes = \sum_{m=1}^{\infty} f_m \otimes g_m \in \mathcal{H}_{K_0} \otimes \mathcal{G}$,

$$L_n h = \sum_{m=1}^{\infty} f_m(x_n) \cdot g_m \cong \sum_{m=1}^{\infty} (\tilde{L}_n \otimes \mathrm{id}_{\mathcal{G}})(f_m \otimes g_m) = (\tilde{L}_n \otimes \mathrm{id}_{\mathcal{G}}) h_\otimes,$$

In the more general case where $K$ is only UBD instead of diagonal, we prove that such a factorization also exists (Proposition E.12). In any case, this factorization allows us to write $\mathcal{H}_K \cong \mathcal{H}_{K_0} \otimes \mathcal{V}$. This enables evaluating (a modification of) the function $G_N$ on pure tensors $\tilde{h} \otimes v$, where $\tilde{h} \in \mathcal{H}_{K_0}$ and $v \in \mathcal{V}$. We then leverage a *double mixture* argument: one for the variable $\tilde{h}$, and one for the variable $v$. The one on $\tilde{h}$ is the same as in Abbasi-Yadkori [12]: a Gaussian measure on $\mathcal{H}_{K_0}$, which outputs the desired determinant involving $K_0$. The one on $v$ is not Gaussian and enables recovering the norm we wish to bound, $\|S_N\|_{(V_N + \lambda \mathrm{id}_{\mathcal{H}_K})^{-1}}$ (up to some regularization term), when combined with Jensen's inequality. Our double mixture strategy is fundamentally different from what is used for Chowdhury and Gopalan [19, Theorem 1], since we use two different types of mixture distributions, and the second mixture enters in a multiplicative manner (instead of an additive one). We can then conclude using standard arguments involving Ville's inequality for positive supermartingales, similarly to Whitehouse et al. [21]. Summarizing, this method enables showing the following result, which is a specialization of the main outcome of this section to the case $\mathcal{H} = \mathcal{H}_K$ and $L_n = K(\cdot, x_n)^\star$.

**Theorem E.6** (Theorem E.14, RKHS case). *Under the notations of Sections E.1 and E.3.1, assume that $\mathcal{H} = \mathcal{H}_K$, where $K$ is UBD with block decomposition $K_0 : \mathcal{X} \times \mathcal{X} \to \tilde{\mathcal{G}}$ and isometry $\iota_{\mathcal{G}} \in \mathcal{L}_b(\tilde{\mathcal{G}} \otimes \mathcal{V}, \mathcal{G})$, where $\tilde{\mathcal{G}}$ and $\mathcal{V}$ are separable Hilbert spaces. Assume that $K_0$ is of trace class, that Assumption E.5 holds, that $L_n = K(\cdot, x_n)^\star$, $n \in \mathbb{N}_{>0}$, and that $\eta$ is 1-subgaussian. Then, there exists an isometry $\iota_{\mathcal{H}_K} : \mathcal{H}_{K_0} \otimes \mathcal{V} \to \mathcal{H}_K$ such that, for all $Q \in \mathcal{L}_b(\mathcal{V})$ self-adjoint, positive semi-definite, and trace-class, and $\tilde{V} \in \mathcal{L}(\mathcal{H}_{K_0})$ self-adjoint, positive definite, diagonal, and with bounded inverse, it holds for all $\delta \in (0, 1)$ that*

$$\mathbb{P}\left[\forall N \in \mathbb{N}_{>0}, \, \|\iota_{\mathcal{H}_K}^{-1} S_N\|_{(\tilde{V}+\tilde{V}_N)^{-1} \otimes Q} \leq \sqrt{2\mathrm{Tr}(Q) \left[\det\left(\mathrm{id}_{\tilde{\mathcal{G}}^N} + \tilde{M}_N \tilde{V}^{-1} \tilde{M}_N^\star\right)\right]^{1/2}}\right] \geq 1 - \delta,$$

*where $\tilde{V}_N = \tilde{M}_N^\star \tilde{M}_N$.*

*Proof.* This follows immediately from Theorem E.14, since Proposition E.12 guarantees that its assumptions hold. □

This result has two consequences that can be immediately leveraged for KRR.

**Corollary E.7** (Corollary E.20, RKHS case). *Under the setup and assumptions of Theorem E.6, do* not *assume that $\eta$ is 1-subgaussian, but rather that there exists $\rho \in \mathbb{R}_{>0}$ and $R_{\mathcal{V}} \in \mathcal{L}_b(\mathcal{V})$ self-adjoint, positive semi-definite, and trace-class, such that $\eta$ is $R$-subgaussian, where $R := \rho^2 \iota_{\mathcal{G}} (\mathrm{id}_{\tilde{\mathcal{G}}} \otimes R_{\mathcal{V}}) \iota_{\mathcal{G}}^{-1}$. Then, for all $\delta \in (0, 1)$,*

$$\mathbb{P}\left[\forall N \in \mathbb{N}_{>0}, \|S_N\|_{(V+V_N)^{-1}} \leq \rho\sqrt{2\mathrm{Tr}(R_{\mathcal{V}}) \ln\left(\frac{1}{\delta}\left[\det(\mathrm{id}_{\tilde{\mathcal{G}}^N} + \tilde{M}_N \tilde{V}^{-1} \tilde{M}_N^\star)\right]^{1/2}\right)}\right] \geq 1 - \delta,$$

*where $V = \iota_{\mathcal{H}}(\tilde{V} \otimes \mathrm{id}_{\mathcal{V}})\iota_{\mathcal{H}}^{-1}$.*

**Corollary E.8** (Corollary E.20, RKHS with trace-class kernel case). *Under the setup and assumptions of Theorem E.6, assume that $K$ itself is of trace class. Also, do* not *assume that $\eta$ is 1-subgaussian, but rather that it is $\rho$-subgaussian for some $\rho \in \mathbb{R}_{>0}$. Then, we can take $K_0 = K$, $\tilde{\mathcal{G}} = \mathcal{G}$, $\mathcal{V} = \mathbb{R}$, and $Q = \mathrm{id}_{\mathbb{R}}$ in Theorem E.6. In particular, for all $\delta \in (0, 1)$,*

$$\mathbb{P}\left[\forall N \in \mathbb{N}_{>0}, \|S_N\|_{(V+V_N)^{-1}} \leq \rho\sqrt{2\ln\left(\frac{1}{\delta}\left[\det(\mathrm{id}_{\mathcal{G}^N} + M_N V^{-1} M_N^\star)\right]^{1/2}\right)}\right] \geq 1 - \delta,$$

*where $V = \tilde{V}$.*

*Proofs of Corollaries E.7 and E.8.* These results follow immediately from Corollary E.20. □

The next two sections are dedicated to stating and proving Theorem E.14 and Corollary E.20, which generalize Theorem E.6 and its corollaries to the case where $\mathcal{H}$ is not necessarily an RKHS. It relies on assuming the existence of the aforementioned appropriate factorization of $\mathcal{H}$ and of the operators $(L_n)_n$, similarly as in the case of a UBD kernel. We thus begin by formalizing the assumption.

### E.3.2 Tensor factorization of Hilbert spaces and coefficients transformations

We have identified that, in the case $\mathcal{H} = \mathcal{H}_{k \cdot \mathrm{id}_{\mathcal{G}}}$, the core assumption for Theorem E.6 was that both $\mathcal{H}$ was the closure of the span of vectors of the form $f \cdot g$, with $f \in \mathcal{H}_k$ and $g \in \mathcal{G}$, and that $L_n$ could be identified with $\tilde{L}_n \otimes \mathrm{id}_{\mathcal{G}}$. This is the idea we generalize with the construction of this section. Specifically, we rigorously introduce tensor factorizations of Hilbert spaces, and formalize the assumption we make on the operators $(L_n)_n$: they should have an appropriate form in the chosen factorization.

**Definition E.9.** Let $\mathcal{H}$ be a separable Hilbert space. A *(tensor) factorization* of $\mathcal{H}$ is a tuple $(H_1, H_2, \iota_{\mathcal{H}})$ such that $H_1$ and $H_2$ are separable Hilbert spaces and $\iota_{\mathcal{H}} \in \mathcal{L}_b(H_1 \otimes H_2, \mathcal{H})$ is an isometric isomorphism. We write $\mathcal{H} \cong_{\iota_{\mathcal{H}}} H_1 \otimes H_2$. The spaces $H_1$ and $H_2$ are respectively called the *coefficient* and *base* spaces of the factorization. If $\mathcal{G}$ is another separable Hilbert space, a tensor factorization of $\mathcal{H}$ and $\mathcal{G}$ with common base space consists of factorizations $\mathcal{H} \cong_{\iota_{\mathcal{H}}} \tilde{\mathcal{H}} \otimes \mathcal{V}$ and $\mathcal{G} \cong_{\iota_{\mathcal{G}}} \tilde{\mathcal{G}} \otimes \mathcal{V}$.

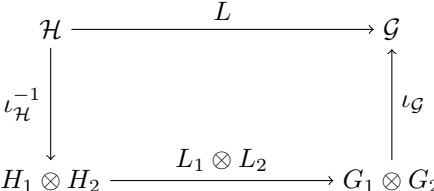

Figure 8: Action of a tensor-factorizable operator $L$ expressed as a commutative diagram. If $K$ is a UBD kernel with block $K_0$, then the RKHS $\mathcal{H}_K$ and sampling operator $K(\cdot, x)^\star$ can be factorized as such with $L_1 = K_0(\cdot, x)^\star$ and $L_2 = \mathrm{id}_\mathcal{V}$, with $\mathcal{V} := H_2 = G_2$, as per Proposition E.12.

The existence of an isometry $\iota_\mathcal{H}$ such that $\mathcal{H} \cong_{\iota_\mathcal{H}} H_1 \otimes H_2$ for given (separable) $\mathcal{H}$, $H_1$, and $H_2$ can be assessed based on purely dimensional arguments: there exists such an $\iota_\mathcal{H}$ if, and only if, $\dim(\mathcal{H}) = \dim(H_1) \cdot \dim(H_2)$ (with $0 \cdot \infty := 0$ and $\infty \cdot \infty := \infty$). For instance, there always exists the trivial factorization of a Hilbert space $\mathcal{H} \cong_{\iota_1} \mathbb{R} \otimes \mathcal{H}$, and if $n = p \cdot q$, $\mathbb{R}^n \cong_{\iota_2} \mathbb{R}^p \otimes \mathbb{R}^q$. It follows that the relationship $\mathcal{H} \cong_{\iota_\mathcal{H}} H_1 \otimes H_2$ does not impose a lot of constraints on any of the spaces involved. The main interest of the notion is when relating it to operators, as operators between Hilbert spaces may have a "nice" form when seen as operators between their factorizations, as illustrated in Figure 8:

**Definition E.10.** Let $\mathcal{H}$ and $\mathcal{G}$ be separable Hilbert spaces, with respective factorizations $\mathcal{H} \cong_{\iota_\mathcal{H}} H_1 \otimes H_2$ and $\mathcal{G} \cong_{\iota_\mathcal{G}} G_1 \otimes G_2$, and let $L \in \mathcal{L}_\mathrm{b}(\mathcal{H}, \mathcal{G})$. We say that $L$ is $(\iota_\mathcal{H}, \iota_\mathcal{G})$-*tensor-factorizable*[6] if there exists a pair $(L_1, L_2) \in \mathcal{L}_\mathrm{b}(H_1, G_1) \times \mathcal{L}_\mathrm{b}(H_2, G_2)$ such that

$$L = \iota_\mathcal{G}(L_1 \otimes L_2)\iota_\mathcal{H}^{-1}.$$

The pair $(L_1, L_2)$ is called an $(\iota_\mathcal{H}, \iota_\mathcal{G})$-factorization of $L$. If now $(L_n)_{n \in \mathbb{N}_{>0}} \subset \mathcal{L}_\mathrm{b}(\mathcal{H}, \mathcal{G})$ is a family of operators, we say that they are co-$(\iota_\mathcal{H}, \iota_\mathcal{G})$-factorizable if $L_n$ is $(\iota_\mathcal{H}, \iota_\mathcal{G})$-factorizable, for all $n \in \mathbb{N}_{>0}$.

Intuitively, an operator $L$ that is $(\iota_\mathcal{H}, \iota_\mathcal{G})$-factorizable can be written as a pure tensor operator when $\mathcal{H}$ and $\mathcal{G}$ are identified with the tensor spaces $H_1 \otimes H_2$ and $G_1 \otimes G_2$. It is clear from this interpretation that *factorizability is not an intrinsic property of an operator*, but largely depends on the chosen factorizations of the Hilbert spaces. For instance, *every* operator is factorizable for the trivial factorizations of $\mathcal{H}$ and $\mathcal{G}$. Co-factorizability enforces that a family of operators consists of pure tensor operators when the factorizations of $\mathcal{H}$ and $\mathcal{G}$ are uniform in the choice of the operator in the family. Finally, we state the assumption the we require for the generalization of Theorem E.6 to hold.

**Definition E.11.** Let $\mathcal{H} \cong_{\iota_\mathcal{H}} \tilde{\mathcal{H}} \otimes \mathcal{V}$ and $\mathcal{G} \cong_{\iota_\mathcal{G}} \tilde{\mathcal{G}} \otimes \mathcal{V}$ be separable Hilbert spaces with joint base space factorizations. Let $L \in \mathcal{L}_\mathrm{b}(\mathcal{H}, \mathcal{G})$. We say that $L$ is an $(\iota_\mathcal{H}, \iota_\mathcal{G})$-*coefficients transformation* if there exists $\tilde{L} \in \mathcal{L}_\mathrm{b}(\tilde{\mathcal{H}}, \tilde{\mathcal{G}})$ such that $(\tilde{L}, \mathrm{id}_\mathcal{V})$ is an $(\iota_\mathcal{H}, \iota_\mathcal{G})$-factorization of $L$.

Crucially, in an RKHS with UBD kernel, the sampling operators $L_n = K(\cdot, x_n)$ are co-factorizable and consist of coefficients transformations. This is formalized in the following result, which justifies that the setup we just introduced is indeed a generalization of the case $\mathcal{H} = \mathcal{H}_K$ with $K$ UBD.

**Proposition E.12.** *Let $\mathcal{H} = \mathcal{H}_K$, where $K : \mathcal{X} \times \mathcal{X} \to \mathcal{L}_\mathrm{b}(\mathcal{G})$ is a kernel. Assume that $K$ is UBD with block decomposition $K_0 : \mathcal{X} \times \mathcal{X} \to \mathcal{L}_\mathrm{b}(\tilde{\mathcal{G}})$ for the isometry $\iota_\mathcal{G} \in \mathcal{L}_\mathrm{b}(\tilde{\mathcal{G}} \otimes \mathcal{V}, \mathcal{G})$, where $\tilde{\mathcal{G}}$ and $\mathcal{V}$ are separable Hilbert spaces. Then, there exists an isometry $\iota_K \in \mathcal{L}_\mathrm{b}(\mathcal{H}_{K_0} \otimes \mathcal{V}, \mathcal{H}_K)$ such that $\mathcal{H} \cong_{\iota_K} \mathcal{H}_{K_0} \otimes \mathcal{V}$, $\mathcal{G} \cong_{\iota_\mathcal{G}} \tilde{\mathcal{G}} \otimes \mathcal{V}$, and the family $L = (L_x)_{x \in \mathcal{X}} = (K(\cdot, x)^\star)_{x \in \mathcal{X}}$ is co-$(\iota_K, \iota_\mathcal{G})$-factorizable and consists of coefficients transformations*

$$L_x = \iota_\mathcal{G}(\tilde{L}_x \otimes \mathrm{id}_\mathcal{V})\iota_K^{-1},$$

*where $\tilde{L}_x = K_0(\cdot, x)^\star$ for all $x \in \mathcal{X}$.*

*Proof.* Define
$$\Psi : u \in \mathcal{H}_{K_0} \otimes \mathcal{V} \mapsto \left(x \mapsto \iota_\mathcal{G}\left(K_0(\cdot, x)^\star \otimes \mathrm{id}_\mathcal{V}\right) u\right),$$

---

[6]For brevity, we refer to this property as simply "$(\iota_\mathcal{H}, \iota_\mathcal{G})$-factorizability" in the remainder of the paper.

and let $\mathcal{F} = \Psi(\mathcal{H}_{K_0} \otimes \mathcal{V})$. We show that $\mathcal{F} = \mathcal{H}_K$ as a Hilbert space when $\mathcal{F}$ is equipped with an appropriate norm.

First, the map $\Psi$ is well-defined, takes values in the space of functions from $\mathcal{X}$ to $\mathcal{G}$, and is linear. Indeed, for any $x \in \mathcal{X}$, $K_0(\cdot, x)^\star$ is a well-defined bounded operator from $\mathcal{H}_{K_0}$ to $\tilde{\mathcal{G}}$, and thus $K_0(\cdot, x)^\star \otimes \mathrm{id}_\mathcal{V} \in \mathcal{L}_\mathrm{b}(\mathcal{H}_{K_0} \otimes \mathcal{V}, \tilde{\mathcal{G}} \otimes \mathcal{V})$. This shows that $\Psi$ is well-defined and function-valued as claimed, and linearity is clear. Second, $\Psi$ is also injective. To see this, take $u \in \ker(\Psi)$. It holds that $(K_0(\cdot, x)^\star \otimes \mathrm{id}_\mathcal{V}) u = 0_{\tilde{\mathcal{G}} \otimes \mathcal{V}}$ for all $x \in \mathcal{X}$, since $\iota_\mathcal{G}$ is injective. Let now $(h_n)_{n \in I_{K_0}}$ and $(v_m)_{m \in I_\mathcal{V}}$ be orthonormal bases (ONBs) of $\mathcal{H}_{K_0}$ and of $\mathcal{V}$, respectively. It is known that $(h_n \otimes v_m)_{(n,m) \in I_{K_0} \times I_\mathcal{V}}$ is an ONB of $\mathcal{H}_{K_0} \otimes \mathcal{V}$. Therefore, let us introduce $(\alpha_{n,m})_{(n,m) \in I_{K_0} \times I_\mathcal{V}}$ such that $u = \sum_{n,m} \alpha_{n,m} h_n \otimes v_m$. By definition of $K_0(\cdot, x)^\star \otimes \mathrm{id}_\mathcal{V}$,

$$(K_0(\cdot, x)^\star \otimes \mathrm{id}_\mathcal{V}) u = \sum_{n,m} \alpha_{n,m} (K_0(\cdot, x)^\star h_n) \otimes v_m = \sum_{n,m} \alpha_{n,m} h_n(x) \otimes v_m.$$

As a result, for any $g \in \tilde{\mathcal{G}}$ and $p \in I_\mathcal{V}$,

$$0 = \langle (K_0(\cdot, x)^\star \otimes \mathrm{id}_\mathcal{V}) u, g \otimes v_p \rangle_{\tilde{\mathcal{G}} \otimes \mathcal{V}} = \left\langle \sum_{n \in I_{K_0}} \alpha_{n,p} \cdot h_n(x), g \right\rangle_{\tilde{\mathcal{G}}},$$

showing that $\sum_{n \in I_{K_0}} \alpha_{n,p} \cdot h_n(x) = 0$ for all $x \in \mathcal{X}$. Therefore, the function $h^{(p)} = \left( x \mapsto \sum_{n \in I_{K_0}} \alpha_{n,p} \cdot h_n(x) \right)$ satisfies $h^{(p)} = 0$. It immediately follows that $h^{(p)} \in \mathcal{H}_{K_0}$; we show that $h^{(p)} = \sum_{n \in I_{K_0}} \alpha_{n,p} \cdot h_n$ where the convergence is in $\|\cdot\|_{K_0}$. It follows from the definition of $h^{(p)}$ that this results from the convergence in $\|\cdot\|_{K_0}$ of the series $\sum_{n \in I_{K_0}} \alpha_{n,p} \cdot h_n$, which in turn holds since

$$\left\| \sum_{n \in I_{K_0}} \alpha_{n,p} \cdot h_n \right\|_{K_0} = \sum_{n,n'} \alpha_{n,p} \alpha_{n',p} \langle h_n, h_{n'} \rangle_{K_0} = \sum_n \alpha_{n,p}^2 \leq \sum_{n,m} \alpha_{n,m}^2 = \|u\|_{\mathcal{H}_{K_0} \otimes \mathcal{V}} < \infty.$$

Therefore, since $(h_n)_{n \in I_{K_0}}$ is an ONB of $\mathcal{H}_{K_0}$, we deduce that $\alpha_{n,p} = 0$ for all $n \in I_{K_0}$. Since this holds for all $p \in I_\mathcal{V}$, this shows that $u = 0$ and that $\Psi$ is injective. It follows that $\Psi$ is a bijection between $\mathcal{H}_{K_0} \otimes \mathcal{V}$ and $\mathcal{F}$, since $\mathcal{F}$ is defined as its range.

We now show that the evaluation operator in $x \in \mathcal{X}$, denoted as $S_x \in \mathcal{L}(\mathcal{F}, \mathcal{G})$, satisfies

$$S_x = \iota_\mathcal{G} (K_0(\cdot, x)^\star \otimes \mathrm{id}_\mathcal{V}) \Psi^{-1}. \tag{25}$$

Indeed, for any $f \in \mathcal{F}$, letting $u = \Psi^{-1} f$,

$$S_x f = S_x \Psi u = (\Psi u)(x) = \iota_\mathcal{G} (K_0(\cdot, x)^\star \otimes \mathrm{id}_\mathcal{V}) u = \iota_\mathcal{G} (K_0(\cdot, x)^\star \otimes \mathrm{id}_\mathcal{V}) \Psi^{-1} f,$$

showing the claim. Finally, we equip $\mathcal{F}$ with the norm $\|\cdot\|_\mathcal{F}$ defined for all $f \in \mathcal{F}$ as

$$\|f\|_\mathcal{F} = \|\Psi^{-1} f\|_{\mathcal{H}_{K_0} \otimes \mathcal{V}}.$$

It is immediate to verify that $\|\cdot\|_\mathcal{F}$ is indeed a norm, by bijectivity of $\Psi$. Furthermore, $\Psi$ is an isometry, since for all $u \in \mathcal{H}_{K_0} \otimes \mathcal{V}$, $\|u\|_{\mathcal{H}_{K_0} \otimes \mathcal{V}} = \|\Psi^{-1} \Psi u\|_{\mathcal{H}_{K_0} \otimes \mathcal{V}} = \|\Psi u\|_\mathcal{F}$. This suffices to show that $(\mathcal{F}, \|\cdot\|_\mathcal{F})$ is a Hilbert space. It follows immediately from (25) that $S_x$ is bounded for all $x \in \mathcal{X}$, showing that $\mathcal{F}$ is an RKHS. Furthermore, its kernel $K_\mathcal{F}$ is defined for all $(x, x') \in \mathcal{X}^2$ as

$$K_\mathcal{F}(x, x') = S_x S_{x'}^\star$$
$$= \iota_\mathcal{G} (K_0(\cdot, x)^\star \otimes \mathrm{id}_\mathcal{V}) \Psi^{-1} \left[ \iota_\mathcal{G} (K_0(\cdot, x)^\star \otimes \mathrm{id}_\mathcal{V}) \Psi^{-1} \right]^\star$$
$$= \iota_\mathcal{G} \left[ (K_0(\cdot, x)^\star K_0(\cdot, x')) \otimes \mathrm{id}_\mathcal{V} \right] \iota_\mathcal{G}^{-1}$$
$$= \iota_\mathcal{G} K_0(x, x') \iota_\mathcal{G}^{-1}$$
$$= K(x, x').$$

It follows from uniqueness of the RKHS associated to a kernel that $\mathcal{F} = \mathcal{H}_K$ as Hilbert spaces.

We are now ready to conclude. Taking $\iota_K = \Psi$, we have shown that $\mathcal{H}_K = \iota_K(\mathcal{H}_{K_0} \otimes \mathcal{V})$, with $\iota_K$ an isometry. Furthermore, for all $x \in \mathcal{X}$, it follows from (25) that

$$K(\cdot, x)^\star = \iota_\mathcal{G} (K_0(\cdot, x)^\star \otimes \mathrm{id}_\mathcal{V}) \iota_K^{-1},$$

showing that the family $(K(\cdot, x)^\star)_{x \in \mathcal{X}}$ is co-$(\iota_K, \iota_\mathcal{G})$-factorizable and consists of the claimed coefficients transformations. $\qquad\square$

### E.3.3 Self-normalized concentration of vector-valued martingales with coefficients-transforming measurements

The main result of this section is the following theorem, which is the generalized version of Theorem E.6. We name the relevant assumptions for future reference.

**Assumption E.13.** *Under the notations of Section E.1, $\mathcal{H}$ and $\mathcal{G}$ admit the factorizations with joint base space $\mathcal{H} \cong_{\iota_{\mathcal{H}}} \tilde{\mathcal{H}} \otimes \mathcal{V}$ and $\mathcal{G} \cong_{\iota_{\mathcal{G}}} \tilde{\mathcal{G}} \otimes \mathcal{V}$, $(L_n)_{n \in \mathbb{N}_{>0}}$ is co-$(\iota_{\mathcal{H}}, \iota_{\mathcal{G}})$-factorizable and consists of coefficient transformations, a.s., and $(\tilde{L}_n, \mathrm{id}_{\mathcal{V}})$ is an $(\iota_{\mathcal{H}}, \iota_{\mathcal{G}})$-factorization of $L_n$ with $\tilde{L}_n \tilde{L}_n^{\star}$ of trace class, a.s. and for all $n \in \mathbb{N}_{>0}$.*

**Theorem E.14.** *Under the notations of Section E.1, assume Assumptions E.5 and E.13 hold. Assume that $\eta$ is 1-subgaussian conditionally on $\mathcal{F}$. Let $Q \in \mathcal{L}_{\mathrm{b}}(\mathcal{V})$ be self-adjoint, positive semi-definite, and trace-class, and $\tilde{V} \in \mathcal{L}(\tilde{\mathcal{H}})$ be self-adjoint, positive definite, diagonal, and with bounded inverse. Then, $\tilde{M}_N \tilde{V}^{-1} \tilde{M}_N^{\star}$ is a.s. of trace class for all $N \in \mathbb{N}_{>0}$ and, for all $\delta \in (0, 1)$, it holds with probability at least $1 - \delta$ that*

$$\forall N \in \mathbb{N}_{>0}, \|\iota_{\mathcal{H}}^{-1} S_N\|_{(\tilde{V}+\tilde{V}_N)^{-1} \otimes Q} \le \sqrt{2\mathrm{Tr}(Q) \ln\left(\frac{1}{\delta}\left[\det(\mathrm{id}_{\tilde{\mathcal{G}}^N} + \tilde{M}_N \tilde{V}^{-1} \tilde{M}_N^{\star})\right]^{1/2}\right)}, \quad (26)$$

*where we introduced*

$$\tilde{M}_N : \tilde{h} \in \tilde{\mathcal{H}} \mapsto (\tilde{L}_n^{\star}\tilde{h})_{n \in [N]} \in \tilde{\mathcal{G}}^N, \quad \text{and} \quad \tilde{V}_N = \tilde{M}_N^{\star}\tilde{M}_N \in \mathcal{L}_{\mathrm{b}}(\tilde{\mathcal{H}}). \quad (27)$$

We prove this theorem first under the assumption that $\tilde{V}^{-1}$ is of trace class rather than $\tilde{L}_n \tilde{L}_n^{\star}$ is, as it enables detailing all of the arguments in a technically simpler setup. We then generalize the proof to handle the setup of Theorem E.14.

**Lemma E.15.** *Under the same setup and assumptions as Theorem E.14, do* not *assume that $\tilde{L}_n \tilde{L}_n^{\star}$ is of trace class for all $n \in \mathbb{N}_{>0}$, but rather that $\tilde{V}^{-1}$ is of trace class. Then, for all $\delta \in (0, 1)$, (26) holds.*

To show this, we follow the method announced in Section E.3.1.

**Lemma E.16.** *Under the setup and assumptions as Lemma E.15, define for all $N \in \mathbb{N}_{>0}$*

$$G_N : \tilde{\mathcal{H}} \otimes \mathcal{V} \to \mathbb{R}_{\ge 0}$$
$$h \mapsto \exp\left[\langle \iota_{\mathcal{H}} h, S_N \rangle_{\mathcal{H}} - \frac{1}{2}\|\iota_{\mathcal{H}} h\|_{V_N}^2\right]. \quad (28)$$

*Then, for all $v \in \mathcal{V}$,*

$$\int_{\tilde{\mathcal{H}}} G_N[\tilde{h} \otimes v]\mathrm{d}\mathcal{N}(\tilde{h} \mid 0, \tilde{V}^{-1})$$
$$= \left[\det\left(\mathrm{id}_{\tilde{\mathcal{G}}^N} + \|v\|^2 \tilde{M}_N \tilde{V}^{-1} \tilde{M}_N\right)\right]^{-1/2} \exp\left[\frac{1}{2}\|\tilde{S}_N(v)\|_{(\tilde{V}+\|v\|^2 \tilde{V}_N)^{-1}}^2\right], \quad (29)$$

*where*

$$\tilde{S}_N(v) = \sum_{n=1}^N \mathrm{Cont}(v)(\tilde{L}_n^{\star} \otimes \mathrm{id}_{\mathcal{V}})\iota_{\mathcal{G}}^{\star}\eta_n,$$

*where $\mathrm{Cont}(v) \in \mathcal{L}_{\mathrm{b}}(\tilde{\mathcal{H}} \otimes \mathcal{V}, \tilde{\mathcal{H}})$ is the tensor contraction of $v$, that is, the unique bounded operator that satisfies $\mathrm{Cont}(v)(u \otimes w) = \langle v, w \rangle \cdot u$ for all $u \otimes w \in \tilde{\mathcal{H}} \otimes \mathcal{V}$.*

*Proof.* For $v = 0$, the claim is trivial, so assume that $v \neq 0$. We have

$$
\begin{aligned}
\langle \iota_{\mathcal{H}}(\tilde{h} \otimes v), S_N \rangle_{\mathcal{H}} &= \sum_{n=1}^{N} \langle \iota_{\mathcal{H}}(\tilde{h} \otimes v), L_n^{\star} \eta_n \rangle_{\mathcal{H}} \\
&= \sum_{n=1}^{N} \langle L_n \iota_{\mathcal{H}}(\tilde{h} \otimes v), \eta_n \rangle_{\mathcal{G}} \\
&= \sum_{n=1}^{N} \langle \iota_{\mathcal{G}}(\tilde{L}_n \otimes \mathrm{id}_{\mathcal{V}}) \iota_{\mathcal{H}}^{-1} \iota_{\mathcal{H}}(\tilde{h} \otimes v), \eta_n \rangle_{\mathcal{G}} \\
&= \sum_{n=1}^{N} \langle (\tilde{L}_n \otimes \mathrm{id}_{\mathcal{V}})(\tilde{h} \otimes v), \iota_{\mathcal{G}}^{\star} \eta_n \rangle_{\tilde{\mathcal{G}} \otimes \mathcal{V}} \\
&= \sum_{n=1}^{N} \langle \tilde{h} \otimes v, (\tilde{L}_n^{\star} \otimes \mathrm{id}_{\mathcal{V}}) \iota_{\mathcal{G}}^{\star} \eta_n \rangle_{\tilde{\mathcal{H}} \otimes \mathcal{V}} \\
&= \sum_{n=1}^{N} \langle \tilde{h}, \mathrm{Cont}(v)(\tilde{L}_n^{\star} \otimes \mathrm{id}_{\mathcal{V}}) \iota_{\mathcal{G}}^{\star} \eta_n \rangle_{\tilde{\mathcal{H}}} \\
&= \langle \tilde{h}, \tilde{S}_N(v) \rangle_{\tilde{\mathcal{H}}}.
\end{aligned}
$$

Furthermore,

$$
\begin{aligned}
\| \iota_{\mathcal{H}}(\tilde{h} \otimes v) \|_{V_N}^2 &= \sum_{n=1}^{N} \| L_n \iota_{\mathcal{H}}(\tilde{h} \otimes v) \|_{\mathcal{G}}^2 \\
&= \sum_{n=1}^{N} \| \iota_{\mathcal{G}}(\tilde{L}_n \otimes \mathrm{id}_{\mathcal{V}})(\tilde{h} \otimes v) \|_{\mathcal{G}}^2 \\
&= \sum_{n=1}^{N} \| (\tilde{L}_n \otimes \mathrm{id}_{\mathcal{V}})(\tilde{h} \otimes v) \|_{\tilde{\mathcal{G}} \otimes \mathcal{V}}^2 \\
&= \sum_{n=1}^{N} \| \tilde{L}_n \tilde{h} \|_{\tilde{\mathcal{G}}}^2 \| v \|_{\mathcal{V}}^2 \\
&= \| \tilde{h} \|_{\tilde{V}_N} \| v \|_{\mathcal{V}}^2.
\end{aligned}
$$

Now, since $v \neq 0$,

$$
\begin{aligned}
\int_{\tilde{\mathcal{H}}} G_N(\tilde{h} \otimes v) \mathrm{d}\mathcal{N}(\tilde{h} \mid 0, \tilde{V}^{-1}) &= \int_{\tilde{\mathcal{H}}} \exp \left[ \langle \tilde{h}, \tilde{S}_N(v) \rangle_{\tilde{\mathcal{H}}} - \frac{1}{2} \| v \|_{\mathcal{V}}^2 \| \tilde{h} \|_{\tilde{V}_N}^2 \right] \mathrm{d}\mathcal{N}(\tilde{h} \mid 0, \tilde{V}^{-1}) \\
&= \int_{\tilde{\mathcal{H}}} \exp \left[ \langle u, \tilde{S}_N(\bar{v}) \rangle_{\tilde{\mathcal{H}}} - \frac{1}{2} \| u \|_{\tilde{V}_N}^2 \right] \mathrm{d}\mathcal{N}(u \mid 0, \| v \|_{\mathcal{V}} \tilde{V}^{-1}),
\end{aligned}
$$

by performing the change of variables $u = \| v \|_{\mathcal{V}} \tilde{h}$ and introducing $\bar{v} = \| v \|_{\mathcal{V}}^{-1} v$. Since

$$
(\| v \|_{\mathcal{V}}^2 \tilde{V}^{-1})^{1/2} (-\tilde{V}_N)(\| v \|_{\mathcal{V}}^2 \tilde{V}^{-1})^{1/2} = -\| v \|_{\mathcal{V}}^2 \tilde{V}^{-1/2} \tilde{V}_N \tilde{V}^{-1/2} \prec \mathrm{id}_{\tilde{\mathcal{H}}},
$$

the resulting integral can be evaluated using Proposition 1.2.8 in Da Prato and Zabczyk [44], showing that it is equal to

$$
\left[ \det \left( \mathrm{id}_{\tilde{\mathcal{H}}} + \| v \|_{\mathcal{V}}^2 \tilde{V}^{-1/2} \tilde{V}_N \tilde{V}^{-1/2} \right) \right]^{-1/2}
$$

$$
\times \exp \left[ \frac{1}{2} \left\| \left( \mathrm{id}_{\tilde{\mathcal{H}}} + \| v \|_{\mathcal{V}}^2 \tilde{V}^{-1/2} \tilde{V}_N \tilde{V}^{-1/2} \right)^{-1/2} \| v \|_{\mathcal{V}} \tilde{V}^{-1/2} \tilde{S}_N(\bar{v}) \right\|_{\tilde{\mathcal{H}}}^2 \right].
$$

We handle the determinant by leveraging the Weinstein-Aronszajn identity for Fredholm determinants, which states that $\det(\mathrm{id} + AB) = \det(\mathrm{id} + BA)$, where $A$ and $B$ are operators which can be composed

in both directions with $AB$ and $BA$ both of trace class and the identity is taken on the appropriate space. In this case, by noting that $\tilde{V}_N = \tilde{M}_N^\star \tilde{M}_N$, we take $A = B^\star = \|v\|_\mathcal{V} \tilde{V}^{-1/2} \tilde{M}_N^\star$ and obtain

$$\det\left(\mathrm{id}_{\tilde{\mathcal{H}}} + \|v\|_\mathcal{V}^2 \tilde{V}^{-1/2} \tilde{V}_N \tilde{V}^{-1/2}\right) = \det\left(\mathrm{id}_{\tilde{\mathcal{G}}^N} + \|v\|_\mathcal{V}^2 \tilde{M}_N \tilde{V}^{-1} \tilde{M}_N^\star\right).$$

We now rewrite the squared norm in the argument of the exponential as

$$
\begin{aligned}
\left\| \left(\mathrm{id}_{\tilde{\mathcal{H}}} + \|v\|_\mathcal{V}^2 \tilde{V}^{-1/2} \tilde{V}_N \tilde{V}^{-1/2}\right)^{-1/2} \|v\|_\mathcal{V} \tilde{V}^{-1/2} \tilde{S}_N(\bar{v}) \right\|_{\tilde{\mathcal{H}}}^2 & \\
= \left\langle \tilde{V}^{-1/2} \left(\mathrm{id}_{\tilde{\mathcal{H}}} + \|v\|_\mathcal{V}^2 \tilde{V}^{-1/2} \tilde{V}_N \tilde{V}^{-1/2}\right)^{-1} \tilde{V}^{-1/2} \tilde{S}_N(v), \tilde{S}_N(v) \right\rangle_{\tilde{\mathcal{H}}} & \\
= \left\langle \left(\tilde{V} + \|v\|_\mathcal{V}^2 \tilde{V}_N\right)^{-1} \tilde{S}_N(v), \tilde{S}_N(v) \right\rangle_{\tilde{\mathcal{H}}} & \\
= \|\tilde{S}_N(v)\|_{(\tilde{V} + \|v\|_\mathcal{V}^2 \tilde{V}_N)^{-1}}, &
\end{aligned}
\tag{30}
$$

where we used $\|v\|_\mathcal{V} \tilde{S}_N(\bar{v}) = \tilde{S}_N(v)$ in the first equality. Altogether, we found (29). $\qquad\square$

We now introduce a second mixture on the resulting integral (29) to handle the remaining dependency in $v \in \mathcal{V}$. We choose for the measure of the mixture the one given in the following lemma.

**Lemma E.17.** *Let $Q \in \mathcal{L}_\mathrm{b}(\mathcal{G})$ be self-adjoint, positive semi-definite, and trace-class, with $Q \neq 0$. Then, there exists a Borel probability measure $\mu_Q$ on $\mathcal{G}$ such that $\mathrm{supp}(\mu) \subset \mathbb{S}_\mathcal{G}^1$, the unit sphere in $\mathcal{G}$, and*

$$\int_\mathcal{G} g \otimes g \mathrm{d}\mu(g) = \frac{1}{\mathrm{Tr}(Q)} Q.$$

*Proof.* Under the assumptions of the lemma, the operator $Q$ is also compact. The spectral theorem for compact, self-adjoint operators then ensures the existence of a complete orthonormal family $(e_j)_{j \in J}$ and a summable sequence $(\xi_j)_{j \in J}$ such that $Q = \sum_{j \in J} \xi_j e_j \otimes e_j$. Define now

$$\nu_Q = \sum_{j \in J} \xi_j \delta_{\{e_j\}}.$$

It is immediate to verify that $\nu_Q$ is a nonnegative measure with support included in $\{e_j \mid j \in J\} \subset \mathbb{S}_\mathcal{G}^1$, where nonnegativity comes from the fact that $\xi_j \geq 0$ for all $j \in J$ since $Q$ is positive semi-definite. Furthermore, for any $u, v \in \mathcal{G}$, we get using standard calculation rules for Bochner- and Lebesgue integrals,

$$
\begin{aligned}
\left\langle u, \int_\mathcal{G} g \otimes g \mathrm{d}\nu_Q(g) v \right\rangle_\mathcal{G} &= \int_\mathcal{G} \langle u, (g \otimes g)v \rangle_\mathcal{G} \mathrm{d}\left(\sum_{j \in J} \xi_j \delta_{\{e_j\}}\right)(g) \\
&= \sum_{j \in J} \xi_j \int_\mathcal{G} \langle u, (g \otimes g)v \rangle_\mathcal{G} \mathrm{d}\left(\delta_{\{e_j\}}\right)(g) \\
&= \sum_{j \in J} \xi_j \langle u, (e_j \otimes e_j)v \rangle_\mathcal{G} \\
&= \left\langle u, \left(\sum_{j \in J} \xi_j (e_j \otimes e_j)\right) v \right\rangle_\mathcal{G} \\
&= \langle u, Qv \rangle_\mathcal{G},
\end{aligned}
$$

showing that $\int_\mathcal{G} g \otimes g \mathrm{d}\nu_Q(g) = Q$. Finally,

$$\nu_Q(\mathcal{G}) = \sum_{j \in J} \xi_j \delta_{\{e_j\}}(\mathcal{G}) = \sum_{j \in J} \xi_j = \mathrm{Tr}(Q) \in \mathbb{R}_{>0},$$

and thus taking $\mu_Q = \frac{1}{\mathrm{Tr}(Q)} \nu_Q$ satisfies all of the requirements. $\qquad\square$

**Lemma E.18.** *Under the setup and assumptions of Lemma E.15, let $G_N$ be defined for all $N \in \mathbb{N}_{>0}$ as in (28). Assume that $Q \neq 0$, and let $\mu_Q$ be a probability measure on $\mathcal{V}$ as given in Lemma E.17. Then,*

$$\int_\mathcal{V} \int_{\tilde{\mathcal{H}}} G_N(\tilde{h} \otimes v) \mathrm{d}\mathcal{N}(\tilde{h} \mid 0, \tilde{V}^{-1}) \mathrm{d}\mu_Q(v)$$

$$\geq \left[ \det \left( \mathrm{id}_{\tilde{\mathcal{G}}^N} + \tilde{M}_N \tilde{V}^{-1} \tilde{M}_N^\star \right) \right]^{-1/2} \exp \left[ \frac{1}{2\mathrm{Tr}(Q)} \| \iota_{\tilde{\mathcal{H}}}^{-1} S_N \|_{(\tilde{V} + \tilde{V}_N)^{-1} \otimes Q}^2 \right].$$

*Proof.* It follows from Lemma E.16 and the fact that $\mu_Q$ has support in $\mathbb{S}_\mathcal{V}^1$ that

$$\int_\mathcal{V} \int_{\tilde{\mathcal{H}}} G_N(\tilde{h} \otimes v) \mathrm{d}\mathcal{N}(\tilde{h} \mid 0, \tilde{V}^{-1}) \mathrm{d}\mu_Q(v)$$

$$= \int_{\mathbb{S}_\mathcal{V}^1} \left[ \det \left( \mathrm{id}_{\tilde{\mathcal{G}}^N} + \|v\|_\mathcal{V}^2 \tilde{M}_N \tilde{V}^{-1} \tilde{M}_N \right) \right]^{-1/2} \exp \left[ \frac{1}{2} \| \tilde{S}_N(v) \|_{(\tilde{V} + \|v\|_\mathcal{V}^2 \tilde{V}_N)^{-1}}^2 \right] \mathrm{d}\mu_Q(v)$$

$$= \left[ \det \left( \mathrm{id}_{\tilde{\mathcal{G}}^N} + \tilde{M}_N \tilde{V}^{-1} \tilde{M}_N \right) \right]^{-1/2} \int_{\mathbb{S}_\mathcal{V}^1} \exp \left[ \frac{1}{2} \| \tilde{S}_N(v) \|_{(\tilde{V} + \tilde{V}_N)^{-1}}^2 \right] \mathrm{d}\mu_Q(v).$$

Now, by Jensen's inequality,

$$\int_{\mathbb{S}_\mathcal{V}^1} \exp \left[ \frac{1}{2} \| \tilde{S}_N(v) \|_{(\tilde{V} + \tilde{V}_N)^{-1}}^2 \right] \mathrm{d}\mu_Q(v) \geq \exp \left[ \frac{1}{2} \int_{\mathbb{S}_\mathcal{V}^1} \| \tilde{S}_N(v) \|_{(\tilde{V} + \tilde{V}_N)^{-1}}^2 \mathrm{d}\mu_Q(v) \right].$$

Let us now introduce families $(e_i^{(n)})_{i \in J} \subset \tilde{\mathcal{G}}$ and $(f_i^{(n)})_{i \in J} \subset \mathcal{V}$ such that $\iota_\mathcal{G}^\star \eta_n = \sum_{i \in J} e_i^{(n)} \otimes f_i^{(n)}$, for all $n \in \mathbb{N}_{>0}$. We have

$$\int_{\mathbb{S}_\mathcal{V}^1} \| \tilde{S}_N(v) \|_{(\tilde{V} + \tilde{V}_N)^{-1}}^2 \mathrm{d}\mu_Q(v)$$

$$= \int_{\mathbb{S}_\mathcal{V}^1} \sum_{n,m=1}^N \sum_{i,j \in J} \langle (\tilde{V} + \tilde{V}_N)^{-1} \mathrm{Cont}(v)(\tilde{L}_n^\star \otimes \mathrm{id}_\mathcal{V})(e_i^{(n)} \otimes f_i^{(n)}),$$

$$\mathrm{Cont}(v)(\tilde{L}_m^\star \otimes \mathrm{id}_\mathcal{V})(e_j^{(m)} \otimes f_j^{(m)}) \rangle_{\tilde{\mathcal{H}}} \mathrm{d}\mu_Q(v)$$

$$= \int_{\mathbb{S}_\mathcal{V}^1} \sum_{n,m=1}^N \sum_{i,j \in J} \langle v, f_i^{(n)} \rangle_\mathcal{V} \langle v, f_j^{(m)} \rangle_\mathcal{V} \langle (\tilde{V} + \tilde{V}_N)^{-1} \tilde{L}_n^\star e_i^{(n)}, \tilde{L}_m^\star e_j^{(m)} \rangle_{\tilde{\mathcal{H}}} \mathrm{d}\mu_Q(v)$$

$$= \sum_{n,m=1}^N \sum_{i,j \in J} \langle (\tilde{V} + \tilde{V}_N)^{-1} \tilde{L}_n^\star e_i^{(n)}, \tilde{L}_m^\star e_j^{(m)} \rangle_{\tilde{\mathcal{H}}} \int_{\mathbb{S}_\mathcal{V}^1} \langle v, f_i^{(n)} \rangle_\mathcal{V} \langle v, f_j^{(m)} \rangle_\mathcal{V} \mathrm{d}\mu_Q(v)$$

$$= \frac{1}{\mathrm{Tr}(Q)} \sum_{n,m=1}^N \sum_{i,j \in J} \langle (\tilde{V} + \tilde{V}_N)^{-1} \tilde{L}_n^\star e_i^{(n)}, \tilde{L}_m^\star e_j^{(m)} \rangle_{\tilde{\mathcal{H}}} \langle Q f_i^{(n)}, f_j^{(m)} \rangle_\mathcal{V}$$

$$= \frac{1}{\mathrm{Tr}(Q)} \sum_{n,m=1}^N \sum_{i,j \in J} \left\langle \left[ (\tilde{V} + \tilde{V}_N)^{-1} \otimes Q \right] (\tilde{L}_n^\star \otimes \mathrm{id}_\mathcal{V})(e_i^{(n)} \otimes f_i^{(n)}), \right.$$

$$\left. (\tilde{L}_m^\star \otimes \mathrm{id}_\mathcal{V})(e_j^{(m)} \otimes f_j^{(m)}) \right\rangle_{\tilde{\mathcal{H}} \otimes \mathcal{V}}$$

$$= \frac{1}{\mathrm{Tr}(Q)} \sum_{n,m=1}^N \left\langle \left[ (\tilde{V} + \tilde{V}_N)^{-1} \otimes Q \right] (\tilde{L}_n^\star \otimes \mathrm{id}_\mathcal{V}) \iota_\mathcal{G}^\star \eta_n, (\tilde{L}_m^\star \otimes \mathrm{id}_\mathcal{V}) \iota_\mathcal{G}^\star \eta_m \right\rangle_{\tilde{\mathcal{H}} \otimes \mathcal{V}},$$

where we used the fact that $\mathrm{Cont}(v)(u \otimes w) = \langle v, w \rangle_\mathcal{V} u$ for the second equality and the definition of $\mu_Q$ for the third one. We now leverage the fact that

$$S_N = \sum_{n=1}^N L_n^\star \eta_n = \sum_{n=1}^N (\iota_{\tilde{\mathcal{H}}}^{-1})^\star (\tilde{L}_n^\star \otimes \mathrm{id}_\mathcal{V}) \iota_\mathcal{G}^\star \eta_n = \sum_{n=1}^N \iota_{\tilde{\mathcal{H}}} (\tilde{L}_n^\star \otimes \mathrm{id}_\mathcal{V}) \iota_\mathcal{G}^\star \eta_n,$$

showing that

$$\int_{\mathbb{S}_{\mathcal{V}}^1} \|\tilde{S}_N(v)\|^2_{(\tilde{V}+\tilde{V}_N)^{-1}} \mathrm{d}\mu_Q(v) = \frac{1}{\mathrm{Tr}(Q)} \left\langle \left[(\tilde{V}+\tilde{V}_N)^{-1} \otimes Q\right] \iota_{\mathcal{H}}^{-1} S_N, \iota_{\mathcal{H}}^{-1} S_N \right\rangle_{\tilde{\mathcal{H}} \otimes \mathcal{V}}$$

$$= \frac{1}{\mathrm{Tr}(Q)} \|\iota_{\mathcal{H}}^{-1} S_N\|_{(\tilde{V}+\tilde{V}_N)^{-1} \otimes Q},$$

and concluding the proof. $\qquad\square$

**Lemma E.19.** *Under the setup and assumption of Lemma E.15, define $G_N$ as in* (28). *Then, for all* $h \in \tilde{\mathcal{H}} \otimes \mathcal{V}$, $(G_N(h))_N$ *is an $\mathcal{F}$-supermartingale and* $\mathbb{E}[G_N(h)] \leq 1$ *for all* $N \in \mathbb{N}_{>0}$.

*Proof.* Let $h \in \tilde{\mathcal{H}} \otimes \mathcal{V}$, and define $G_0(h) = 1$ a.s. We have

$$\mathbb{E}[G_N(h) \mid \mathcal{F}_{N-1}]$$

$$= \mathbb{E}\left[\exp\left(\sum_{n=1}^N \langle \iota_{\mathcal{H}} h, L_n^\star \eta_n \rangle_{\mathcal{H}} - \frac{1}{2}\|\iota_{\mathcal{H}} h\|_{L_n^\star L_n}\right)\middle| \mathcal{F}_{N-1}\right]$$

$$= \mathbb{E}\left[\exp\left(-\frac{1}{2}\langle \iota_{\mathcal{H}} h, L_N^\star L_N \iota_{\mathcal{H}} h \rangle_{\mathcal{H}}\right)\exp(\langle L_N \iota_{\mathcal{H}} h, \eta_N \rangle_{\mathcal{G}})G_{N-1}(h)\middle| \mathcal{F}_{N-1}\right]$$

$$= \exp\left(-\frac{1}{2}\|L_N \iota_{\mathcal{H}} h\|_{\mathcal{G}}^2\right)\mathbb{E}\left[\exp(\langle L_N \iota_{\mathcal{H}} h, \eta_N \rangle_{\mathcal{G}}) \mid \mathcal{F}_{N-1}\right] \cdot G_{N-1}(h)$$

$$\leq \exp\left(-\frac{1}{2}\|L_N \iota_{\mathcal{H}} h\|_{\mathcal{G}}^2\right)\exp\left(\frac{1}{2}\|L_N \iota_{\mathcal{H}} h\|_{\mathcal{G}}^2\right) \cdot G_{N-1}(h)$$

$$= G_{N-1}(h),$$

where we used the fact that $L_N$ and $G_{N-1}(h)$ are $\mathcal{F}_{N-1}$-measurable and 1-subgaussianity of $\eta_N$ conditionally on $\mathcal{F}_{N-1}$. This shows that $(G_N(h))_N$ is a supermartingale, and it follows immediately that $\mathbb{E}[G_N(h)] \leq \mathbb{E}[G_0(h)] = 1$, concluding the proof. $\qquad\square$

*Proof of Lemma E.15.* First, the claim is trivial if $Q = 0$. Assume that $Q \neq 0$. Under the assumptions of the lemma, we apply Lemma E.19 and obtain that $\mathbb{E}[G_N(h)] \leq 1$ for all $N \in \mathbb{N}_{>0}$ and $h \in \tilde{\mathcal{H}} \otimes \mathcal{V}$. In particular, this is the case for any $h$ of the form $h = \tilde{h} \otimes v$, where $\tilde{h} \in \tilde{\mathcal{H}}$ and $v \in \mathcal{V}$. Therefore, by Fubini's theorem, the process $(G_N)_N$ defined for all $N \in \mathbb{N}_{>0}$ as

$$G_N = \int_{\mathcal{V}} \int_{\tilde{\mathcal{H}}} G_N(\tilde{h} \otimes v) \mathrm{d}\mathcal{N}(\tilde{h} \mid 0, \tilde{V}^{-1}) \mathrm{d}\mu_Q(v),$$

is again a supermartingale with $\mathbb{E}[G_N] \leq 1$. As a result, we can apply Ville's inequality and obtain that for any $\delta \in (0, 1)$

$$\mathbb{P}\left[\forall N \in \mathbb{N}_{>0}, G_N \leq \frac{1}{\delta}\right] \geq 1 - \delta.$$

Lemma E.18 now shows that it holds with probability at least $1 - \delta$ that

$$\forall N \in \mathbb{N}_{>0}, \left(\det\left(\mathrm{id}_{\tilde{\mathcal{G}}^N} + \tilde{M}_N \tilde{V}^{-1} M_N\right)\right)^{-1/2} \exp\left(\frac{1}{2\mathrm{Tr}(Q)}\|\iota_{\mathcal{H}}^{-1} S_N\|^2_{(\tilde{V}+\tilde{V}_N)^{-1} \otimes Q}\right) \leq \frac{1}{\delta}.$$

The result follows by rearranging. $\qquad\square$

We are now ready to prove Theorem E.14. It does not directly follow from Lemma E.15, but rather its proof is an adaptation of that of this lemma.

*Proof of Theorem E.14.* Take $D = \dim(\tilde{\mathcal{H}}) \in \mathbb{N}_{>0} \cup \{\infty\}$, $(b_m)_{m \in [D]}$ an ONB of $\tilde{\mathcal{H}}$, and a family $(\lambda_m)_{m \in [D]} \subset \mathbb{R}_{\geq 0}$ such that $\tilde{V} = \sum_{m=1}^D \lambda_m b_m \otimes b_m$, which all exist since $\tilde{V}$ is diagonal. The inverse of $\tilde{V}$ exists and is given by

$$\tilde{V}^{-1} = \sum_{m=1}^D \lambda_m^{-1} b_m \otimes b_m.$$

Note that $\tilde{V}^{-1}$ is not necessarily of trace class, but it is bounded by assumption. Define for $M \in [D]$

$$W_M = \sum_{m=1}^{M} \lambda_m^{-1} b_m \otimes b_m,$$

and observe that $W_M$ is of rank $M$ and hence trivially of trace class, and it is self-adjoint and positive semidefinite. Although Lemma E.16 does not directly apply by setting $\tilde{V}^{-1}$ to $W_M$ since $W_M$ may not be invertible, we see from the proof that the only point where this invertibility is used is in (30). Therefore, the proof carries over until this point, showing that for all $v \in \mathcal{V}$

$$\int_{\tilde{\mathcal{H}}} G_N(\tilde{h} \otimes v) \mathrm{d}\mathcal{N}(\tilde{h} \mid 0, W_M)$$

$$= \left[ \det \left( \mathrm{id}_{\tilde{\mathcal{G}}^N} + \|v\|_{\mathcal{V}}^2 \tilde{M}_N W_M \tilde{M}_N^\star \right) \right]^{-1/2}$$

$$\times \exp \left[ \frac{1}{2} \left\| \left( \mathrm{id}_{\tilde{\mathcal{H}}} + \|v\|_{\mathcal{V}}^2 W_M^{1/2} \tilde{V}_N W_M^{1/2} \right)^{-1/2} W_M^{1/2} \tilde{S}_N(v) \right\|_{\tilde{\mathcal{H}}}^2 \right].$$

Furthermore, $W_M \to \tilde{V}^{-1}$ in the weak operator topology, since for $u, u' \in \tilde{\mathcal{H}}$ we have

$$|\langle u, W_M u' \rangle_{\tilde{\mathcal{H}}} - \langle u, \tilde{V}^{-1} u' \rangle_{\tilde{\mathcal{H}}}| = \left| \left\langle u, \sum_{m=1}^{M} \lambda_m^{-1} \langle u', b_m \rangle_{\tilde{\mathcal{H}}} b_m \right\rangle - \left\langle u, \sum_{m=1}^{D} \lambda_m^{-1} \langle u', b_m \rangle_{\tilde{\mathcal{H}}} b_m \right\rangle \right|$$

$$\leq \|u\|_{\tilde{\mathcal{H}}} \left\| \sum_{m=M+1}^{D} \lambda_m^{-1} \langle u', b_m \rangle_{\tilde{\mathcal{H}}} b_m \right\|_{\tilde{\mathcal{H}}} \to 0,$$

for $M \to D$. It follows by continuity of the operations taken on $W_M$ w.r.t. the weak operator topology that

$$\left\| \left( \mathrm{id}_{\tilde{\mathcal{H}}} + W_M^{1/2} \tilde{V}_N W_M^{1/2} \right)^{-1/2} W_M^{1/2} \tilde{S}_N(v) \right\|_k^2$$

$$\to \left\| \left( \mathrm{id}_{\tilde{\mathcal{H}}} + \tilde{V}^{-1/2} \tilde{V}_N \tilde{V}^{-1/2} \right)^{-1/2} \tilde{V}^{-1/2} \tilde{S}_N(v) \right\|_k^2,$$

as $M \to D$. The same computations as in (30) show that this limit is equal to $\|\tilde{S}_N(v)\|_{(\tilde{V}+\tilde{V}_N)^{-1}}^2$.

Next, we claim that the convergence $\tilde{M}_N W_M \tilde{M}_N^\star \to \tilde{M}_N V^{-1} \tilde{M}_N^\star$ holds in the trace norm topology, and not only in the weak operator topology. Indeed, denoting the trace norm (also called the Frobenius norm) by $\|\cdot\|_1$,

$$\|\tilde{M}_N W_M \tilde{M}_N^\star - \tilde{M}_N \tilde{V}^{-1} \tilde{M}_N^\star\|_1 = \left\| \sum_{m=M+1}^{D} \lambda_m^{-1} (\tilde{M}_N b_m) \otimes (\tilde{M}_N b_m) \right\|_1$$

$$\leq \sum_{m=M+1}^{D} \lambda_m^{-1} \|(\tilde{M}_N b_m) \otimes (\tilde{M}_N b_m)\|_1$$

$$\leq \sup_{m \in [D]} (\lambda_m^{-1}) \cdot \sum_{m=M+1}^{D} \|(\tilde{M}_N b_m) \otimes (\tilde{M}_N b_m)\|_1.$$

Now, since $(\tilde{M}_N b_m) \otimes (\tilde{M}_N b_m)$ is of rank 1, it is clear that $\|(\tilde{M}_N b_m) \otimes (\tilde{M}_N b_m)\|_1 = \|\tilde{M}_N b_m\|^2$. Furthermore, since $\tilde{M}_N \tilde{M}_N^\star$ is a.s. of trace class, the series $\sum_{m=1}^{D} \|\tilde{M}_N u_n\|_{\mathcal{G}^N}^2$ converges for any ONB $(u_m)_{m \in \mathbb{N}_{>0}}$ of $\mathcal{H}$ a.s. (and is equal to $\mathrm{Tr}(\tilde{M}_N \tilde{M}_N^\star)$). This is in particular the case for $(b_m)_m$. Finally, $\sup_{m \in [D]} \lambda_m^{-1} \leq \|\tilde{V}^{-1}\|_{\mathcal{L}_b(\mathcal{H})}$, which is finite since $\tilde{V}^{-1}$ is bounded. As a result,

$$\|\tilde{M}_N W_M \tilde{M}_N^\star - \tilde{M}_N \tilde{V}^{-1} \tilde{M}_N^\star\|_1 \to 0,$$

as $M \to D$, and thus $\tilde{M}_N W_M \tilde{M}_N^\star \to \tilde{M}_N \tilde{V}^{-1} \tilde{M}_N^\star$ in the trace norm. Crucially, the Fredholm determinant is continuous in the trace norm topology, and thus

$$\lim_{M \to D} \det(\mathrm{id}_{\tilde{\mathcal{G}}^N} + \|v\|_{\mathcal{V}}^2 \tilde{M}_N W_M \tilde{M}_N^\star) = \det(\mathrm{id}_{\tilde{\mathcal{G}}^N} + \|v\|_{\mathcal{V}}^2 \tilde{M}_N \tilde{V}^{-1} \tilde{M}_N^\star).$$

Taken together, this shows that for all $N \in \mathbb{N}_{>0}$ and $v \in \mathcal{V}$

$$\bar{G}_N(v) := \lim_{M \to \infty} \int_{\tilde{\mathcal{H}}} G_N(\tilde{h} \otimes v) \mathrm{d}\mathcal{N}(\tilde{h} \mid 0, W_M)$$

exists and

$$\bar{G}_N(v) = \det\left(\mathrm{id}_{\tilde{\mathcal{G}}^N} + \|v\|_{\mathcal{V}}^2 \tilde{M}_N \tilde{V}^{-1} \tilde{M}_N^\star\right)^{-\frac{1}{2}} \exp\left(\frac{1}{2}\|\tilde{S}_N(v)\|_{(\tilde{V} + \|v\|_{\mathcal{V}}^2 \tilde{M}_N^\star \tilde{M}_N)^{-1}}^2\right).$$

By the exact same computations as in the proof of Lemma E.18 (which do not rely on whether or not $\tilde{V}^{-1}$ is trace-class), it follows that

$$\bar{G}_N := \int_{\mathcal{V}} \bar{G}_N(v) \mathrm{d}\mu_Q(v)$$

$$\geq \left[\det\left(\mathrm{id}_{\tilde{\mathcal{G}}^N} + \tilde{M}_N \tilde{V}^{-1} \tilde{M}_N^\star\right)\right]^{-1/2} \exp\left[\frac{1}{2\mathrm{Tr}(Q)}\|\iota_{\mathcal{H}}^{-1} S_N\|_{(\tilde{V} + \tilde{V}_N)^{-1} \otimes Q}^2\right].$$

Furthermore, by construction $\bar{G}_N$ is nonnegative. Finally, we have a.s. from Lemma E.19 that

$$\mathbb{E}[\bar{G}_N \mid \mathcal{F}_{N-1}] = \mathbb{E}\left[\int_{\mathcal{V}} \lim_{M \to \infty} \int_{\tilde{\mathcal{H}}} G_N(\tilde{h} \otimes v) \mathrm{d}\mathcal{N}(\tilde{|0}, W_M) \mathrm{d}\mu_Q(v) \mid \mathcal{F}_{N-1}\right]$$

$$= \int_{\mathcal{V}} \lim_{M \to \infty} \int_{\tilde{\mathcal{H}}} \mathbb{E}\left[G_N(\tilde{h} \otimes v) \mid \mathcal{F}_{N-1}\right] \mathrm{d}\mathcal{N}(\lambda \mid 0, W_M) \mathrm{d}\mu_Q(v)$$

$$\leq \int_{\mathcal{V}} \lim_{M \to \infty} \int_{\tilde{\mathcal{H}}} G_{N-1}(\tilde{h} \otimes v) \mathrm{d}\mathcal{N}(\lambda \mid 0, W_M) \mathrm{d}\mu_Q(v)$$

$$= \bar{G}_{N-1}.$$

This ensures that $(\bar{G}_N)_N$ is still a supermartingale. Since $\bar{G}_0 = 1$, we also have $\mathbb{E}[\bar{G}_N] \leq 1$ for all $N \in \mathbb{N}_{>0}$. We can thus leverage Ville's inequality, and conclude similarly as in the proof of Lemma E.15. $\qquad\square$

Before concluding this section, we provide an important corollary of Theorem E.14, which provides an upper bound on $\|S_N\|_{(\tilde{V} + \tilde{V}_N)^{-1} \otimes \mathrm{id}_{\mathcal{V}}}$ rather than on $\|S_N\|_{(\tilde{V} + \tilde{V}_N)^{-1} \otimes Q}$ at the price of slightly stronger assumptions on the noise.

### E.3.4 Corollary: bound for the noise term of vector-valued least-squares

We now specialize our results to the situation that appears in vector-valued least-squares. The following bound allows to derive a time-uniform high-probability bound for vector-valued least-squares when combined with Lemma E.2, cf. Theorem E.23 below.

**Corollary E.20.** *Under the setup of Section E.1, assume that Assumptions E.5 and E.13 hold. Let $V \in \mathcal{L}(\mathcal{G})$ be self-adjoint, positive-definite, diagonal, and with bounded inverse. Assume that there exists $\rho \in \mathbb{R}_{>0}$ and $R_\mathcal{V} \in \mathcal{L}_\mathrm{b}(\mathcal{V})$ self-adjoint, positive semi-definite, and trace-class, such that $\eta$ is $R$-subgaussian conditionally on $\mathcal{F}$, where $R := \rho^2 \iota_\mathcal{G}(\mathrm{id}_{\tilde{\mathcal{G}}} \otimes R_\mathcal{V}) \iota_\mathcal{G}^{-1}$. Then, for all $\delta \in (0, 1)$,*

$$\mathbb{P}\left[\forall N \in \mathbb{N}_{>0}, \|S_N\|_{(V + V_N)^{-1}} \leq \rho\sqrt{2\mathrm{Tr}(R_\mathcal{V}) \ln\left(\frac{1}{\delta}\left[\det(\mathrm{id}_{\tilde{\mathcal{G}}^N} + \tilde{M}_N \tilde{V}^{-1} \tilde{M}_N^\star)\right]^{1/2}\right)}\right] \geq 1 - \delta,$$

*where $V = \iota_{\tilde{\mathcal{H}}}(\tilde{V} \otimes \mathrm{id}_\mathcal{V}) \iota_{\tilde{\mathcal{H}}}^{-1}$.*

Before proving this result, let us briefly comment on the subgaussianity assumption used.

*Remark* E.21. The subgaussianity assumption of Corollary E.20 is in general a strengthening of the assumption that $\eta$ is $\bar{\rho}$-subgaussian for $\bar{\rho} \in \mathbb{R}_{>0}$, but also a weakening of the one that $\eta$ is $\bar{R}$-subgaussian for $\bar{R} \in \mathcal{L}_\mathrm{b}(\mathcal{G})$ self-adjoint, positive semi-definite, and of trace class. Indeed, the operator $R$ given in the corollary is not trace-class when $\tilde{\mathcal{G}}$ is infinite-dimensional. Furthermore, for any such operator $\bar{R}$, there exist $\rho \in \mathbb{R}_{>0}$ and $Q \in \mathcal{L}_\mathrm{b}(\mathcal{V})$ self-adjoint, positive semi-definite, and of trace class such that $\bar{R} \preceq R := \rho^2 \iota_\mathcal{G}(\mathrm{id}_{\tilde{\mathcal{G}}} \otimes Q) \iota_\mathcal{G}^{-1}$. Indeed, define the trace-class operator $\bar{R}' = \iota_\mathcal{G}^{-1} \bar{R} \iota_\mathcal{G} \in \mathcal{L}_\mathrm{b}(\tilde{\mathcal{H}} \otimes \mathcal{V})$, and take $\rho = (\|\mathrm{Tr}_\mathcal{V}(\bar{R}')\|_{\mathcal{L}_\mathrm{b}(\tilde{\mathcal{H}})})^{1/2}$, and $Q = \mathrm{Tr}_{\tilde{\mathcal{H}}}(\bar{R}')$, where $\mathrm{Tr}_W(A)$ is the partial trace over $W$ of a trace-class operator $A$ defined on a tensor-product space involving $W$. We forego proving this construction for brevity.

The proof of Corollary E.20 relies on the following lemma.

**Lemma E.22.** *Let $R \in \mathcal{L}_{\mathrm{b}}(\mathcal{G})$ be self-adjoint, positive semi-definite. Let $\eta$ be a $\mathcal{G}$-valued, zero-mean random variable, and assume that $\eta$ is $R$-subgaussian. Then, $\eta$ takes values in $\mathrm{cl}(\mathrm{ran}(R))$ a.s.*

*Proof.* Let $e \in \mathrm{ran}(R)^{\perp}$. The variable $X := \langle e, \eta \rangle_{\mathcal{G}}$ is a real-valued, $R_e := \|e\|_R$-subgaussian random variable, as $\mathbb{E}[X] = \langle e, \mathbb{E}[\eta] \rangle = 0$ and

$$\mathbb{E}[\exp(cX)] = \mathbb{E}[\langle ce, \eta \rangle] \leq \exp\left[\frac{1}{2}\langle R(ce), ce \rangle_{\mathcal{G}}\right] = \exp\left[\frac{1}{2}c^2 R_e^2\right], \ \forall c \in \mathbb{R}.$$

Classical results on scalar subgaussian variables then guarantee that $\mathrm{Var}[X] \leq R_e^2$. Yet, $e \in \mathrm{ran}(R)^{\perp} = \ker(R^{\star}) = \ker(R)$, since $R$ is self-adjoint, and thus $Re = 0$. As a result, $R_e = \|e\|_R = (\langle Re, e \rangle)^{1/2} = 0$, and thus $\mathrm{Var}[X] = 0$ and $X$ is constant a.s., and this constant is $0$ since $X$ is $0$-mean. We deduce by separability of $\mathcal{G}$ and continuity of the scalar product that $\mathbb{P}[\exists e \in \mathrm{ran}(R)^{\perp}, \ \langle \eta, e \rangle \neq 0] = 0$, which shows that it holds with full probability that $\eta \in (\mathrm{ran}(R)^{\perp})^{\perp} = \mathrm{cl}(\mathrm{ran}(R))$, concluding the proof. $\qquad\square$

*Proof of Corollary E.20.* Under the current assumptions, the scaled noise $\bar{\eta}_n = (R^{\sharp})^{1/2}\eta_n$ is $1$-subgaussian conditionally on $\mathcal{F}$, where $R^{\sharp}$ is the Moore-Penrose pseudo-inverse of $R$. Indeed, it is (conditionally) $0$-mean and for any $g \in \mathcal{G}$,

$$\mathbb{E}[\exp[\langle g, (R^{\sharp})^{1/2}\eta_n \rangle] \mid \mathcal{F}_{n-1}] = \mathbb{E}[\exp[\langle (R^{\sharp})^{1/2}g, \eta_n \rangle]\mathcal{F}_{n-1}]$$
$$\leq \exp\left[\frac{1}{2}\langle (R^{\sharp})^{1/2}g, R(R^{\sharp})^{1/2}g \rangle\right]$$
$$= \exp\left[\frac{1}{2}\|g\|_{RR^{\sharp}}^2\right]$$
$$\leq \exp\left[\frac{1}{2}\|g\|_{\mathcal{G}}^2\right].$$

The second equality comes from the fact that $R^{1/2}$ and $R^{\sharp}$ commute; the proof is a technical exercise that follows from the spectral theorem applied to $R_{\mathcal{V}}$ and the spectral definition of $R^{\sharp}$. We also leveraged in the last inequality the fact that $RR^{\sharp}$ is an orthogonal projection, and thus $\|g\|_{RR^{\sharp}} \leq \|g\|_{\mathcal{G}}$; see for instance Proposition 2.3 in Engl et al. [45]. Theorem E.14 applied with $Q = R_{\mathcal{V}}$ then guarantees that for all $\delta \in (0, 1)$, it holds with probability at least $1 - \delta$ that, for all $N \in \mathbb{N}_{>0}$,

$$\|\iota_{\mathcal{H}}^{-1} M_N^{\star}(R_N^{\sharp})^{1/2}\eta_{:N}\|_{(\tilde{V}+\tilde{V}_N)^{-1} \otimes R_{\mathcal{V}}} \leq \sqrt{2\mathrm{Tr}(R_{\mathcal{V}})\ln\left(\frac{1}{\delta}\left[\det(\mathrm{id}_{\tilde{\mathcal{G}}^N} + \tilde{M}_N^{\star}\tilde{V}^{-1}\tilde{M}_N)\right]^{1/2}\right)}.$$

where we introduced $R_N^{\sharp} : (g_n)_{n \in [N]} \in \mathcal{G}^N \mapsto (R^{\sharp}g_n)_{n \in [N]} \in \mathcal{G}^N$. We conclude by showing that

$$\|\iota_{\mathcal{H}}^{-1} M_N^{\star}(R_N^{\sharp})^{1/2}\eta_{:N}\|_{(\tilde{V}+\tilde{V}_N)^{-1} \otimes R_{\mathcal{V}}} = \rho^{-1}\|M_N^{\star}\eta_{:N}\|_{(V+V_N)^{-1}}, \text{ a.s.}$$

Indeed,

$$M_N^{\star}(R_N^{\sharp})^{1/2}\eta_{:N} = \sum_{n=1}^{N} L_n^{\star}(R^{\sharp})^{1/2}\eta_n$$
$$= \rho^{-1}\sum_{n=1}^{N} L_n^{\star}\iota_{\mathcal{G}}\left(\mathrm{id}_{\tilde{\mathcal{G}}} \otimes (R_{\mathcal{V}}^{\sharp})^{1/2}\right)\iota_{\mathcal{G}}^{-1}\eta_n$$
$$= \rho^{-1}\sum_{n=1}^{N} \iota_{\mathcal{H}}\left(\tilde{L}_n \otimes \mathrm{id}_{\tilde{\mathcal{G}}}\right)\iota_{\mathcal{G}}^{-1}\iota_{\mathcal{G}}\left(\mathrm{id}_{\tilde{\mathcal{G}}} \otimes (R_{\mathcal{V}}^{\sharp})^{1/2}\right)\iota_{\mathcal{G}}^{-1}\eta_n$$
$$= \rho^{-1}\iota_{\mathcal{H}}\sum_{n=1}^{N}\left(\tilde{L}_n \otimes (R_{\mathcal{V}}^{\sharp})^{1/2}\right)\iota_{\mathcal{G}}^{-1}\eta_n,$$

and thus

$$\|\iota_{\tilde{\mathcal{H}}}^{-1} M_N^\star (R_N^\sharp)^{1/2} \eta_{:N}\|_{(\tilde{V}+\tilde{V}_N)^{-1}\otimes R}^2$$

$$= \rho^{-2} \sum_{n,m=1}^N \left\langle \left( \left[ (\tilde{V}+\tilde{V}_N)^{-1} \tilde{L}_n^\star \right] \otimes \left[ R_{\mathcal{V}} (R_{\mathcal{V}}^\sharp)^{1/2} \right] \right) \iota_{\mathcal{G}}^{-1} \eta_n, \left( \tilde{L}_m^\star \otimes (R_{\mathcal{V}}^\sharp)^{1/2} \right) \iota_{\mathcal{G}}^{-1} \eta_m \right\rangle_{\tilde{\mathcal{H}}\otimes\mathcal{V}}$$

$$= \rho^{-2} \sum_{n,m=1}^N \left\langle \left( \left[ (\tilde{V}+\tilde{V}_N)^{-1} \tilde{L}_n^\star \right] \otimes \left[ (R_{\mathcal{V}}^\sharp)^{1/2} R_{\mathcal{V}} (R_{\mathcal{V}}^\sharp)^{1/2} \right] \right) \iota_{\mathcal{G}}^{-1} \eta_n, (\tilde{L}_m^\star \otimes \mathrm{id}_{\mathcal{V}}) \iota_{\mathcal{G}}^{-1} \eta_m \right\rangle_{\tilde{\mathcal{H}}\otimes\mathcal{V}}$$

$$= \rho^{-2} \sum_{n,m=1}^N \left\langle \left[ (\tilde{V}+\tilde{V}_N)^{-1} \otimes \mathrm{id}_{\mathcal{V}} \right] \left[ \tilde{L}_n^\star \otimes \mathrm{id}_{\mathcal{V}} \right] \left[ \mathrm{id}_{\tilde{\mathcal{G}}} \otimes R_{\mathcal{V}} \right] \left[ \mathrm{id}_{\tilde{\mathcal{G}}} \otimes R_{\mathcal{V}} \right]^\sharp \iota_{\mathcal{G}}^{-1} \eta_n, \right.$$

$$\left. (\tilde{L}_m^\star \otimes \mathrm{id}_{\mathcal{V}}) \iota_{\mathcal{G}}^{-1} \eta_m \right\rangle_{\tilde{\mathcal{H}}\otimes\mathcal{V}}$$

$$= \rho^{-2} \sum_{n,m=1}^N \left\langle \iota_{\tilde{\mathcal{H}}}^{-1} (V+V_N)^{-1} L_n^\star R R^\sharp \eta_n, \iota_{\tilde{\mathcal{H}}}^{-1} L_m^\star \eta_m \right\rangle_{\tilde{\mathcal{H}}\otimes\mathcal{V}}$$

$$= \rho^{-2} \sum_{n,m=1}^N \left\langle (V+V_N)^{-1} L_n^\star R R^\sharp \eta_n, L_m^\star \eta_m \right\rangle_{\mathcal{H}},$$

where we leveraged in the third equality the fact that $(R_{\mathcal{V}}^\sharp)^{1/2}$ and $R_{\mathcal{V}}$ commute, and that $(\tilde{V}+\tilde{V}_N)^{-1} \otimes \mathrm{id}_{\mathcal{V}} = \iota_{\tilde{\mathcal{H}}}^{-1} (V+V_N)^{-1} \iota_{\mathcal{H}}$ in the fourth one. Next, $RR^\sharp$ is the orthogonal projection on $\mathrm{cl}(\mathrm{ran}(R))$; see for instance Proposition 2.3 in [45]. Yet, $\eta_n$ is a.s. in $\mathrm{cl}(\mathrm{ran}(R))$ by Lemma E.22. It follows that $RR^\sharp \eta_n = \eta_n$, a.s., for all $n \in \mathbb{N}_{>0}$. As a result, it holds a.s. that

$$\|M_N^\star (R^\sharp)^{1/2} \eta_{:N}\|_{(\tilde{V}+\tilde{V}_N)^{-1}\otimes R_{\mathcal{V}}}^2 = \rho^{-2} \sum_{n,m=1}^N \left\langle (V+V_N)^{-1} L_n^\star \eta_n, L_m^\star \eta_m \right\rangle_{\mathcal{H}}$$

$$= \rho^{-2} \|M_N^\star \eta_{:N}\|_{(V+V_N)^{-1}}^2.$$

The result follows. $\qquad\square$

## E.4  Main result

We now combine all that precedes to obtain the announced concentration bound in vector-valued least-squares.

**Theorem E.23.** *Under the notations of Section E.1, let $R \in \mathcal{L}_{\mathrm{b}}(\mathcal{G})$ be self-adjoint, positive semi-definite, and $S \in \mathbb{R}_{>0}$. Assume that $\|h^\star\|_{\mathcal{H}} \leq S$, and assume one of the following and define $\beta_\lambda$ and $\mathcal{N}$ accordingly:*

1. *Assumptions E.5 and E.13 hold, $R = \rho^2 \iota_{\mathcal{G}} (\mathrm{id}_{\tilde{\mathcal{G}}} \otimes R_{\mathcal{V}}) \iota_{\mathcal{G}}^{-1}$ for some $\rho \in \mathbb{R}_{>0}$ and $R_{\mathcal{V}} \in \mathcal{L}_{\mathrm{b}}(\mathcal{V})$ self-adjoint, positive semi-definite, and trace-class, and $\eta$ is R-subgaussian conditionally on $\mathcal{F}$. Then, define $\mathcal{N} = \mathbb{N}_{>0}$ and, for all $(n,\delta) \in \mathbb{N}_{>0} \times (0,1)$,*

$$\beta_\lambda(N,\delta) = S + \frac{\rho}{\sqrt{\lambda}} \sqrt{2\mathrm{Tr}(R_{\mathcal{V}}) \ln\left[ \frac{1}{\delta} \left\{ \det\left( \mathrm{id}_{\tilde{\mathcal{G}}^N} + \frac{1}{\lambda} \tilde{M}_N \tilde{M}_N^\star \right) \right\}^{1/2} \right]},$$

*where $\tilde{M}_N$ is introduced in (27).*

2. *Assumption E.3 holds, $R$ is of trace class, and $\eta$ is R-subgaussian. Then, define $\mathcal{N} = \{N_0\}$ for some $N_0 \in \mathbb{N}_{>0}$ and, for all $(n,\delta) \in \mathbb{N}_{>0} \times (0,1)$,*

$$\beta_\lambda(N,\delta) = S + \frac{1}{\sqrt{\lambda}} \sqrt{\mathrm{Tr}(T_{N,\lambda}) + 2\sqrt{\ln\left(\frac{1}{\delta}\right)} \|T_{N,\lambda}\|_2 + 2\ln\left(\frac{1}{\delta}\right) \|T_{N,\lambda}\|_{\mathcal{L}_{\mathrm{b}}(\mathcal{H})}},$$

*where we introduced $T_{N,\lambda} = (V_N + \lambda\mathrm{id}_{\mathcal{H}})^{-1/2} M_N^\star (R \otimes \mathrm{id}_{\mathbb{R}^N}) M_N (V_N + \lambda\mathrm{id}_{\mathcal{H}})^{-1/2}$.*

*In both cases, for all $\delta \in (0,1)$,*

$$\mathbb{P}\left[\forall N \in \mathcal{N}, \forall \bar{L} \in \mathcal{L}_{\mathrm{b}}(\mathcal{G}, \bar{\mathcal{G}}), \|\bar{L}h_{N,\lambda} - \bar{L}h^\star\|_{\bar{\mathcal{G}}} \leq \beta_\lambda(N, \delta) \cdot \sigma_{N,\lambda}(\bar{L})\right] \geq 1 - \delta. \qquad (31)$$

*Proof.* Let $\delta \in (0,1)$. In case 2, it follows from Lemma E.4 that it holds with probability at least $1 - \delta$ that

$$\|S_{N_0}\|_{(V+V_N)^{-1}} = \|(V + V_{N_0})^{-1/2} M_{N_0}^\star \eta_{:N_0}\|$$

$$\leq \sqrt{\mathrm{Tr}(T_{N_0,\lambda}) + 2\sqrt{\ln\left(\frac{1}{\delta}\right)} \|T_{N_0,\lambda}\|_2 + 2\ln\left(\frac{1}{\delta}\right) \|T_{N_0,\lambda}\|_{\mathcal{L}_{\mathrm{b}}(\mathcal{H})}}$$

$$=: \gamma_{N_0,\lambda}(\delta),$$

where we applied the lemma with $A = (\mathrm{id}_{\mathcal{H}} + V_{N_0})^{-1/2} M_{N_0}^\star$, yielding $T = T_{N_0,\lambda} = A \cdot (R \otimes \mathrm{id}_{\mathbb{R}^{N_0}}) A^\star$ and. In case 1, it follows from Corollary E.20 that it holds with probability at least $1 - \delta$ that, for all $N \in \mathbb{N}_{>0}$,

$$\|S_N\|_{(V+V_N)^{-1}} \leq \rho \sqrt{2\mathrm{Tr}(R_\mathcal{V}) \left\{\ln\left[\frac{1}{\delta} \det\left(\mathrm{id}_{\tilde{\mathcal{G}}^N} + \frac{1}{\lambda}\tilde{M}_N \tilde{M}_N^\star\right)\right]\right\}^{1/2}} =: \gamma_{N,\lambda}(\delta)$$

In both cases, it holds with probability at least $1 - \delta$ that

$$\forall N \in \mathcal{N}, \|S_N\|_{(V+V_N)^{-1}} \leq \gamma_{N,\lambda}(\delta).$$

The result then immediately follows from Lemma E.2, (20), and (22), noting that $\beta_\lambda(N, \delta) = S + \frac{1}{\sqrt{\lambda}}\gamma_{N,\lambda}(\delta)$. □

## E.5 Proof of Theorem 4.3

*Proof.* The result follows immediately by applying Theorem E.23 with $\mathcal{H} = \mathcal{H}_K$, $L_n = K(\cdot, X_n)^\star$, and $\eta_n = y_n - \mathbb{E}[p](X_n)$, for all $n \in \mathbb{N}_{>0}$. Indeed, for these choices, the optimization problem (15) is equivalent to that in (4), since for all $h \in \mathcal{H}_K$, $g \in \mathcal{G}$, and $n \in \mathbb{N}_{>0}$

$$\langle h(X_n), g \rangle_{\mathcal{G}} = \langle h, K(\cdot, X_n)g \rangle_K = \langle K(\cdot, X_n)^\star h, g \rangle_{\mathcal{G}},$$

showing that the evaluation operator in $X_n$ is $K(\cdot, X_n)^\star$. Furthermore, the process $\eta$ satisfies the appropriate assumption E.3 or E.5 depending on the considered case, since $D$ is assumed to be a process of transition pairs of $p$, or a process of independent transition pairs of $p$. In case 2, the assumptions of case 2 in Theorem E.23 are satisfied, and the result follows. Similarly, In case 1, the assumptions of case 1 in Theorem E.23 are satisfied by Proposition E.12, and the result follows as well. □

## F    Proof of Theorem 4.2

*Proof.* For any $p_1, p_2 \in \Theta$, we have

$$\eta_{\mathrm{I}}(p_1, p_2, \mathcal{S}, \mathcal{N}_1, \mathcal{N}_2)$$

$$= \mathbb{P}[\exists (n_1, n_2) \in \mathcal{N}_1 \times \mathcal{N}_2 : \mathcal{S} \cap E_{\mathrm{I}}(p_1, p_2, D^{(1)}_{p_1,:n_1}, D^{(2)}_{p_2,:n_2}) \neq \emptyset]$$

$$= \mathbb{P}\left[\exists (n_1, n_2, x) \in \mathcal{N}_1 \times \mathcal{N}_2 \times \mathcal{S} : \mathbb{E}(p_1)(x) = \mathbb{E}(p_2)(x)\right.$$

$$\left. \wedge \left\| f^{(1)}_{D_{:n_1}^{(1)}}(x) - f^{(2)}_{D_{:n_2}^{(2)}}(x) \right\| > \sum_{i=1}^2 B_i\left(D^{(i)}_{:n}, x\right) \right]$$

$$\leq \mathbb{P}\left[\exists (n_1, n_2, x) \in \mathcal{N}_1 \times \mathcal{N}_2 \times \mathcal{S} : \sum_{i=1}^2 \left\| f^{(i)}_{D_{:n_i}^{(i)}}(x) - \mathbb{E}(p_i)(x) \right\| > \sum_{i=1}^2 B_i\left(D^{(i)}_{:n}, x\right) \right]$$

$$\leq \mathbb{P}\left[\bigcup_{i=1}^2 \left\{\exists (n, x) \in \mathcal{N}_i \times \mathcal{S} \left\| f^{(i)}_{D_{:n}^{(i)}}(x) - \mathbb{E}(p_i)(x) \right\| > B_i\left(D^{(i)}_{:n}, x\right)\right\} \right]$$

$$\leq \sum_{i=1}^2 \mathbb{P}\left[\exists (n, x) \in \mathcal{N}_i \times \mathcal{S} : \left\| f^{(i)}_{D_{:n}^{(i)}}(x) - \mathbb{E}(p_i)(x) \right\| > B_i\left(D^{(i)}_{:n}, x\right) \right]$$

$$\leq \delta_1 + \delta_2$$

where we used the fact that $\mathbb{E}(p_1)(x) = \mathbb{E}(p_2)(x)$ for any $x \in E_{\mathrm{I}}(p_1, p_2, D_1, D_2)$ and $(D_1, D_2) \in \mathcal{D}^2$. $\qquad\square$

# G Experiment details

This section presents the detailed setups for each of our numerical experiments. Additionally, the code to reproduce all experimental results is available at https://github.com/Data-Science-in-Mechanical-Engineering/conditional-test.

Unless stated, all our experiments with bootstrapped test thresholds use the naive resampling scheme outlined in Appendix B.

## G.1 Illustrative example (Figure 1)

Figure 1 illustrates our test on a simple example with one-dimensional inputs and outputs. We use a Gaussian kernel on $\mathcal{X} = [-1, 1]$ and the inhomogeneous linear kernel on $\mathcal{Z} = \mathbb{R}$. We pick two mean functions $f_1, f_2 \in \mathcal{H}_k$, from which we collect data sets $D_i = \{(x_j^{(i)}, f_i(x_j^{(i)}) + \epsilon_j^{(i)})\}_{j=1}^n$, $i \in \{1, 2\}$. The covariates $x_j^{(i)}$ are sampled uniformly from $\mathcal{X}$ and $\epsilon_j^{(i)} \sim \mathcal{N}(\cdot \mid 0, s^2)$, such that the Markov kernel corresponding to each data set is $p_i(\cdot, x) = \mathcal{N}(\cdot \mid f_i(x), s^2)$, where $\mathcal{N}(\cdot \mid \mu, s^2)$ is the Gaussian measure on $\mathbb{R}$ with mean $\mu$ and variance $s^2$. Consequently, $H_0(x, p_1, p_2)$ is equivalent to $f_1(x) = f_2(x)$.

We apply our test with analytical thresholds (Figure 1, left) and bootstrapped thresholds (Figure 1, right). For the analytical thresholds, we use the ground truth RKHS function norm (that is, 1) and the Gaussian noise standard deviation for the corresponding upper bounds on these quantities in Theorem 4.3.

Table 2 reports the hyperparameters for this experiment.

## G.2 Empirical error rates

This section provides implementation and design details for experiments evaluating the empirical type I and type II errors of our tests in controlled, well-specified settings. All experiments in this section were conducted on an Intel Xeon 8468 Sapphire CPU, using 10 GB of RAM.

### G.2.1 General setup

**Data generation** We evaluate our test by generating two data sets, $D_1$ and $D_2$, each containing $n \in \mathbb{N}$ transition pairs of the form

$$D_i = \{(x_j^{(i)}, f_i(x_j^{(i)}) + \epsilon_j^{(i)})\}_{j=1}^n, \quad i \in \{1, 2\}.$$

Here, the inputs $x_j^{(i)}$ are sampled (uniformly, unless stated otherwise) from $\mathcal{X}$, $\epsilon_j^{(i)}$ are i.i.d. noise, and $f_i$ is a random unit-norm element of $\mathcal{H}_k$ whose sampling is detailed below. In particular, we choose $f_i$ directly from the RKHS $\mathcal{H}_k$ of $k$. In all our experiments in this section, we use a Gaussian kernel

$$k(x, y) = \exp\left(-\frac{\|x - y\|_{\mathcal{X}}^2}{2\gamma^2}\right) \tag{32}$$

on $\mathcal{X} \subset \mathbb{R}^2$ with bandwidth $\gamma^2 = 0.25$.

*Remark G.1.* In the case where $\kappa$ is the linear kernel, the procedure above ensures that $\mathbb{E}[p_i] \in \mathcal{H}_K$ with $K = k \cdot \mathrm{id}_{\mathcal{H}_\kappa}$. This guarantee is lost when $\kappa$ is a more complex kernel. While this means that the simulations may use the test in contexts not covered by the theory, this is precisely the interest of the bootstrapping schemes we propose: the test can still be applied, and its performance evaluated empirically.

**Sampling of mean functions** Following Fiedler et al. [25], we sample mean functions $f_i \in \mathcal{H}_k$ randomly by first sampling $m$ points $x_1, ..., x_m$ uniformly at random from $\mathcal{X}$ and then sampling $f_i$ uniformly from the sphere

$$\left\{f = \sum_{j=1}^m \alpha_j k(\cdot, x_j) \mid \alpha_1, ..., \alpha_m \in \mathbb{R}, \|f\|_{\mathcal{H}_k} = r\right\} \subseteq \mathcal{H}_k, \tag{33}$$

where the RKHS norm $r$ depends on the concrete experiment setting. When reporting hyperparameters, we refer to $m$ as the "mean function dimension".

**Empirical error rates** We quantify both the type I and type II error rates through the test's empirical positive rate — the proportion of data set pairs on which at least one covariate triggers rejection in the region of interest — under two data-generation regimes. Under the global null ($H_0(x, p_1, p_2)$ holds for all $x \in \mathcal{X}$), this positive rate directly estimates the type I error. Under an alternative ($H_0(x, p_1, p_2)$ fails for some $x \in \mathcal{X}$), the same quantity measures the test's power, from which we compute the type II error as one minus that power.

To compute the positive rate in practice, we draw $T$ independent data set pairs $\{(D_1^{(j)}, D_2^{(j)})\}_{j=1}^T$. For each pair, we take the observed covariates

$$\mathcal{S}^{(j)} = \{x \in \mathcal{X} \mid \exists z \in \mathcal{Z}, (x, z) \in D_1^{(j)} \cup D_2^{(j)}\} \tag{34}$$

as the region of interest and compute the covariate rejection region

$$\begin{aligned} \hat{\chi}(D_1^{(j)}, D_2^{(j)}) &\coloneqq \chi(D_1^{(j)}, D_2^{(j)}) \cap \mathcal{S}^{(j)} \\ &= \{x \in \mathcal{S} \mid \mathcal{T}(x, D_1 \cup D_2) = 1\}. \end{aligned} \tag{35}$$

We then take

$$\frac{1}{T} \sum_{j=1}^T \begin{cases} 0 & \text{if } \hat{\chi}(D_1^{(j)}, D_2^{(j)}) = \emptyset \\ 1 & \text{otherwise} \end{cases} \tag{36}$$

as the empirical positive rate.

Repeating this under the null and alternative regimes while varying the significance level $\alpha$ and other parameters, we obtain detailed error-rate curves that reveal how closely the observed type I error matches its nominal level and how the test power depends on the various parameters. Enforcing either regime requires appropriate choices of the mean functions $f_1$ and $f_2$, and noise distributions. For instance, to enforce the global null, it is sufficient (but depending on the kernel $\kappa$, not strictly necessary) to generate both data sets from identical mean functions $f_1 = f_2$ with identical noise distribution. For the alternative, we can introduce a controlled discrepancy, such as choosing $f_2 = f_1 + h$ for some function $h \in \mathcal{H}_k$, or alter the noise distribution in the case where we introduce a kernel $\kappa$ to test for other properties such as higher moments. Because the exact construction for enforcing null and alternative hypotheses depends on the specific experiment setting, we defer those specific design details to the individual experiment descriptions that follow.

Finally, we always compute positive rates from $T = 100$ independently sampled data sets and report curves that are averaged over 100 independent experiment runs (i.e., samples of mean function pairs), together with 2.5% and 97.5% quantiles taken over the experiment runs.

### G.2.2 Global and local sensitivity (Figure 2)

Figure 2 presents the empirical positive rate of our test in three different scenarios: (1) when the $H_0$ is enforced enforced everywhere on $\mathcal{X}$ (Figure 2, left); (2) when $H_0$ is violated at frequently sampled covariates (Figure 2, middle); and (3) when $H_0$ is violated at rarely sampled covariates (Figure 2, right). In all three scenarios, we use the inhomogeneous linear kernel for $\kappa$ and Gaussian noise $\mathcal{N}(0, s^2)$ such that $H_0(x)$ holds if and only if $f_1(x) = f_2(x)$. In the following, we describe how each scenario enforces its corresponding hypothesis.

**Scenario 1: Global null** We pick a unit-norm function $f_1 \in \mathcal{H}_k$ and set $f_2 = f_1$. Consequently, $p_1 = p_2$ such that every rejection is a false positive, and the empirical positive rate estimates our type I error.

**Scenario 2: Frequent violations** We pick a unit-norm function $f_1 \in \mathcal{H}_k$, and independently pick a perturbation $h \in \mathcal{H}_k$ with norm $\xi$. By setting $f_2 = f_1 + h$, $f_1$ and $f_2$ differ almost everywhere almost surely, thereby creating a global discrepancy between $p_1$ and $p_2$ whose magnitude is controlled by $\xi$. In this setting, a rejection is taken to be a true positive, neglecting the edge case where $h(x) = 0$ incidentally causes $f_1(x) = f_2(x)$. We then take the empirical positive rate as the true positive rate, translating directly to our type II error.

**Scenario 3: Rare violations** We pick a unit-norm function $f_1 \in \mathcal{H}_k$ and set $f_2 = f_1 + k(\cdot, 0)$. Then, $p_1(\cdot, x)$ and $p_2(\cdot, x)$ are approximately equal for $x \in \mathcal{X}_{\text{same}} \coloneqq \{x \in \mathcal{X} \mid k(\cdot, x) < 10^{-2}\}$, but

differ significantly for $x \in \mathcal{X}_{\text{diff}} = \mathcal{X} \setminus \mathcal{X}_{\text{same}}$. The value $10^{-2}$ was chosen as an order of magnitude at which it is unrealistic to expect detecting the difference from the considered data set sizes. To control how often this anomalous region $\mathcal{X}_{\text{diff}}$ is sampled, we now sample covariates from a mixture; namely from $\mathcal{X}_{\text{diff}}$ with probability $\theta$, and from $\mathcal{X}_{\text{same}}$ with probability $1 - \theta$. Varying $\theta$ then controls the difficulty of detecting a violation, and the empirical positive rate again translates to our type II error under this more challenging alternative.

**Baseline comparison** We compare our test to the conditional two-sample procedure of Hu and Lei [5], which casts hypothesis testing as a weighted conformal-prediction problem and builds a rank-sum statistic from estimated density ratios. In their framework, one classifier is trained on covariates alone to learn their marginal density ratios, and a second on the joint covariate–response pairs to learn the conditional density ratios. Because the baseline's performance hinges on the accuracy of the density ratio learning method, we provide it with oracle-style information in each scenario. Firstly, we inform the method in all scenarios that the marginal densities are identical between the two samples. Secondly, in Scenarios 1 and 2, we supply the true Gaussian noise distribution and restrict the conditional density-ratio model to the Gaussian parametric family with known variance, so the learner only estimates conditional means via KRR (as in our test). In Scenario 3, we go one step further and provide the baseline with the ground truth conditional density ratios. This alignment makes outcomes directly comparable and focuses the comparison on the heart of each test, namely, global testing using one aggregated statistic versus our local, covariate-specific decisions.

Table 3 reports the hyperparameters for Scenarios 1 and 2, and Table 4 those for Scenario 3. The experiments used 50 core hours.

### G.2.3 Comparison of higher-order moments (Figure 4)

Figure 4 shows the positive rate of our test for different kernels $\kappa$ on $\mathcal{Z} = \mathbb{R}$ when the conditional means coincide, but the conditional distributions still differ. We pick a unit-norm mean function $f$ and set $f_1 = f_2 = f$, such that both data sets $D_1$ and $D_2$ in each pair are generated from the same mean function. However, $D_1$ and $D_2$ differ in the noise applied to the observations. The observations in $D_1$ are corrupted with Gaussian noise $\mathcal{N}(0, s^2)$. In contrast, the observations in $D_2$ are corrupted with noise sampled from an equally weighted mixture of $\mathcal{N}(-\mu, s^2)$ and $\mathcal{N}(\mu, s^2)$. For this data-generating process, the conditional *means* coincide while the conditional *distributions* differ. We compute the empirical positive rates in this setting for (i) a linear kernel; and (ii) a Gaussian kernel on $\mathcal{Z}$.

Table 6 reports the hyperparameters for this experiment. The experiment used 20 core hours.

### G.2.4 Influence of using an overly rich kernel (Figure 5)

Figure 5 shows the type I and II errors of our test for different Gaussian and polynomial kernels $\kappa$ on $\mathcal{Z} = \mathbb{R}$. For the type I error, we pick a single unit-norm mean function $f = f_1 = f_2$ and report the empirical positive rate. For the type II error, we pick two mean functions $f_1$ and $f_2$ independently. In either case, we apply the same Gaussian noise $\mathcal{N}(0, s^2)$ to both data sets.

Table 7 reports the hyperparameters for this experiment. The experiment used 15 core hours.

### G.2.5 Bootstrapping schemes (Figures 6 and 7)

Figure 6 shows the type I and II errors of our test for different bootstrapping schemes when varying the data set size and regularization of the KRR. For the type I error, we pick a single unit-norm mean function $f = f_1 = f_2$ and report the empirical positive rate. For the type II error, we pick two mean functions $f_1$ and $f_2$ independently. We use a linear kernel $\kappa$ on $\mathcal{Z}$ and apply the same Gaussian noise $\mathcal{N}(0, s^2)$ to both data sets. In Figure 6a, we fix the regularization $\lambda = 0.5$ and vary the data set size $n$. In Figure 6b, we fix $n = 50$ and vary $\lambda$.

Figure 7 reports type I and II errors for analytical and bootstrapped test thresholds, using the same setup as before but fixing both $n = 100$ and $\lambda = 0.25$.

Tables 8 and 9 report the hyperparameters for these experiments. The experiments used 30 core hours.

Table 2: Hyperparameters used in generating Figure 1.

| Parameter | Value |
| --- | --- |
| Input set | $\mathcal{X} = [-1, 1]$ |
| Input kernel bandwidth | $\gamma^2 = 0.25$ |
| Data set size | $n = 25$ |
| Noise variance | $s^2 = 0.05^2$ |
| Regularization | $\lambda = 0.01$ |
| Bootstrap resamples | $M = 1000$ |

Table 3: Hyperparameters used in generating Figure 2 (left and middle).

| Parameter | Value |
| --- | --- |
| Input set | $\mathcal{X} = [-1, 1]^2$ |
| Mean function dimension | $m = 12$ |
| Data set size | $n = 100$ |
| Noise variance | $s^2 = 0.1^2$ |
| Regularization | $\lambda = 0.1$ |
| Bootstrap resamples | $M = 500$ |

### G.3 Process monitoring (Figure 3)

Figure 3 shows the ratio between the test statistic and threshold averaged over the sliding window following the trajectory of a perturbed linear dynamical system (left), and the average value of $\sigma_{D,\lambda}(x)$ over the window when $D$ is the reference data set. We detail this experiment here.

We set $\mathcal{X} = \mathbb{R}^d$, $d \in \mathbb{N}_{>0}$, and choose the linear operator-valued kernel $K(x, x') = \langle x, x' \rangle_{\mathbb{R}^d} \cdot \mathrm{id}_{\mathbb{R}^d}$, as we wish to compare linear functions mapping $\mathbb{R}^d$ to itself. We randomly sample a matrix $A \in \mathrm{O}(d)$, the group of orthonormal matrices in dimension $d$, pick a fixed initial state $X_0 = \frac{1}{\sqrt{d}}[1 \cdots 1]^\top \in \mathbb{R}^d$, and generate a single trajectory $(X_n)_{n \in [N]}$ as $X_{n+1} = AX_n + \epsilon_n$, where $\epsilon_n \sim \mathcal{N}(0, s^2 I_d)$ is i.i.d. Gaussian noise. We enforce that $A$ is orthonormal so the trajectory neither shrinks nor explodes exponentially. Then, the observations $Y = (Y_n)_{n \in \mathbb{N}_{>0}}$ are taken as $Y_n = X_{n+1}$. It is immediate to verify that $((X_n, Y_n))_{n \in \mathbb{N}_{>0}}$ is a process of transition pairs of the Markov kernel $p(\cdot, x) = \mathcal{N}(\cdot \mid Ax, s^2 I_d)$, where $\mathcal{N}(\cdot \mid \mu, s^2 I_d)$ is the Gaussian measure on $\mathbb{R}^d$ with mean $\mu \in \mathbb{R}^d$ and covariance matrix $s^2 I_d \in \mathbb{R}^{d \times d}$.

Using this procedure for 5 independent system trajectories of length 400, we collect a data set of $n_{\mathrm{ref}} = 2000$ data points, which we call the reference data set $D$. We then follow the trajectory with a sliding window of size $n_{\mathrm{win}} = 50$ data points, and perform the test at every step on the points in the sliding window. After 200 steps, we introduce a change in the matrix $A$, replacing it with the matrix $A' = A \exp(\frac{\xi}{\|H\|_1} H)$. Here $H = \frac{1}{2}(B - B^\top)$ is antisymmetric and the entries of $B$ are sampled i.i.d. from $\mathcal{N}(0, 1)$, such that $\mathrm{dist}_{\mathrm{O}(d)}(A, A') = \xi$, where $\mathrm{dist}_{\mathrm{O}(d)}(A, A') = \|\log(A^\top A')\|_{\mathrm{HS}}$ is the usual geodesic distance on $\mathrm{O}(d)$. This construction ensures that $A'$ is orthonormal as well, and controls the distance between $A$ and $A'$ in a way consistent with the geometry of $\mathrm{O}(d)$.

Figure 3 (left) shows the result of this procedure, for varied values of $d$ and of $\xi$ ($\xi$ is only varied with $d = 16$), and averaged over 100 independent choices of the data sets and 50 independent choices of the matrix $A$. We observe that the change is detected rapidly, even before the sliding window is entirely filled with data from the new dynamics. Notably, higher dimensions require higher disturbances to be successfully detected with a fixed amount of data. Figure 3 (right) suggests a reason for this: the variance of the reference data set at the current state increases after change, indicating that the dynamics with $A'$ bring the trajectory into regions unexplored when following the dynamics with $A$. In other words, the lack of power in high dimensions seems to result from a lack of local data, as it is "easier" to go in unsampled regions by changing the dynamics.

Table 5 presents the hyperparameters for this experiment. The experiment used 2000 core hours.

Table 4: Hyperparameters used in generating Figure 2 (right).

| Parameter | Value |
|---|---|
| Input set | $\mathcal{X} = [-3, 3]^2$ |
| Mean function dimension | $m = 36$ |
| Data set size | $n = 500$ |
| Noise variance | $s^2 = 0.025^2$ |
| Regularization | $\lambda = 0.1$ |
| Bootstrap resamples | $M = 500$ |

Table 5: Hyperparameters used in generating Figure 3.

| Parameter | Value |
|---|---|
| Regularization | $\lambda = 0.01$ |
| Noise variance | $s^2 = 0.01^2$ |
| Significance level | $\alpha = 0.05$ |
| Bootstrap resamples | $M = 100$ |

Table 6: Hyperparameters used in generating Figure 4.

| Parameter | Value |
|---|---|
| Input set | $\mathcal{X} = [-1, 1]^2$ |
| Gaussian output kernel bandwidth | $\gamma^2 = 0.05$ |
| Mean function dimension | $m = 12$ |
| Data set size | $n = 100$ |
| Noise variance | $s^2 = 0.025^2$ |
| Regularization | $\lambda = 0.5$ |
| Bootstrap resamples | $M = 500$ |

Table 7: Hyperparameters used in generating Figure 5.

| Parameter | Value |
|---|---|
| Input set | $\mathcal{X} = [-1, 1]^2$ |
| Mean function dimension | $m = 12$ |
| Noise variance | $s^2 = 0.2^2$ |
| Regularization | $\lambda = 0.5$ |
| Bootstrap resamples | $M = 500$ |

Table 8: Hyperparameters used in generating Figure 6a and 6b.

| Parameter | Value |
|---|---|
| Input set | $\mathcal{X} = [-1, 1]^2$ |
| Mean function dimension | $m = 12$ |
| Noise variance | $s^2 = 0.2^2$ |
| Bootstrap resamples | $M = 500$ |

Table 9: Hyperparameters used in generating Figure 7.

| Parameter | Value |
|---|---|
| Input set | $\mathcal{X} = [-1, 1]^2$ |
| Mean function dimension | $m = 12$ |
| Data set size | $n = 100$ |
| Regularization | $\lambda = 0.25$ |
| Bootstrap resamples | $M = 500$ |

