# OpenReview forum: "Kernel conditional tests from learning-theoretic bounds"
_NeurIPS.cc/2025/Conference — NeurIPS 2025 poster_

### Official Review · Reviewer_nADy · 2025-06-08

**Clarity:** 2
**Significance:** 2
**Originality:** 2
**Rating:** 3
**Confidence:** 4

**Summary:**

This paper introduces a novel framework for hypothesis testing on conditional probability distributions, specifically developing a conditional two-sample test. The primary goal is to determine if the relationship between inputs (termed covariates) and outputs is the same across two different phenomena, with applications in fields like process monitoring, robotics, and medical studies.

**Questions:**

1. While the paper introduces bootstrapping schemes to avoid tuning inaccessible parameters, are there specific scenarios or types of datasets where these schemes might face practical limitations or lead to less accurate results? For instance, how robust are they to outliers or highly noisy data?

2. The framework is built upon kernel methods. What guidance can be provided on the selection of appropriate kernel functions for different types of data and applications to ensure optimal performance of the conditional two-sample test? Are there specific types of kernels that are more suited for certain conditional distribution characteristics?

**Ethical Concerns:**

["NO or VERY MINOR ethics concerns only"]

**Quality:**

2

**Strengths And Weaknesses:**

**Strengths**

1. A central idea is the transformation of confidence bounds from learning methods into conditional two-sample tests. This principle is applied and demonstrated using Kernel Ridge Regression (KRR) and conditional kernel mean embeddings.

2. To enhance practical usability, the paper introduces bootstrapping schemes.


**Weaknesses**

1. The central null hypothesis, as stated in Equation (5), is E(p1)(x) = E(p2)(x). This is a test for the equality of conditional expectations, not a conditional two-sample test in the general sense.

2. The authors dedicate a substantial portion of the paper to laying out a highly abstract mathematical foundation involving Markov kernels, subgaussian processes in Hilbert spaces, and reproducing kernel Hilbert spaces (RKHS) for vector-valued functions. While this formalism is necessary for the paper's specific theoretical proofs, it is disproportionate to the narrowness of the final result.

3. The entire framework is built on transforming the confidence bounds of Kernel Ridge Regression (KRR) into a statistical test. Consequently, the method inherits all the limitations of KRR. The most critical of these is the "well-specified" assumption, which requires that the true conditional expectation function lies within the chosen RKHS. This is a strong assumption that is difficult, if not impossible, to verify in practice, casting doubt on the reliability of the test's guarantees in real-world applications where model misspecification is the norm.

4.  The paper claims an advantage in its ability to provide "local" guarantees, identifying at which covariates two functions differ. However, the value of this localization is severely diminished when the test can only detect differences in the mean. A practitioner could use this test and find no significant difference across the entire covariate space, incorrectly concluding the conditional distributions are identical, while in reality, one might have collapsing variance or other critical behavioral changes that the test simply cannot see.

---

> ### Author Rebuttal · Authors · 2025-07-31
>
> Thank you very much for your detailed review.
> We address each of your points in details below.
> If you find our responses satisfactory, we would be grateful if you would consider raising your score.
>
> First, we would like to point out that the uncertainty bound in Theorem 4.3 / D.23 are, in our opinion, another key contribution of our paper in addition to the ones you emphasize in your summary, as they are the most general ones available in the literature.
> Their proof also involves novel steps and a novel assumption, which are major technical advances in the field.
>
> >The central null hypothesis, as stated in Equation (5), is $\mathbb E(p_1)(x) = \mathbb E(p_2)(x)$. This is a test for the equality of conditional expectations, not a conditional two-sample test in the general sense.
>
> We would like to stress that our framework and the resulting tests are **not** limited to just testing equality of conditional means.
> The test of Section 4 for conditional means holds for outputs in an abstract Hilbert space.
> We *specialize it* into a distribution-level test by a preliminary embedding of the data of the form $Y = \kappa(\cdot, Z)$.
> This yields a *conditional mean embedding*, but just as in the case of "unconditional" kernel two-sample tests [1], this does **not** mean that only equality of means can be tested. The properties of conditional distributions that can be tested depend on the kernel $\kappa$. For instance, a linear kernel can test for equality of means, polynomial kernels test for higher-order moments, and sufficiently expressive (*characteristic* [26, Section 3.1.1]) kernels like the Gaussian kernel can test for all moments. In Appendix A.1, we give a detailed exposition of these considerations and also conduct careful numerical experiments investigating the different testing behavior with respect to higher-order moments arising from different embedding kernels.
>
> We will clarify this in the revision by renaming Section 4 to "Testing of conditional expectations" and moving parts of Appendix A.1 to a dedicated subsection of Section 4 explaining how to specialize the test for full two-sample testing.
>
> >...While this formalism is necessary for the paper's specific theoretical proofs, it is disproportionate to the narrowness of the final result.
>
> Our results and methods actually use the abstract foundations in its formulation and usage, not just in its proof. In particular, the specialization to comparing higher-order moments of distribution fully relies on the fact that the output lies in an abstract Hilbert space to accommodate general embeddings for the data, which eventually allow us to use rich characteristic kernels and compare full distributions, eventually resulting in tests for complex properties of conditional distributions. The level of abstraction is thus a *feature* and is essential for practical usage.
>
> Furthermore, the final result that you emphasize (Theorem 4.4) is only **one** of our main contributions, as the concentration bounds for KRR we establish in Theorem 4.3 are also essential to our contribution. More restrictive variants of Theorem 4.3 have been widely used in bandits and Bayesian optimization with great success, leading to many practical and popular algorithms [R1,R2,20], and hence our generalization to more permissive cases is an important milestone in our opinion, as it enables a new class of applications by allowing conditional mean embeddings of distributions.
>
> >... the method inherits all the limitations of KRR. The most critical of these is the "well-specified" assumption...
>
> We preface this answer by saying that a similar point has been raised by Reviewers qzic and CBs5, and we refer you to our answers there for other aspects of the discussion.
>
> The known results of impossibility of conditional two-sample testing [3] indicate that **assumptions limiting the class of conditional expectations considered are necessary for guarantees**. Without such assumptions, no-free-lunch theorems apply and adversarial examples on which learning is impossible can be constructed.
> Our Theorem 4.1 formalizes that **one can escape the no-free-lunch of conditional testing via classical assumptions in learning theory**.
> In that regard, the well-specified assumption of KRR is a standard assumption in learning theory to enable successful learning.
> Whether or not it is restrictive is fully determined by context: it is widely accepted in Bayesian optimization [R2], and more debated for arbitrary conditional mean embeddings [R3].
> We would appreciate additional details on why you believe this assumption is too strong to enable further discussion on this point.
>
> Regarding how hard it is to check, it is again an unavoidable consequence of no-free-lunch results: such assumptions should capture the *prior knowledge* on the problem at hand, as there is no way of checking them a priori.
> We emphasize that this comment is *not* restricted to RKHS membership: it would be equally impossible to verify a priori that a function is Lipschitz continuous with a given Lipschitz constant [R4], or that it belongs to an interpolation space between an RKHS and $L_2$ (which is the classical assumption to handle the misspecified case in KRR) [40].
> While practitioners may have a better sense of whether *some* of these assumptions effectively hold, they ultimately remain assumptions that one cannot check.
>
> Summarizing, while we agree that it is unfortunate that such assumptions cannot be checked a priori, it is also a consequence of known no-free-lunch results in conditional testing, and *any* conditional test with guarantees must rely on such assumptions.
>
>
> >...the value of this localization is severely diminished when the test can only detect differences in the mean...
>
> As discussed above, our framework *does go beyond comparing conditional means*. We agree that in practice, capturing differences in higher moments is crucial—this is precisely what our general setting enables.
>
> >While the paper introduces bootstrapping schemes to avoid tuning inaccessible parameters, are there specific scenarios or types of datasets where these schemes might face practical limitations or lead to less accurate results? For instance, how robust are they to outliers or highly noisy data?
>
> These are very relevant questions for practical applications, but they exceed the scope of this study, which is a methodological paper introducing the new method from its theoretical foundation to the implementation.
> For this reason, we limit the scope of the experiment to illustrate the capabilities and behaviours in simple and well-understood scenarios, mainly for illustrative purposes and checks of performance.
> We do not try to explore all possible applications or explore the limitations of the heuristic.
>
> That being said, we also emphasize that the bootstrapping scheme we introduce is only a **heuristic** used to enhance power and avoid tuning hyperparameters.
> It can be directly replaced with any alternative without affecting the method.
> As an example, we will include in the revision results leveraging the bootstrapping scheme of [R5].
> Yet, a full study on benchmarking bootstrapping schemes for KRR to find the best performing ones and their limitations is left for future work.
>
> >The framework is built upon kernel methods. What guidance can be provided on the selection of appropriate kernel functions for different types of data and applications to ensure optimal performance of the conditional two-sample test? Are there specific types of kernels that are more suited for certain conditional distribution characteristics?
>
> This is a very relevant question. In fact, for the concrete kernel-based test we propose, one has to make two choices of kernels: One for the conditional kernel mean embedding, and one for kernel ridge regression.
>
> Regarding the first one, the choice of kernel determines which distributional properties of the conditional distributions are tested for (we discuss this in Appendix A.1 in detail), and ultimately it depends on the concrete application.
> We would like to stress that these considerations appear in all kernel embedding-based approach (for testing or even regression, like in distributional regression with two-stage sampling), and hence this has been discussed thoroughly in the literature.
> As a rule of thumb, if only lower moments are of interest, a linear or polynomial kernel should be chosen, whereas if all moments are of interest, an expressive kernel like the Gaussian kernel should be used. However, there can be tradeoffs, as we demonstrate with numerical experiments in Appendix A.1.
>
> Regarding the choice of kernel for KRR, this is a standard problem and many approaches have been proposed in the literature, cf. [37] for an overview and [26, Section 6.1] for a discussion of the fact that this is a general question in kernel methods.
> A detailed discussion of this question is therefore beyond the scope of this work.
> In general, the kernel should reflect the known or assumed smoothness properties of the regression function, and it should be sufficiently expressive.
> For example, in the context of conditional testing for physical dynamical systems, one can often assume a considerable degree of smoothness, and hence the smooth, yet expressive Gaussian kernel is a good default choice.
> Finally, we would like to point out that our UBD assumption allows a very broad range of admissible constructions of kernel with infinite-dimensional output spaces, and hence offers substantial freedom for the choice of the kernel for KRR.
>
> [R1] Srinivas et al., Information-theoretic regret bounds for gaussian process optimization in the bandit setting, 2010
>
> [R2] Sui et al., Safe exploration for optimization with Gaussian processes, 2015
>
> [R3] Klebanov et al., A rigorous theory of conditional mean embeddings, 2020
>
> [R4] Fiedler et al., On safety in Bayesian optimization, 2024
>
> [R5] Singh and Vijaykumar, Kernel ridge regression inference, 2023

---

> > ### Comment · Reviewer_nADy · 2025-08-08
> >
> > Thank you for your responses. I will maintain my rating.

---

> > > ### Author Response · Authors · 2025-08-08
> > >
> > > Thank you for your answer. We would like to point out that two of the four weaknesses stated in your initial review (item 1 and 4) are apparently based on a misunderstanding of our approach, as we have discussed in detail in our rebuttal, cf. also our discussion with reviewer CBs5. *In particular, these weaknesses do not affect our approach*. We would therefore kindly ask you whether you disagree with our explanation in the rebuttal of this issue.
> > >
> > > Regarding item 2, we would like to stress again that this level of abstraction is required for outputs in infinite-dimensional spaces and for the UBD kernels that make conditional mean embeddings feasible beyond trace-class restrictions. This is not only proof machinery: it is what enables the KRR confidence bounds for vector-valued outputs under weak assumptions and, consequently, finite-sample conditional two-sample testing with non-i.i.d. data.
> > >
> > > Regarding item 3, finite-sample conditional testing without structural assumptions is impossible: there is no non-trivial test with a prescribed level against all alternatives. Our framework makes the required learning-theoretic assumptions explicit and connects them to test guarantees. In this sense, the "well-specified" assumption is not an avoidable idiosyncrasy of our method but a way to escape the known impossibility; any alternative with finite-sample conditional guarantees must impose some prior structure, and for this we rely on established assumptions.
> > >
> > > Since we believe we address all issues raised in your review, we kindly ask you to clarify what leaves you unconvinced.

---

### Official Review · Reviewer_psDa · 2025-07-02

**Clarity:** 2
**Significance:** 3
**Originality:** 3
**Rating:** 5
**Confidence:** 4

**Summary:**

This paper's major contributions are:

- A framing of the relationship between conditional two-sample testing – particularly when it is desired to find for _which_ conditioning values those distributions differ – and learning-theoretic bounds on estimation error of regression.
- A new learning-theoretic bound for the estimation error of kernel ridge regression.
- A bootstrapping scheme for KRR that avoids using the overly-conservative and hard-to-estimate precise form of the bounds derived for the test.

**Questions:**

- How does Theorem 4.3/D.23, particularly in case 2, compare to the results of https://www.jmlr.org/papers/volume25/23-1663/23-1663.pdf that follow up [40]?
- Can the Singh and Vijaykumar KRR bootstrap help here?

**Ethical Concerns:**

["NO or VERY MINOR ethics concerns only"]

**Final Justification:**

The importance of the difference between UBD and the form used by Li et al., although they are clearly related, has been clarified a bit for me. I think an appendix going into further detail about this relationship will be quite important for the final version of this paper.

**Limitations:**

yes

**Paper Formatting Concerns:**

-

**Quality:**

3

**Strengths And Weaknesses:**

- The connection between estimation and testing is clear, and while the formal result of Theorem 4.1 is straightforward, setting it out in this way is illuminating. Connecting to the no free lunch theorem and the hardness result of Shah and Peters for conditional dependence is particularly helpful.
  - It might be _more_ helpful to expand further, perhaps in an appendix, on the relationship between conditional two-sample testing to conditional independence testing. It is probably not obvious to all readers that the problems can be directly reduced to one another, nor that doing so actually breaks the jointly-continuous assumption that was vital to the construction of Shah and Peters but does not invalidate hardness in this setting.
  - That this construction allows directly using time-uniform bounds is exciting! The paper could probably emphasize the practical implications of this a little bit more.

- Theorem 4.3, and the generalization D.23, appear novel and interesting.
  - It's a little disappointing that they don't yield practical tests, but this is also fairly expected; for instance, the 2012 paper after which this paper styles its title constructed bounds based on concentration inequalities (much simpler in the non-conditional setting).
  - Appendix C's discussion of the relationship to the results of [40] in particular seems quite sparse. These authors also had a more recent followup generalizing the setting: [https://www.jmlr.org/papers/v25/23-1663.html](Li et al. 2024). Especially, they remove the trace-class assumption in a slightly different way, but their construction (as discussed eg in Remark 2) seems closely connected to your UBD definition.

- The bootstrap scheme is straightforward and seems fine as a heuristic.
  - It would be good to relate this to other bootstraping schemes for kernel ridge regression, especially the scheme of [Singh and Vijaykumar (2023)](https://arxiv.org/abs/2302.06578) who provide finite-sample guarantees. Can their scheme be used here?

- Generally: I imagine some of the other reviewers will complain about the readability of this paper. It is certainly not a particularly approachable paper to non-experts in the field, but that seems somewhat inevitable for a paper taking on this set of results and framed in this way. This highlights a bit that Theorem 4.3/D.23, while relevant to the test procedure, is (a) a completely standalone result and (b) not actually used in the test procedure. Another strategy could be to write one more-accessible paper about conditional testing and its relationship to estimation error along with the heuristic for estimating it, and another paper about the theorem; I'm not sure this is necessarily a better strategy, but some reviewers might disagree.

---

> ### Author Rebuttal · Authors · 2025-07-31
>
> Thank you very much for your detailed review and positive evaluation.
> We will address each of your concerns and questions in turn.
> If you find our answers satisfactory, we would be grateful if you consider raising your score.
>
> >It might be _more_ helpful to expand further, perhaps in an appendix, on the relationship between conditional two-sample testing to conditional independence testing. It is probably not obvious to all readers that the problems can be directly reduced to one another, nor that doing so actually breaks the jointly-continuous assumption that was vital to the construction of Shah and Peters but does not invalidate hardness in this setting.
>
> Thank you very much for this suggestion. This would indeed be helpful for readers, and we will include a corresponding discussion in the appendix.
>
> >Theorem 4.3, and the generalization D.23, appear novel and interesting.
> It's a little disappointing that they don't yield practical tests, but this is also fairly expected; for instance, the 2012 paper after which this paper styles its title constructed bounds based on concentration inequalities (much simpler in the non-conditional setting).
>
> We agree that the situation somewhat parallels the situation in [1]. However, we would like to point out that while the uncertainty bounds from Theorem 4.3 / D.23 are in general not directly applicable in practice (due to the need for an RKHS norm bound and potential conservatism), they are useful for deriving a practical testing procedure by providing a theoretically-sound and convenient form of the uncertainty bound, that in turn suggests a practical Bootstrap approach.
>
> >That this construction allows directly using time-uniform bounds is exciting! The paper could probably emphasize the practical implications of this a little bit more.
>
> This is a very good point and we will add corresponding remarks in the revised version. Indeed, time-uniform uncertainty bounds play an important role in many practical algorithms, from bandits / Bayesian optimization, sequential testing, to learning-based control. For example, our uncertainty bounds might be of direct use in multi-task Bayesian optimization, cf. [20].
>
> >Appendix C's discussion of the relationship to the results of [40] in particular seems quite sparse. These authors also had a more recent followup generalizing the setting: https://www.jmlr.org/papers/v25/23-1663.html (Li et al. 2024). Especially, they remove the trace-class assumption in a slightly different way, but their construction (as discussed eg in Remark 2) seems closely connected to your UBD definition.
>
> Thank you for pointing this out. We will extend the corresponding passages in Appendix C, and we provide a detailed discussion in the answer to your first question below.
>
> >Generally: I imagine some of the other reviewers will complain about the readability of this paper. It is certainly not a particularly approachable paper to non-experts in the field, but that seems somewhat inevitable for a paper taking on this set of results and framed in this way. This highlights a bit that Theorem 4.3/D.23, while relevant to the test procedure, is (a) a completely standalone result and (b) not actually used in the test procedure. Another strategy could be to write one more-accessible paper about conditional testing and its relationship to estimation error along with the heuristic for estimating it, and another paper about the theorem; I'm not sure this is necessarily a better strategy, but some reviewers might disagree.
>
> Thank you very much for your thoughtful comment.
> We agree that the manuscript is rather technical, but since we want to develop a general, theoretically-grounded framework that is also applicable, we need some machinery to realize this.
> To increase the readability and accessibility of the manuscript, we have already included a tutorial-style overview on practical aspects in Appendix A.1 and a convenient summary of the algorithms in Appendix A.2, and we will include your and the other reviewer's comments to further improve the user-friendliness.
> Furthermore, we agree that Theorem 4.3 / D.23 is indeed a result of significant independent interest. However, it is essential for our testing approach since 1) it informs the type of bootstrapping approach and 2) it provides a theoretical justification of the overall approach,
> and hence the application to our general testing framework provides a compelling and concrete demonstration of the utility of Theorem 4.3 / D.23.
> For these reasons, we think that combining our general framework, our results on uncertainty bounds, and the concrete kernel-based tests, leads to a cohesive article, and that the advantages of this approach outweigh the potential disadvantages.
>
> >How does Theorem 4.3/D.23, particularly in case 2, compare to the results of https://www.jmlr.org/papers/volume25/23-1663/23-1663.pdf that follow up [40]?
>
> Conceptually, in both [40] / Li et al '24 and in our work, a tensor structure (of the vRKHS) is used to circumvent a trace-class assumption. More precisely, in both cases a multiplicative structure of the kernel leads to a tensor product structure on the vRKHS, which in turn is used for the analysis. However, we would like to point out the following important differences.
> 1. We introduce a novel structural assumption (UBD) generalizing the common multiplicative structure of kernels which has been used in [40] / Li et al '24. The UBD assumption essentially encapsulates the minimal structure that is required to make our proof technique work. We suspect that the UBD structure might actually be suitable for further generalizing the results from [40] / Li et al '24, but this is an interesting direction for future work and beyond the scope of this manuscript.
> 2. The actual use of the tensor structure is rather different. Whereas [40] / Li et al '24 use it to generalize the classic integral operator technique without a trace class assumption, we use it (actually, the more general UBD structure) to adapt the method of mixtures (more precisely, to use a multiplicative double-mixture argument), leading in our case to a novel uncertainty bound.
> 3. [40] / Li et al '24 are concerned with the analysis of the KRR learning algorithm (in particular, learning rates), whereas we derive new uncertainty bounds for KRR, which in turn can be used in downstream algorithms or statistical procedures (like the conditional tests in our case). The issue of the trace-class assumption becomes particularly relevant here, since in existing uncertainty bounds a Fredholm determinant appears, which is not even defined for non-trace class operators. In other words, we need this tensor product approach to even formulate the right result, instead of just analyzing an existing method.
> 3. Finally, the concrete use of the tensor structure (ignoring for now the generalization to UBD) is different. [40] / Li et al '24 rely primarily on an identification with Hilbert-Schmidt operators, whereas we work directly with the tensor structure.
>
> >Can the Singh and Vijaykumar KRR bootstrap help here?
>
> Yes. Our bootstrapping scheme can be drop-in replaced with any available alternative, which we will emphasize more in the revision.
> Furthermore, we will include results with the Singh and Vijaykumar KRR bootstrap in the revision. Their method is computationally more scalable and more robust to misspecification of the regularization parameter. However, preliminary results indicate that it also leads to more conservative bounds and lower power in low-data settings. Please also note that their guarantees do not hold in our more general setting (e.g., they assume i.i.d. data and scalar-valued outputs).

---

> > ### Comment · Reviewer_psDa · 2025-08-07
> >
> > Thanks for your replies. I think this more thorough discussion of the relationship of the work to Li et al. is important to add to the paper, and appreciate the pointer towards new results with the Singh/Vijaykumar KRR bootstrap, which will also help with the paper.
> >
> > > [40] / Li et al '24 are concerned with the analysis of the KRR learning algorithm (in particular, learning rates), whereas we derive new uncertainty bounds for KRR
> >
> > I was a little confused by this statement, but I think I've figured out how to interpret it now:
> > - You give bounds on the convergence of $f\_{D\_{:n}}(x)$ to $\mathbb{E}[p]\(x)$ for all $\mathcal X$, which can be thought of like a $\mathcal G,\infty$ norm on the regression.
> > - Their bounds are about the convergence of $[\hat F\_\lambda]$ to $F\_*$ in their $\gamma$ norm, which for $\gamma = 1$ I *think* corresponds to the Hilbert-Schmidt norm for $\mathcal H \to \mathcal Y$ operators. Using the reproducing property and Cauchy-Schwarz, I think this should mean that their bound (under the conditions that you can get $\gamma = 1$ with their theorem) implies a bound of your form that's larger by a factor of $\sup\_{x \in \mathcal X} \sqrt{k(x, x)}$. Right?
> >
> > Anyway, I think it is important for you to work these kinds of details out in an appendix in your revised version, but I'm satisfied that your results are a different enough setting that I'm happy to tick up my score.

---

### Official Review · Reviewer_PbDf · 2025-07-03

**Clarity:** 3
**Significance:** 2
**Originality:** 3
**Rating:** 5
**Confidence:** 2

**Summary:**

The authors establish conditions under which confidence bounds for KRR are equivalent to credible intervals given by Gaussian Process regression. They then show how they can be converted into conditional hypothesis tests. The main idea seems to be to base the hypothesis test on the pointwise overlap between two GP posteriors: the more overlap, the more likely one fails to reject the null hypothesis (at that particular input).

**Questions:**

1) Why frame the learning algorithm as KRR with confidence bounds? The equations appear much more similar to GP regression with credible intervals given by the posterior covariance. A similar point is mentioned in passing in line 308, but to me this seems like the core insight behind the whole theorem! If that's true, then I think this should be emphatically mentioned earlier.

2) If it is true that the proposed test is equivalent to examining the (pointwise) posterior distribution of the difference between the two estimators and testing whether the credible interval contains 0, then can you clarify how Theorem 4.3 is different from the standard theory of GP regression?

3) It is known that functions that are less smooth w.r.t. to chosen kernel (i.e., have larger RKHS norm) are harder to learn (in KRR and GP regression). Intuitively, it seems that one should be less likely to reject the null in the case that the data generating processes is not smooth (and thus hard to learn). However, my understanding is that the posterior variance is blind to the alignment between the kernel and E[p](x), so the hypothesis test cannot depend on the smoothness of E[p](x). Can you provide some intuition as to why the hypothesis test does not seem to depend on this? Is it simply that the estimation error is contained in the posterior mean rather than the variance? (It could be that I'm simply mistaken about the relationship between estimation error and hypothesis testing.)

**Ethical Concerns:**

["NO or VERY MINOR ethics concerns only"]

**Final Justification:**

My main concern was understanding the significance of the work. The result initially seemed very similar to standard results in Gaussian process regression. The authors clarified that their results *generalize* the previous results. In particular, in order to devise tests comparing various *distributional* quantities, one needs to consider regression with infinite-dimensional outputs. (I did not independently verify this claim, but it seems plausible.) The authors' technical contribution is in devising confidence bounds for regression problems of this class. My evaluation of the significance of this work is now much higher.

I would rate my confidence as 2 or 3. I am not an expert in this area but the explanations are consistent with what I have seen before.

**Limitations:**

yes

**Quality:**

4

**Strengths And Weaknesses:**

I preface this review by stating that the research area and proof techniques are quite unfamiliar to me. I can't fairly evaluate the impact or originality of the work.

**Strengths.**
* The writing is very clear, and the main argument is not too hard to follow.
* There are no clear errors in the mathematical framing.
* The idea is pretty neat -- in general it seems useful to find Rosetta stones between learning theory and statistical testing, and this result seems to be a big one. To be honest, I'm a bit surprised that this has not been done before!

**Weaknesses.**
* The technical formality obscures the main idea, in my opinion. I understand that rigor is important to some researchers, but I found it quite difficult to parse the theorem statements. Perhaps informal versions of the theorems can be given early in the paper.

---

> ### Author Rebuttal · Authors · 2025-07-31
>
> Thank you very much for your review and the positive evaluation.
> Below we address each of your concerns and questions in turn.
> If you find our answers satisfactory, we would be grateful if you consider raising your score.
>
> > The technical formality obscures the main idea, in my opinion. I understand that rigor is important to some researchers, but I found it quite difficult to parse the theorem statements. Perhaps informal versions of the theorems can be given early in the paper.
>
> We understand that this comment is particularly referring to Theorems 4.3 and 4.4, given that an intuitive interpretation of Theorem 4.1 is available right after its formal statement.
> We thank you for your comment and will provide informal summaries in the text surrounding them.
> We believe this is more suitable than stating them informally earlier (e.g., in the introduction) given that the whole of Section 3 is about making precise the terminology of conditional testing.
>
> > Why frame the learning algorithm as KRR with confidence bounds? The equations appear much more similar to GP regression with credible intervals given by the posterior covariance. A similar point is mentioned in passing in line 308, but to me this seems like the core insight behind the whole theorem! If that's true, then I think this should be emphatically mentioned earlier.
>
> Your identification between GP regression and KRR is correct; the two estimators coincide up to rescaling of the noise/regularization parameter [R1].
> The name given to the algorithm is thus mainly a matter of preference.
> Historically, the bounds of the form of Theorem 4.3 have been mainly used in bandits [R2] and Bayesian optimization [R3].
> They are thus often called "frequentist uncertainty bounds for Gaussian process regression".
> We decided to break from this terminology for two main reasons.
> First, Gaussian processes are often used in Bayesian frameworks; yet, the bound is *not* Bayesian. It quantifies the difference between the estimator (the GP mean) and the ground truth, which does not have an equivalent in Bayesian methods. Furthermore, the proof is *not* based on the confidence intervals of the Gaussian distribution nor on the fact that the posterior of a GP is a GP. The posterior variance also appears from an entirely different mechanism than conditioning (specifically, it appears in the beginning of the proof in (19) in Lemma D.2).
> Second, Gaussian processes are often used in settings with a simple output space (e.g., just scalar outputs), whereas here we allow even outputs in an abstract Hilbert space.
> This is relatively uncommon for Gaussian processes, but more usual with KRR.
>
> We do agree that there is value in the Gaussian process interpretation, however, and will highlight it further in the revision, both in the introduction and around Theorem 4.4.
>
> > If it is true that the proposed test is equivalent to examining the (pointwise) posterior distribution of the difference between the two estimators and testing whether the credible interval contains 0, then can you clarify how Theorem 4.3 is different from the standard theory of GP regression?
>
> Your interpretation is almost perfect: the posterior variances need to be scaled by the appropriate factor $\beta$, which can be interpreted as a correction term to obtain frequentist bounds from the Bayesian ones.
> This being clarified, Theorem 4.3 is *not* different from the standard theory of GP regression; it is a generalization.
> Specifically, GP regression admits well-known frequentist confidence intervals.
> The most well-known ones hold in the scalar output case [R2,12,19].
> Recent generalizations allow *infinite-dimensional outputs with a trace-class kernel*, and require strong additional assumptions on the kernel and the input set (bounded kernel on a compact set).
> In contrast, our result is the first to allow *infinite-dimensional outputs with non-trace-class kernels*, which allows the classical case of $K = k\cdot\mathrm{id}_{\mathcal G}$, and also removes *all* regularity assumptions connecting the kernel to the input set, allowing a more varied class of such sets.
>
> > it seems that one should be less likely to reject the null in the case that the data generating processes is not smooth (...) my understanding is that the posterior variance is blind to the alignment between the kernel and $\mathbb E[p]$ (...) Can you provide some intuition as to why the hypothesis test does not seem to depend on [the smoothness of $\mathbb E[p]$]?
>
> Your understanding that testing for less smooth functions is harder is indeed correct, and the test *does* depend on the smoothness of $\mathbb E[p]$ in two ways:
> 1. We assume that $\mathbb E[p]$ lies in the RKHS.
> 2. We also assume a known upper-bound on $\lVert\mathbb E[p]\rVert_k$, and this upper bound enters the threshold in the constant $\beta$ ($\beta$ has the form $\lambda^{-1/2}(B + \cdots)$, where $B$ is the upper bound). In other words, testing for functions with higher RKHS norms (which can be interpreted as more complex functions) leads to higher thresholds, and thus require more data to trigger.
>
> [R1] Kanagawa et al., Gaussian Processes and Kernel Methods: A Review on Connections and Equivalences, 2018
>
> [R2] Srinivas et al., Information-theoretic regret bounds for gaussian process optimization in the bandit setting, 2010
>
> [R3] Sui et al., Safe exploration for optimization with Gaussian processes, 2015

---

> ### Comment · Reviewer_PbDf · 2025-08-01
> **What is the significance?**
>
> To understand the significance of the work, it seems to me that I should understand the statement "In contrast, our result is the first to allow infinite-dimensional outputs with non-trace-class kernels, which allows the classical case of $K = k\cdot\mathrm{id}_{\mathcal G}$, and also removes all regularity assumptions connecting the kernel to the input set, allowing a more varied class of such sets." Some clarifying questions, then: Do we expect these generalized cases to reveal insights that may be relevant in practice? In other words, are there practical settings which are "closer to" your setting, but far from the settings covered by prior theory? Or would you characterize this result as covering corner cases which are primarily of mathematical interest?

---

> ### Author Response · Authors · 2025-08-03
> **Discussion of significance of the generality of the results**
>
> Thank you very much for acknowledging our answer and engaging in the discussion.
> Summarizing our answer to your follow-up question: **yes**, Theorem 4.3/D.23 has advantages in applications, as it *enables* common choices for kernels with infinite-dimensional output spaces such as $K = k\cdot\mathrm{id}_{\mathcal G}$, which enables in turn working with conditional mean embeddings of distributions and *testing for equality of full distributions* (or other complex properties of distributions).
> In contrast, previous results would only allow testing for equality of finitely many moments (or require complex kernel engineering and a theory of distributions embeddings other than that of CMEs that works with trace-class kernels, which is not available to our knowledge).
> We further detail this answer below.
>
> We begin with some context.
> In order to implement our general testing framework, we transform it (via conditional mean embeddings) into a regression problem with a Hilbert-space valued output, which we then tackle with KRR, and use frequentist uncertainty bounds for the latter in the actual test.
> Since we want to be able to test for complex distributional properties (beyond, say, just the mean or low-order moments), we use an expressive embedding (via a *characteristic* kernel for the CME) which in general maps to an infinite-dimensional (output) Hilbert space.
> We discuss this in detail in Appendix A.1 as well as in our answer to reviewer nADy.
> In other words, **from practical considerations, we need uncertainty bounds for KRR with an infinite-dimensional output space**.
> Furthermore, such bounds are of interest beyond our testing framework; for example, for multi-task bandits [20].
>
> In order to do KRR with an infinite-dimensional output space, the most common approach in practice is to use a kernel of the form $K = k\cdot\mathrm{id}_\mathcal G$, where $k$ is a scalar kernel.
> However, such a kernel *cannot fulfill a trace-class assumption* for an infinite-dimensional output space $\mathcal{G}$ since the identity mapping $\mathrm{id}_{\mathcal G}$ is not trace-class in this case.
> All previous time-uniform uncertainty bounds for KRR need this trace-class assumption, and *the assumption is even necessary to formulate such uncertainty bounds* since the formula contains the Fredholm determinant, which is only defined for trace-class operators in general.
> This is a considerable gap in the literature with practical repercussions, since it prevents the use of kernels of the form $K = k\cdot\mathrm{id}_{\mathcal G}$ where time-uniform uncertainty bounds are needed, which are essential in applications as highlighted above.
> In particular, without our new uncertainty bounds, we could not use the most common type of kernels for infinite-dimensional output spaces for our tests, nor perform conditional testing of full distributions, when time-uniform bounds are necessary.
> More generally, our results allow to use this classic type of kernels whenever time-uniform uncertainty bounds for KRR are necessary (e.g., in kernelized bandits).
>
> Regarding the regularity, one important advantage of kernel methods is the ability to apply them to arbitrary input spaces, as long as suitable kernels are available.
> Requiring additional regularity assumptions or structural restrictions (say, to a Euclidean space) is therefore a potential inconvenience in practice.
> For example, when testing on dynamical systems, one might want to be able to take the structure of the state space into account, or when testing on dynamics generated by PDEs, one might need to be able to work with Hilbert spaces as inputs.
> Since our results work under essentially minimal assumptions on the kernel, they conveniently can cover all of these cases.
>
> In summary, the generality of our results are what bridges the critical step necessary for testing full conditional distributions, while also allowing non-i.i.d. data from unstructured input spaces, which are common in practice.

---

> > ### Comment · Reviewer_PbDf · 2025-08-04
> > **Very clear, thank you. Increasing score to 5.**
> >
> > See title. I'd recommend the authors consider clarifying or expanding on this discussion in the main text. I saw it mentioned in the related work section, but I could not appreciate the significance as a non-expert.
> >
> > Note to AC: I would rate my confidence as 2 or 3. I am not an expert in this area but the explanations are consistent with what I have seen before.

---

> ### Author Response · Authors · 2025-08-04
>
> Thank you for raising your score. We will follow your recommendation and expand on this discussion in the main text.

---

### Official Review · Reviewer_CBs5 · 2025-07-03

**Clarity:** 2
**Significance:** 3
**Originality:** 2
**Rating:** 5
**Confidence:** 4

**Summary:**

The paper proposes a conditional two-sample testing framework using kernel methods with finite-sample guarantees. The approach builds confidence bounds via kernel ridge regression (KRR), leveraging existing tools like Abbasi-Yadkori's bounds. While the core idea of using KRR confidence intervals for testing is intuitive, the claimed ability to perform pointwise inference in regions with vanishing probability mass appears to conflict with known impossibility results (the paper even acknowledges this in Section 2: "Conditional testing is notoriously hard—there exists no nontrivial test with prescribed level against all alternatives." ). The authors propose an ad-hoc bootstrapping scheme to address conservative non-asymptotic confidence regions, but this lacks theoretical justification. More fundamentally, the well-specified assumption (that true conditional means lie in the RKHS) is both unrealistic and fails to address how the method circumvents the acknowledged impossibility results for conditional testing.

The paper proposes a framework for conditional two-sample testing that focuses on identifying covariate values x where the conditional expectations $\mathbb{E}\_{p_1}[Y\mid X=x]$ and $\mathbb{E}\_{p_2}[Y\mid X=x]$ differ. Concretely, the authors reduce the testing problem to comparing two Kernel Ridge Regression (KRR) estimators. The core idea (Theorem 4.2) is that if each KRR estimator $f_{D_i}$ comes with a confidence bound $B_i(D_i,x)$ (i.e. with probability $1-\delta_i$ we have $|f_{D_i}(x) - \mathbb{E}[Y\mid X=x]|\le B_i(D_i,x)$ for all x in some set S and all sample sizes in N), then one can reject the null $H_0:\mathbb{E}\_{p_1}[Y\mid x]=\mathbb{E}\_{p_2}[Y\mid x]$ whenever $\| f_{D_1}(x) - f_{D_2}(x)\| > B_1(D_1,x) + B_2(D_2,x)$. Under these conditions the authors claim that the test has (pointwise) type‐I error bounded by $\delta_1+\delta_2$. So, the paper then aims to supply the required confidence bounds by deriving novel KRR concentration results. In Theorem 4.3 the authors claim a time-uniform (anytime) bound for vector-valued KRR under fairly general “uniform-block-diagonal” (UBD) operator-valued kernels $K(x,x')=\iota(K_0(x,x')\otimes {\rm Id}_V)\iota^{-1}$ and subgaussian noise. This result extends classic scalar KRR bounds to infinite-dimensional outputs and non-trace-class kernels (covering, e.g., conditional mean embeddings). Combining these bounds Theorem 4.4 yields a concrete test: both data sets produce KRR predictors and associated posterior variances $\sigma(x)$, and one rejects $H_0(x)$ if $|f_1(x)-f_2(x)| > \beta_1,\sigma_1(x)+\beta_2,\sigma_2(x)$, where $\beta_i$ depend on confidence levels. Theorem 4.4 shows this test has level $\alpha_1+\alpha_2$ over the covariate region X (or a chosen subset S). The authors also propose a practical bootstrap calibration (Appendix B) to estimate the thresholds $\beta_i$ from data, avoiding manual tuning of RKHS-norm bounds. Empirically, the paper demonstrates in synthetic experiments that the proposed test controls type I error and can achieve higher power than a global conditional-distribution test (Hu & Lei [5]) when the difference in conditional means is localized in rarely-sampled regions of X. They also present toy illustrations of process change-detection using a sliding-window test on linear dynamical systems (Appendix A.3). Overall, the paper claims to provide a “comprehensive foundation” for conditional two-sample testing – bridging theoretical guarantees and practical implementation

**Questions:**

NA

**Ethical Concerns:**

["NO or VERY MINOR ethics concerns only"]

**Final Justification:**

Based on the authors' rebuttal, I believe that they can successfully revise manuscript to integrates all the remarks from myself and other reviewers and meet the acceptance bar. Therefore, I rise my score. That said, I find that revision is very much needed to improve clarity (narrative-wise and theory-wise), especially stressing out fine-line aspects of the impact of assumptions (in particular well-specified) to testing and discussing limitations of the approach taken.

**Limitations:**

Not discussed.

**Quality:**

3

**Strengths And Weaknesses:**

### __Strengths__

- __Related works:__ The authors did a good job reviewing related works and properly citing why conditional testing is a hard problem.

- __Theoretical framework:__ The reduction from conditional testing to learning-theoretic confidence bounds in Theorem 4.2 (any two regression learners with high-probability error bounds yield a valid conditional test) seams elegant.

- __Handling non-i.i.d. data:__ By allowing online sampling (a process of transition pairs) the method covers dependent or sequentially collected data. This is a strength over classical tests that require i.i.d. samples; the authors correctly emphasize that “outputs may not be independent”.


### __Weaknesses__

-  __Theoretical Issues:__ The method's claim to detect differences in vanishing-probability regions ($\theta\rightarrow 0$) lacks proper theoretical support and seem to contradict well-known impossibility results. From my point of view, the  __authors fail to__:
(a) Specify necessary assumptions (e.g., density bounds, smoothness) to circumvent impossibility results.
(b) Explain how their well-specified assumption and UBD kernels with trace-class constraints overcome fundamental limitations
In that regard, I suspect that the well-specified assumption and UBD kernel structure may implicitly restrict the complexity of joint distribution of Y|X to simple scenarios, significantly limiting practical applicability.

- __Scope:__  The core method explicitly tests equality of conditional expectations. If two conditional distributions differ in higher moments (e.g. variance) but have the same mean, the basic test will not flag a difference. The authors mention that one can embed outputs or transform data (Algorithm 3 for CMMD) to compare other moments or full distributions, but these extensions are not the main focus. It should be emphasized that the presented test is primarily for conditional mean differences, and that extending to full distributional tests requires additional work.

- The proposed __bootstrapping method lacks theoretical guarantees__ and may overfit
The well-specified assumption (that the true conditional means lie in the RKHS) is overly restrictive in practice. There is no guarantee that the chosen UBD kernel can adequately model arbitrary conditional distributions, especially without strong prior knowledge.

- __Empirical Shortcomings:__
     - No validation on non-IID or sequential data despite claims
     - Experiments limited to toy examples:
     - Low-dimensional X (in dimension 1 or 2)
     - No testing on high-dimensional complex objects or mis-specfied settings

- __Baseline Comparisons:__
     - Only compared to one baseline ([5]), omitting modern alternatives
     - No ablation study to demonstrate UBD's contribution


I summary, I cannot recommend this submission for acceptance due to:
1. Insufficient theoretical justification for key claims (particularly rare-region testing)
2. Lack of theoretical guarantees for the bootstrapping approach
3. Narrow experimental validation that doesn't support practical utility

---

> ### Author Rebuttal · Authors · 2025-07-31
>
> Thank you very much for your review and comments. You give the following reasons for suggesting non-acceptance:
> * Insufficient theoretical justification,  particularly for rare-region testing
> * Lack of guarantees of bootstrapping
> * Narrow experiments
>
> We address each of these points in our rebuttal below, together with the other points you raise. In summary:
> * We respectfully disagree on the lack of theoretical justification on the rare-region testing;
> * Bootstrapping is only a pragmatic heuristic that can be replaced with any available alternative; our core contributions are the **formalization of conditional testing guarantees**, **new confidence bounds for KRR and vector-valued LS**, and their **instantiation in a conditional test**.
> * Our experiments enable understanding the test in controlled scenarios, cover non-iid data generation up to dimension 16, and compare to a modern baseline.
>
> If you find our responses satisfactory, we would be grateful if you consider raising your score.
>
> > contradicting well-known impossibility results (rare-region sampling, $\theta\to0$)
>
> We believe this comment stems from a misunderstanding: Theorem 4.3 guarantees that, with high probability, the difference between the estimator and the truth is at most $\beta\cdot \sigma$ **everywhere**.
> Hence the test can still trigger in sparsely sampled regions --- though the threshold grows where data are scarce and does not fire where no data exist without a strongly extrapolating kernel.
>
> The key assumptions for this **input-uniform** bound is **RKHS membership** of the conditional expectation.
> For common kernels (e.g. Gaussian, Matérn) this is a smoothness condition.
>
> For those reasons, **our results do not violate known impossibility results**: rather, the assumption of RKHS membership appropriately limits the class of allowed Markov kernels to those for which the learning algorithm has some (mild, and depending on the kernel) extrapolation power.
>
> As you highlight it, **such assumptions are necessary** to escape no-free-lunch results.
> Whether this particular one is "too strong" is thus highly dependent on context. It is a commonly accepted assumption in bandits [R2] and in Bayesian optimization, where it has led to highly successful algorithms [R3,20].
> It is more criticized in the presence of conditional mean embeddings [R4], which our setup allows.
> In any case, we see it as a starting point: yes, this assumption does impose *some* restriction. Such restrictions are *necessary*, but we are currently investigating whether we can relax RKHS membership while preserving input-uniformity.
> Finally, note that our framework also accepts *any* learner for which similar bounds exist; KRR is just one example.
>
> > the authors fail to: (a) Specify necessary assumptions to circumvent impossibility results. (b) Explain how their well-specified assumption and UBD kernels (...) overcome fundamental limitations
>
> Our previous answer fully addresses these concerns: the core necessary assumption is RKHS membership. Subgaussian noise and UBD kernel are *also* essential technical assumptions to make the proof work, but the core idea overcoming known impossibility results is the regularity of conditional expectations imposed through RKHS membership (with known norm upper-bound).
>
> > the well-specified assumption and UBD kernel structure may implicitly restrict the complexity of joint distributions of Y|X to simple scenarios, significantly limiting practical applicability.
>
> It is **correct** that well-specification restricts the complexity of the joint distribution.
> It is **necessary** to restrict this complexity for any guarantees to hold, as shown by the impossibility results you mention.
> We fail to see why our assumption restricts the joint distribution to only simple scenarios or significantly limits the practical applicability, and would appreciate additional clarifications on this point to address remaining concerns you may have.
>
> > the method tests equality of conditional expectations (...)  extensions [to compare other moments or full distributions] are not the main focus (...) extending to full distributional tests requires additional work.
>
> We strongly emphasize that **the method we present in Appendix A to compare other moments or full distributions is not an extension, it is a special case**.
> The test of Section 4 compares conditional expectations *in an abstract Hilbert space*.
> We *specialize* it in a distribution-level test by choosing $Y = \kappa(\cdot, Z)$: for a characteristic kernel $\kappa$, we compare full conditional distributions.
> We claim this pertains to experiment design, however, and does not require additional theoretical work.
> We will (i) rename Section 3 “Testing of conditional expectations’’ and (ii) move the CME specialization from Appendix A to the main text for clarity.
>
> > The proposed bootstrapping method lacks theoretical guarantees and may overfit
>
> We present bootstrapping purely as a power-enhancing **heuristic**; it is not a main contribution. Any alternative can replace it without affecting our theory. In the revision we will additionally report results with the scheme of [R1], which enjoys some guarantees under suitable assumptions.
> Preliminary results indicate that this method is more robust to the misspecification of the regularization parameter, but also leads to significantly more conservative bounds and lower power.
> Formal guarantees for input- and time-uniform bootstrapping with online data is an open research problem to our knowledge and is out-of-scope.
>
> > The well-specified assumption (that the true conditional means lie in the RKHS) is overly restrictive in practice
>
> Our comments above illustrate why an assumption of this kind is necessary to escape impossibility results, as you identify it.
> We would thus appreciate additional details on why you believe this specific assumption is overly restrictive in practice to enable further discussion.
>
> > no guarantee that UBD can adequately model arbitrary conditional distributions, especially without strong prior knowledge
>
> We emphasize again that it is precisely the purpose of our assumptions to **appropriately restrict** the conditional distributions that can be modeled, as is necessary to escape no-free-lunch results.
> In particular, any fixed kernel will not be able to capture some conditional distributions.
> This is in particular the case for any UBD kernel.
> Instead, *the role of the kernel is to encode the prior knowledge about the functions at hand*.
>
> > No validation on non-IID or sequential data despite claims
>
> These experiments are presented in Section A.3, where data consists of input-output pairs of a dynamical system and is generated sequentially by following trajectories. Our test reliably detects changes as the dynamics are changed.
> We will use the additional space in the revision to move this section to the main text.
>
> > Experiments limited to toy examples, Low-dimensional X (in dimension 1 or 2), No testing on high-dimensional complex objects or misspecfied settings
>
> While we do consider 1- or 2-D inputs in toy cases for illustrative purposes, outputs are up to infinite dimensional. Further, some other cases reach 16-dimensional inputs.
> Given that the purpose of these experiments is to illustrate the capabilities in well-understood scenarios, we deem this adequate for a first methodological paper.
> Full-scale applications merit separate, dedicated work.
>
> > Only compared to one baseline ([5]), omitting modern alternatives
>
> The baseline [5] we compare to is less than a year old and leverages conformal methods, which are an active area of research.
> In our opinion, it thus qualifies as a "modern alternative".
> We would be happy to consider additional baselines that you find appropriate.
>
> > No ablation study to demonstrate UBD's contribution
>
> The goal of UBD is to be a theoretical tool enabling the most general pointwise concentration bounds for KRR to this day.
> Furthermore, in practice, the kernel is a design choice supposed to accommodate for prior knowledge.
> In any case, it is not meant as a structure to benchmark on.
> We would appreciate further details on what kind of ablation study you would expect.
>
> > The approach builds confidence bounds via kernel ridge regression (KRR), leveraging existing tools like Abbasi-Yadkori's bounds
>
> We thoroughly emphasize that we **do not** leverage the bounds of Abbasi-Yadkori. Rather, we **generalize these bounds to previously inaccessible setups**.
> The original bounds only consider the scalar case, and available generalizations are limited to trace class kernels and impose strong regularity conditions on the kernel.
> We remove these assumptions and allow for non-trace class kernels.
> What we leverage is the *proof method* of Abbasi-Yadkori, which we adapt to accommodate for the more general assumptions.
>
> [R1] Singh, R., Vijaykumar, S. (2023). Kernel ridge regression inference. arXiv preprint arXiv:2302.06578.
>
> [R2] Srinivas, N., Krause, A., Kakade, S. M., Seeger, M. (2010). Gaussian Process Optimization in the Bandit Setting: No Regret and Experimental Design. In Proceedings of the 27th International Conference on Machine Learning (ICML).
>
> [R3] Garnett, R. (2023). Bayesian optimization. Cambridge University Press.
>
> [R4] Klebanov et al., A rigorous theory of conditional mean embeddings, 2020

---

> ### Comment · Reviewer_CBs5 · 2025-08-04
>
> I thank the authors for their reply. I acknowledge most of the replies. In particular:
>
> Concerning **Theoretical Issues**, indeed, as I wrote,  I was expecting that we well-specified assumption mitigates the problems. I think  that elaborating on this more carefully in the revision would help the reader.
>
> Concerning **Scope**, thank you for clarifying. Again, I believe this aspect to be important to clarify, in particular since the test handles distributions via CME while making well-specified assumption for charasteristic kernel. I believe that the limitations of doing so should be clearly discussed.
>
> Concerning **Abbasi-Yadkori**, sorry, indeed, bad wording bounds on my side in Summary, I intended tools as in Abbasi-Yadkori's seminal work.
>
> Based on this reply, and trusting that the remarks suggested by myself and other reviewers will be implemented in the revision, I will rise my score.

---

> > ### Author Response · Authors · 2025-08-08
> >
> > Thank you very much for acknowledging our answer and engaging in the discussion.
> >
> > >I think that elaborating on this more carefully in the revision would help the reader.
> >
> > We agree and we will update the discussion accordingly, in particular, pointing out the significance of the well-specifiedness assumption in this context.
> >
> > > Again, I believe this aspect to be important to clarify, in particular since the test handles distributions via CME while making well-specified assumption for charasteristic kernel. I believe that the limitations of doing so should be clearly discussed.
> >
> > This is indeed a good point and we will make this aspect clearer. In particular, we will stress the expressiveness of the kernel-based test hinges on the CME, and the guarantees (and overcoming the known negative results) are possible due to the well-specifiedness assumption and how this can be a limitation.
> >
> > >Based on this reply, and trusting that the remarks suggested by myself and other reviewers will be implemented in the revision, I will rise my score.
> >
> > Thank you very much. We will carefully revise the manuscript to incorporate these remarks and resolve any remaining concerns brought up in the reviews and the discussion. We really appreciate these suggestion, which will further improve the manuscript.

---

### Official Review · Reviewer_qzic · 2025-07-13

**Clarity:** 2
**Significance:** 3
**Originality:** 3
**Rating:** 5
**Confidence:** 4

**Summary:**

A general framework is proposed for kernel-based conditional two-sample tests that use sequences of input-output measurements, which are not necessarily independent and identically distributed (i.i.d.). The noises are assumed to be sub-Gaussian. The null hypothesis tested is that of equal conditional expectations given a covariate (input). A key advantage is that the output can be abstract, originating from a separable Hilbert space. This feature, combined with the theory of kernel mean embedding, allows the framework to be applied to test other hypotheses, such as those involving higher moments or even full conditional distributions. The paper also introduces the concept of uniform-block-diagonal kernels. An abstract test is first defined by building on finite-sample confidence bounds for learning algorithms. The paper's main results come in the form of confidence bounds for vector-valued kernel ridge regression (KRR) methods, summarized by Theorem 4.3 and later generalized by Theorem D.23 in the supplementary material. Based on these findings, kernel-based two-sample tests are defined and experimentally demonstrated using toy examples.

**Questions:**

- Are these tests mainly useful for small sample settings or do you expect them to be scaleable?
- What is the primary obstacle to achieving a distribution-free test? Is it possible to eliminate the bootstrap step?

**Ethical Concerns:**

["NO or VERY MINOR ethics concerns only"]

**Final Justification:**

The paper's contribution is strong; it provides a new and valuable method for conditional two-sample tests based on kernel techniques. The authors have addressed most of the issues I have raised, with a few smaller issues remaining (such as the practical applicability and scalability of the KRR-based instantiation). I suggest accepting this paper.

**Limitations:**

The paper adequately addresses its limitations; however, a deeper discussion on its practical applicability is noticeably absent.

**Paper Formatting Concerns:**

Minor: many formulas in the supplementary material extend beyond the margins.

**Quality:**

3

**Strengths And Weaknesses:**

Strengths of the paper include:
- The paper addresses the crucial statistical problem of two-sample homogeneity tests (initially, for conditional expectations, which can applied for conditional distributions, as shown in the Appendices), proposing and rigorously studying a novel and promising framework for these problems.
- The approach is highly abstract, allowing for a wide range of design choices. Its flexibility is demonstrated in the appendix, for example, through an example of process monitoring.
- The concepts are clearly defined, and the theorems are precisely stated and proven.
- The results are placed in context, with their connections to existing bounds further discussed in Appendix C.
- The proposed methods apply to non-i.i.d. data and allow online data generation.
- The specific tests and toy examples provided effectively illustrate the approach.
- The code for the numerical experiments is made available online.

Weaknesses of the paper:
- The current title is overly general. A more specific title that highlights that the tests are based on guarantees of vector-valued KRR would be beneficial.
- Subgaussianity is a core assumption of the framework, and therefore, it should be explicitly mentioned in the abstract.
- In Section 4, both the null hypothesis (H_0) and the alternative hypothesis (H_1) should be explicitly defined.
- Curiously, the paper's core contributions appear to be the confidence bounds for vector-valued KRR methods (Theorem 4.3), with the two-sample tests emerging as a straightforward corollary. Yet, the paper identifies the tests themselves as its main contributions.
- The abstract also refers to material which can only be found in the supplementary material (e.g., process monitoring).
- The numerous technical assumptions required may limit the practical applicability of the results beyond toy examples.
- Since the tests are based on kernel methods, their scalability might be an issue.
- Even the applications to the toy examples need a heuristic bootstrap step.
- The conclusion's claim of presenting a complete pipeline from conceptual foundations to practical implementations seems an exaggeration. Practical issues are only briefly touched upon, with, for example, the bootstrap scheme detailed solely in the appendix.
- A minor, yet annoying issue is that many formulas in the supplementary material extend beyond the margins.
- I had the impression that the authors tried to squeeze too much material into the paper, and valuable contributions are therefore placed in the supplementary material, due to lack of space (e.g., generalized confidence bounds for vector-valued least squares).

---

> ### Author Rebuttal · Authors · 2025-07-31
>
> Thank you very much for your thoughtful review.
> We address each of your points below.
> If you find our responses satisfactory, we would be grateful if you would consider raising your score.
>
> > the core contributions are confidence bounds and the tests are corollaries. Yet, the tests themselves are presented as main contributions.
>
> We see two main contributions:
> * **Conceptual**: the testing framework (Sec. 3) and its connection to learning theory (Thm 4.2).
> * **Technical**: new confidence bounds for KRR (Thm 4.3) and for vector-valued least squares (Thm D.23).
>
> The final kernel test then combines these two ingredients. Because the test both motivates the bounds and showcases their practical value, we chose to present all three elements together and build the story of the article around the final test, which is common practice for such bounds; cf. [12, 19, 20].
>
> > overly general title
>
> The title reflects the intersection of our two contributions: a kernel conditional test instantiating our general setup and derived from new bounds. While the bounds are more general, the test is the most practically relevant outcome, so we chose to highlight it in the title.
>
> > technical assumptions and applicability
>
> We have two main technical assumptions:
> 1. **RKHS membership** of the conditional expectation, and
> 2. **subgaussian noise**
>
> A form of the second assumption is common in the learning literature of kernel methods (we discuss this further in a later comment). The validity of the first assumption is more debated [R1] with CMEs, but is generally accepted in the literature on bandits [12,19,20].
> In any case, our Theorem 4.2 and [3] highlight that **such technical assumptions are unavoidable** to limit the hardness of the underlying problem and escape no-free-lunch results.
> It is a known caveat of all of learning theory that such assumptions are hard to enforce or verify. This is not an artifact of our method, however, but a characteristic of conditional testing, since a learning problem is at the heart of conditional testing.
> Crucially, our method *makes the assumption explicit*. Competing tests that invoke asymptotic normality also hinge on similar assumptions --- they are merely implicit in the convergence rate.
>
> Other conditions are less restrictive: choosing a UBD kernel is a design choice, and our online sampling setting relaxes the usual i.i.d. requirement.
>
> > scalability
>
> It is true that our kernel test inherits the computational footprint of KRR and particularly its bootstrapping. However:
> 1. When necessary, our bootstrap can be drop-in replaced with more scalable alternatives (e.g., [R4]) that do not require re-learning the KRR model for each resample. A GPU implementation of [R4] handles $\approx$ 25 k samples in a few seconds, covering many intermediate-scale problems. We will include a presentation of scalable bootstrapping and results with [R4] in the revision.
> 2. Scalability can be further improved with approximate KRR/GP methods, though this breaks the guarantee that the bound has the form $\beta\cdot\sigma(x)$.
> Extending the guarantees to that setting is interesting future work.
> 3. Finally, the kernel instantiation is just that --- an instantiation. The general framework is also applicable to methods that scale better, if they are available. This is left for future work.
>
> > toy examples need a heuristic bootstrap step
>
> Bootstrapping serves two roles:
> 1. removing inaccessible tuning constants, and
> 2. increasing power.
>
> Even when exact parameters are known (Sec. 5), the second point matters. Theorem 4.3 is **worst-case** over the prior Markov-kernel set, so, even if its raw bounds are sharp (which is still an open question to the best of our knowledge), they tend to be conservative. Bootstrapping leverages problem-specific structure for sharper thresholds.
> We emphasize that bootstrapping is also essential in the unconditional case of [1], which our work generalizes.
>
> > claim of presenting a complete pipeline from conceptual foundations to practical implementations seems an exaggeration
>
> Our intention was simply “from theoretical foundations to a usable Python package” (which we provide). We did **not** mean a full end-to-end empirical study. We will clarify this wording and adjust the abstract accordingly.
>
> > paper has too much material; valuable contributions are in the supplementary material
>
> This is a valid point, and we thank you for acknowledging the value of our contributions.
> The density stems from two deliberate choices:
> 1. Demonstrating that the concentration bounds enable a powerful application (conditional testing), as is customary [19, 20].
> 2. Keeping the bounds general, not limiting them to the particular application of KRR of conditional mean embeddings and requiring an abstract Hilbert-valued output. The proof for KRR alone is barely shorter than the full vector-valued least squares version, so separating them would only save little space.
>
> > obstacle to achieving a distribution-free test
>
> We understand this is related to the assumption of subgaussian noise, and not to the assumption of RKHS membership.
> A tail condition on the noise is a common assumption in learning theory, though it does not necessarily take the form of subgaussianity ([Eq. (9), R2], [R3]).
> That said, an important question is what noise exactly the assumption considers.
> Indeed, for the conditional _two-sample_ test, the noise is mapped through the kernel partial evaluations (Section A.1). With a bounded kernel --- standard for CMEs --- the transformed noise is bounded, and subgaussianity only adds a mild trace-class requirement on its covariance.
> Relaxing even this would likely require a new super-martingale $G_N$ (cf. (28)); see [21] for a possible direction.
>
> > Other comments: subgaussianity in the abstract, $H_1$ in (5), abstract mentioning supplementary results, overflowing formulas
>
> We will correct these in the revision and, given the extra space, move the monitoring application from the supplement into the main text.
>
> [R1] Klebanov et al., A rigorous theory of conditional mean embeddings, 2020
>
> [R2] Caponnetto and De Vito, Optimal rates for the regularized least-squares algorithm, 2007
>
> [R3] Mollenhauer et al., Regularized least squares learning with heavy-tailed noise is minimax optimal, 2025
>
> [R4] Singh and Vijaykumar, Kernel ridge regression inference, 2023

---

> > ### Comment · Reviewer_qzic · 2025-08-05
> >
> > Thank you for your responses. I agree with many of your points, for example, regarding the technical assumptions required, the role of bootstrapping, and the two types of contributions you outlined. However, I still believe that the title is overly broad and should be made more specific. I also find the claim that KRR is "just an instantiation" somewhat unconvincing, given that Theorem 4.3, one of your key contributions, applies exclusively to KRR. In this case, the scalability and other limitations of KRR-based approaches raise significant practical concerns. That said, I understand that your general ideas, including Theorem 4.2, can be combined with other learning methods, as well. Overall, I am satisfied with your rebuttal and consider your contributions valuable. I will raise my score, assuming the paper is revised accordingly, including the title, the abstract, and formatting issues, etc.

---

> > > ### Author Response · Authors · 2025-08-08
> > > **Suggested changes and alternative title**
> > >
> > > Thank you for engaging in the discussion and considering to raise your score. We will revise the abstract in accordance with your review and address the formatting issues you noted. While we remain somewhat hesitant to change the title, we understand your concerns about its breadth and are willing to make it more specific. One possibility we are considering is: "Two-sample Testing with Conditional Kernel Mean Embeddings". If you have an alternative title suggestion that you feel better captures the scope and contributions of the paper, we would be happy to consider it.

---

> ### Comment · Reviewer_qzic · 2025-08-08
>
> I do not have a very good suggestion for the title, but I think that it should reflect the main idea of your paper, that is transforming confidence bounds of a learning method into a conditional two-sample test. Mentioning CKMEs might still be too imprecise, but your suggestion is definitely an improvement over the current title. After some thinking, some possible choices could be “A General Framework for Conditional Two-Sample Testing from Learning-Theoretic Guarantees” or “Conditional Two-Sample Testing from Learning-Theoretic Guarantees” or “From Confidence Bounds of Learning Methods to Conditional Two-Sample Tests”. I understand that none of these are perfect, for instance, including kernels would make them more specific, but they might still give you some ideas.

---

> > ### Author Response · Authors · 2025-08-08
> >
> > Thank you very much for clarifying this aspect and your suggestions. After careful deliberation, we would propose "A kernel conditional two-sample test based on confidence bounds for learning methods", since this title focuses on the main aspect (a concrete kernel-based two-sample test), but also sketches our general approach (using confidence bounds for learning methods).

---

> > > ### Comment · Reviewer_qzic · 2025-08-08
> > >
> > > Yes, thank you. This title sounds perfect.

---

> > > > ### Author Response · Authors · 2025-08-09
> > > >
> > > > Thank you very much for this discussion. We would be very grateful if you could consider increasing your score, if all of your concerns have been addressed.

---

### Decision · Program_Chairs · 2025-09-17

**Decision:**

Accept (poster)

**Comment:**

The paper exploits a relationship between conditional two-sample testing and learning-theoretic bounds on estimation error of regression to propose a conditional two-sample testing framework using kernel methods with finite-sample guarantees. The approach builds confidence bounds via kernel ridge regression (KRR), leveraging existing tools like Abbasi-Yadkori's bounds.
I agree with most reviewers that this is a strong contribution to conditional two-sample testing based on kernel methods that makes an elegant connection with learning theory.